# Extravillous trophoblast cell lineage development is associated with active remodeling of the chromatin landscape

Kaela M. Varberg [1,2,10] ✉, Esteban M. Dominguez[1,2,10], Boryana Koseva[3], Joseph M. Varberg [4], Ross P. McNally [1,2,8], Ayelen Moreno-Irusta [1,2], Emily R. Wesley[3], Khursheed Iqbal [1,2], Warren A. Cheung [3], Carl Schwendinger-Schreck[3], Craig Smail[3], Hiroaki Okae[5,9], Takahiro Arima [5], Michael Lydic[6], Kristin Holoch[6], Courtney Marsh[1,6], Michael J. Soares [1,2,6,7] ✉ & Elin Grundberg [1,2,3] ✉

The extravillous trophoblast cell lineage is a key feature of placentation and successful pregnancy. Knowledge of transcriptional regulation driving extravillous trophoblast cell development is limited. Here, we map the transcriptome and epigenome landscape as well as chromatin interactions of human trophoblast stem cells and their transition into extravillous trophoblast cells. We show that integrating chromatin accessibility, long-range chromatin interactions, transcriptomic, and transcription factor binding motif enrichment enables identification of transcription factors and regulatory mechanisms critical for extravillous trophoblast cell development. We elucidate functional roles for *TFAP2C*, *SNAI1*, and *EPAS1* in the regulation of extravillous trophoblast cell development. *EPAS1* is identified as an upstream regulator of key extravillous trophoblast cell transcription factors, including *ASCL2* and *SNAI1* and together with its target genes, is linked to pregnancy loss and birth weight. Collectively, we reveal activation of a dynamic regulatory network and provide a framework for understanding extravillous trophoblast cell specification in trophoblast cell lineage development and human placentation.

Pregnancy disorders such as recurrent pregnancy loss (RPL), preeclampsia, intrauterine growth restriction, and preterm birth can be the result of inadequate delivery of maternal blood to the growing fetus, a consequence of insufficient spiral artery remodeling[1]. Spiral artery remodeling is largely dependent on the function of a specialized population of trophoblast cells termed extravillous trophoblast (EVT) cells[2]. EVT cells are terminally differentiated trophoblast cells that invade into and restructure the uterine compartment, including

[1]Institute for Reproductive and Developmental Sciences, University of Kansas Medical Center, Kansas City, Kansas 66160, USA. [2]Department of Pathology & Laboratory Medicine, University of Kansas Medical Center, Kansas City, KS 66160, USA. [3]Genomic Medicine Center, Children's Mercy Research Institute, Children's Mercy Kansas City, Kansas City, MO 64108, USA. [4]Stowers Institute for Medical Research, Kansas City, MO 64110, USA. [5]Department of Informative Genetics, Environment and Genome Research Center, Tohoku University Graduate School of Medicine, Sendai 980-8575, Japan. [6]Department of Obstetrics and Gynecology, University of Kansas Medical Center, Kansas City, KS 66160, USA. [7]Center for Perinatal Research, Children's Mercy Research Institute, Children's Mercy Kansas City, Kansas City, MO 64108, USA. [8]Present address: Department of Obstetrics and Gynecology, Northwestern University Feinberg School of Medicine, Chicago, IL 60611, USA. [9]Present address: Department of Trophoblast Research, Institute of Molecular Embryology and Genetics, Kumamoto University, 2-2-1 Honjo, Chuo-ku, Kumamoto 860-0811, Japan. [10]These authors contributed equally: Kaela M. Varberg, Esteban M. Dominguez. ✉e-mail: kvarberg@kumc.edu; msoares@kumc.edu; egrundberg@cmh.edu

transforming tightly coiled spiral arterioles into large, distended vessels capable of increasing blood delivery to the developing fetus[3]. EVT cells take two routes of invasion into the uterine parenchyma: interstitial and endovascular. Interstitial EVT cells migrate within the stromal compartment between the vasculature, whereas endovascular EVT cells invade into vessels where they replace endothelial cells and adopt a pseudo-endothelial phenotype[2,4]. In addition to increasing blood volume, EVT cell-remodeled spiral arteries decrease blood flow velocity, and consequently shear stress damage to uterine vessels[5]. The contributions of EVT cells to uterine transformation and successful pregnancy imparts importance to understanding the acquisition of an EVT cell fate. To date there is a limited knowledge of regulatory networks controlling EVT cell differentiation.

Comparisons of rodent embryonic stem (ES) and trophoblast stem (TS) cells using an assortment of approaches to characterize the transcriptome and epigenome have provided insight into the regulatory landscape critical for establishment of the trophoblast cell lineage[6–14]. This knowledge helped identify core transcription factors (TFs) essential for reprogramming mouse fibroblasts into TS cells[15,16]. Advancements have also been achieved in understanding higher-order regulatory networks controlling aspects of rodent trophoblast cell differentiation, including the identification of putative TF-gene regulated networks[8,12,17–20]. The execution of similar experimentation regarding human trophoblast cell development would be informative.

A range of in vitro model systems for investigating human trophoblast cell lineage development have been proffered but all have fallen short in adequately replicating establishment of the EVT cell lineage[21]. A fundamental problem has been in distinguishing features intrinsic to EVT cells from those associated with the immortalization or transformation processes used to establish the in vitro models. In 2018, culture conditions needed to maintain human TS cell renewal and to differentiate TS cells into EVT cells were established[22]. The human TS cell model is robust and has been utilized to provide insights into the development of the human trophoblast cell lineage[20,23–26]. However, systematic profiling of open chromatin, chromatin contacts, and associated transcriptional signatures in human TS cells and differentiated EVT cells have not been performed to date.

To this end, we applied multiple functional genomics approaches using next-generation sequencing (NGS) and contrasted the regulatory landscape of trophoblast cells in the stem state and following differentiation into EVT cells using human TS cells as a model[22]. The experimentation included systematic assessments of the transcriptome by RNA-sequencing (RNA-Seq), chromatin accessibility using Assay for Transposase-Accessible Chromatin-sequencing (ATAC-Seq), and high throughput chromosome conformation capture (Hi-C), respectively. We integrated these analyses to identify higher-order regulation of human EVT cell differentiation, which was validated by histone modifications (chromatin immunoprecipitation-sequencing, ChIP-Seq) in the TS cell model and single-cell RNA sequencing (scRNA-Seq) of first trimester human placenta samples. Collectively, the analyses provide a rich resource for dissecting functional genomics of placental-related diseases of pregnancy. Using this resource, we identified candidate TFs and demonstrated their critical role in EVT cell lineage development with links to pregnancy complications.

## Results

### Generating an atlas of regulatory DNA and transcriptional profiles of human TS and EVT cells

We developed a comprehensive functional genomics resource of trophoblast cell development by utilizing human TS cells maintained in the stem state and following differentiation to EVT cells[22] (Fig. 1a). We initially performed a time course analysis monitoring the transition from stem state towards EVT cells at four timepoints [stem state (day 0), day 3, day 6, and day 8]. We noted clear differences in global gene expression at day 0, 3, and 6 (Fig. 1b). Expression profiles were more

similar at days 6 and 8 of EVT cell differentiation. The upregulated genes included expression of commonly used EVT lineage markers[20] (Fig. 1c, Supplementary Fig. 1a), as well as induction of *HLA-G* (Fig. 1d, Supplementary Fig. 1b, Supplementary Fig. 2). Similarly, morphological changes as TS cells transition from discrete colonies into individual, elongated EVT cells were verified and most pronounced at day 8 (Fig. 1e, Supplementary Fig. 1b). Based on these results, we focused on the stem state and EVT cells following eight days of differentiation (day 8) and generated RNA-Seq and ATAC-Seq profiles for multiple replicates of two TS cell lines: CT27 (46, XX) and CT29 (46 XY) (Supplementary Data 1). Overall, we identified 2938 transcripts that were significantly upregulated (log2 fold change >1, adjusted $p < 0.05$) in EVT cells compared to trophoblast cells in the stem state and 3055 transcripts that were significantly downregulated (Supplementary Data 2, Fig. 1f). The robustness and consistency of the TS cell model was confirmed in multiple ways. First, TS cell transcriptomes obtained from independent experiments were contrasted to previously reported transcriptomes for the same TS cell line[22] (Pearson correlation 0.64) (Supplementary Fig. 3a, b). Second, transcriptomes and active regulatory regions as assessed by RNA-Seq and ATAC-Seq, from CT27 and CT29 TS cell lines were compared across cell states showing high concordance (Supplementary Fig. 3c–e). Finally, we used publicly available single-cell gene expression data from EVT cells ($n = 693$ independent cells) isolated from normal first trimester placental samples[27,28] and contrasted with expression profiles of a combined group of other trophoblast cells with markers of syncytiotrophoblast (SCT) and villous cytotrophoblast (VCT) ($n = 7834$ independent cells) (Fig. 2a). We found high concordance of differentially expressed genes in the TS cell culture model with the in vivo data (Fig. 2b; Supplementary Fig. 4). Similarly, gene set enrichment analysis (GSEA) using the top 250 in vivo markers of the SCT/VCT and EVT lineages, respectively, from the single-cell gene expression data revealed statistically significant enrichment of expression patterns in the in vitro TS cell model corresponding to false discovery (FDR) q value of 0.2% for both EVT and stem state cells GSEA, respectively (Fig. 2c–e).

Thus, the in vitro human TS cell culture model can be successfully leveraged to gain insight into human EVT cell lineage development. We use CT27 as our reference cell line for all subsequent global and targeted analysis with orthogonal validation in CT29 accordingly.

### Co-occurrence of EVT cell differentiation and global changes in chromatin accessibility

Although reference maps of regulatory DNA have been generated across hundreds of human cell and tissue types and states, trophoblast cell lineage representation within these reference maps is sparse and is restricted to either bulk placental samples or embryonic stem cell-derived trophoblast cells without distinction of specific cell lineages or developmental states[29,30]. Using ATAC-Seq at high sequencing depth across multiple replicates, we have expanded these resources and provided maps of regulatory regions in the trophoblast cell stem state and following differentiation to EVT cells. We analyzed these regulatory regions in multiple ways. First, we called ATAC-Seq peaks, which are markers of open chromatin, jointly across replicates and identified, at 1% false discovery rate (FDR), 165,094 (Supplementary Data 3a) and 142,520 (Supplementary Data 3b) open chromatin regions in stem and EVT cell developmental states, respectively. We then conservatively assessed shared accessible regions by overlapping specific coordinates and included a minimum one base pair (bp) overlap. This conservative assessment identified 63% versus 73% of the accessible regions as shared and 37% versus 27% of the regions as unique to the stem state versus EVT cell state, respectively (Fig. 3a). The accessible regions that were shared between stem and EVT cells were distributed similarly across the genome compared to other reported human tissues and cells, with the majority mapping to intronic (44%) or intergenic (32%) regions and a smaller subset

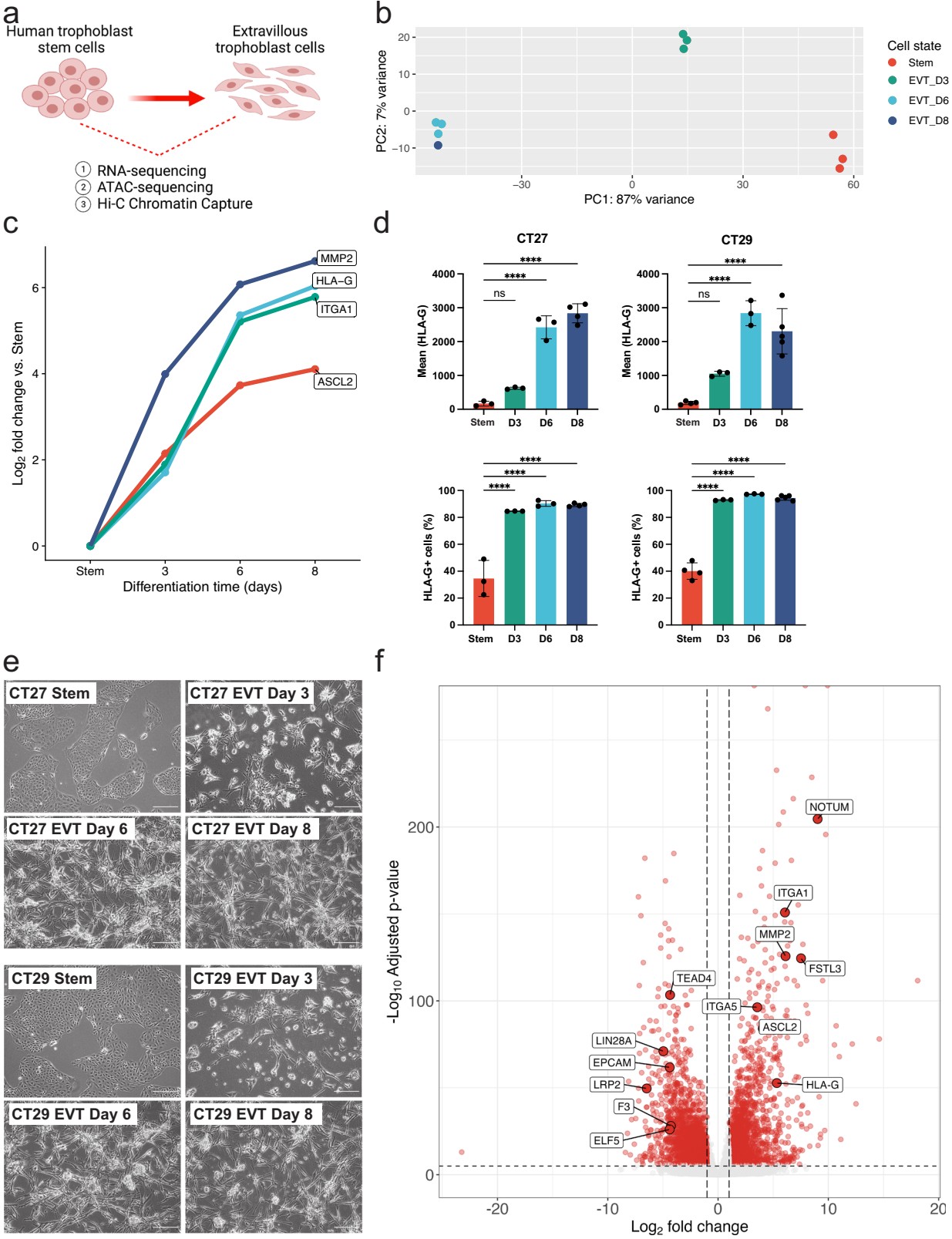

mapping to promoter regions (14%) (Fig. 3b). However, when contrasting stem versus EVT cell-specific regions, we noted that the genomic distribution of EVT cell-specific regions was shifted, with a larger proportion mapping to intergenic regions (40% versus 34%) and a smaller proportion mapping to promoter regions (4% versus 8%). This shift points toward increased enhancer-driven gene regulation in EVT versus stem state cells (Fig. 3b). These observations are reinforced

by contrasting sequence coverage in ATAC-Seq, where the coverage is enriched near transcription start sites (TSS) for shared accessible regions, but not for EVT cell-specific regions (Fig. 3c).

We expanded our annotation of stem and EVT cell open chromatin regions to include the atlas of regulatory DNA that was created from 16 different human tissue types or states[29]. This extensive atlas was used as a comparative reference standard. We found that as many

**Fig. 1 | Morphologic and transcriptomic changes during EVT cell differentiation. a** Schematic depicting three primary analyses (RNA-Seq, assay for transposase-accessible chromatin (ATAC)-Seq, and Hi-C) performed on human TS cells in the stem state and following eight days of extravillous trophoblast (EVT) cell differentiation (BioRender; $n = 1$ biological replicate of stem and EVT cells depicted. **b** Principal component analysis (PCA) plot depicting stem (red), EVT Day 3 (green), EVT Day 6 (light blue), and EVT Day 8 (dark blue) data of normalized read counts from RNA-Seq datasets. **c** Log$_2$ fold-change values of normalized read counts from RNA-Seq of EVT cell-specific transcripts (*ASCL2, HLA-G, ITGA1,* and *MMP2*) at stem state and on day 3, 6, and 8 of EVT cell differentiation. **d** Mean *HLA-G* expression/cell (top) and the percentage (%) of HLA-G positive cells detected by flow cytometry in CT27 (left) and CT29 (right) cells in the stem state and on days 3, 6, and 8 of EVT cell differentiation ($n = 3$ biologically independent replicates for CT27 stem and CT27 and CT29 EVT days 3 and 6; $n = 4$ biologically independent replicates for

CT29 stem and CT27 EVT day 8; $n = 5$ biologically independent replicates for CT29 EVT day 8; ****$p < 0.0001$). Data were analyzed using a one-way ANOVA followed by Dunnett's multiple comparisons test and are presented as mean values ± standard deviation (SD). **e** Phase contrast images of TS cells cultured in the stem state and on days 3, 6, and 8 of EVT cell differentiation in CT27 and CT29 cell lines. Scale bars represent 500 µm. **f** Volcano plot depicting transcriptomic changes in EVT cells (CT27 and CT29) on day 8 of differentiation compared to stem state cells (CT27 and CT29) measured by RNA-Seq. Differentially expressed transcripts (absolute log2 fold change >1, adjusted *p*-value < 0.05) are labeled in red obtained by hypothesis testing by a Wald test followed by Benjamini and Hochberg procedure to adjust for multiple testing. Key stem state (*ELF5, EPCAM, F3, LIN28A, LRP2,* and *TEAD4*) and EVT cell (*ASCL2, FSTL3, HLA-G, ITGA1, ITGA5, MMP2,* and *NOTUM*) transcripts are annotated. Source data are provided as a Source Data file.

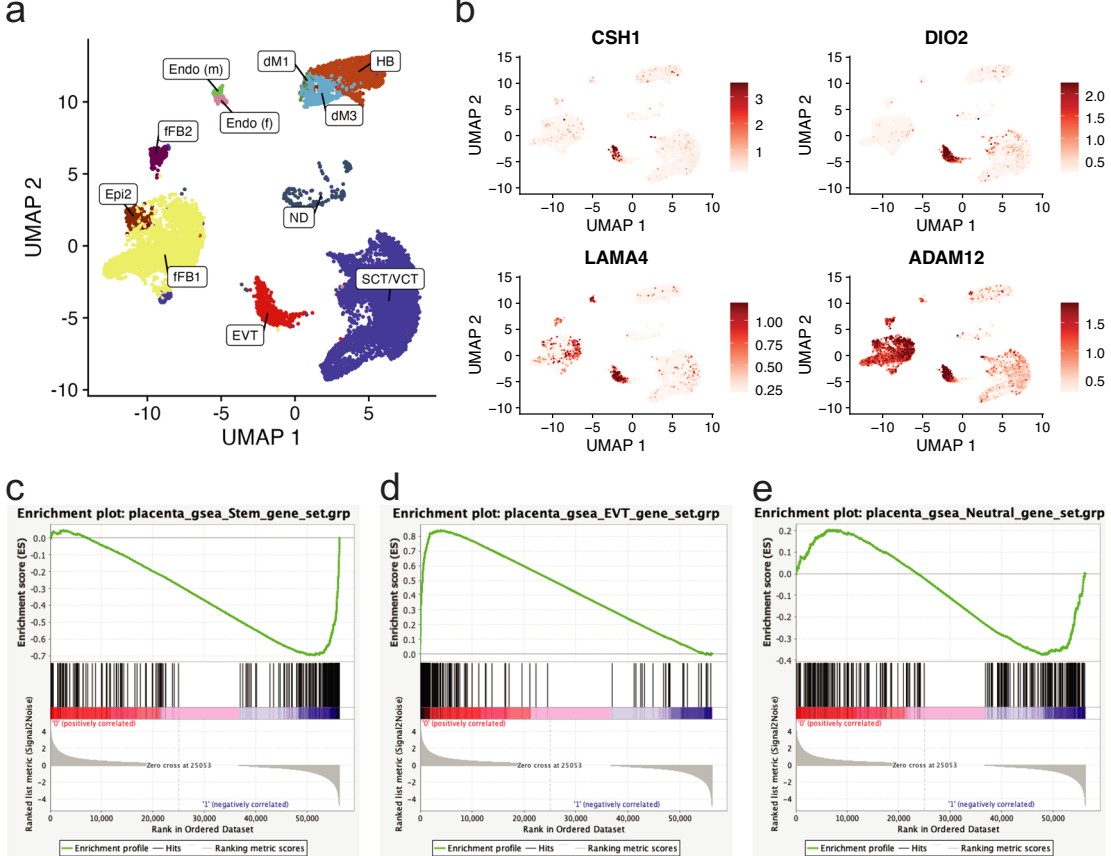

**Fig. 2 | Validation of in vitro signatures of the TS cell model in single cell datasets from human placental samples.** UMAP plots of single cell expression values (normalized, scaled, and natural-log transformed) depicting cell clustering from two publicly available, independent, single-cell RNA sequencing (scRNA-Seq) datasets from first-trimester placentas[27,28] for (**a**) all cell types inferred using marker genes published previously[28] as decidual macrophages (dM), fetal (Endo f) and maternal endothelial cells (Endo m), epithelial glandular cells (Epi), extravillous trophoblast cells (EVT), fetal fibroblasts (FB), Hofbauer cells (HB), syncytiotrophoblast and villous cytotrophoblast (SCT/VCT) and cells not determined (ND) and (**b**) EVT cell-specific genes detected using an in vitro model (red color scale of expression values thresholded at the 5th and 95th percentiles). **c–e** Plots depicting gene set enrichment analysis using three genes sets (each $N = 250$) of

genes significantly upregulated in (**c**) SCT/VCT cells or (**d**), EVT cells or (**e**) genes not differently expressed between SCT/VCT and EVT cells from scRNA-Seq data sets and tested for enrichment for expression pattern in stem state (phenotype 1, CT27) and EVT cells (phenotype 0, CT27) from the in vitro TS cell model. Enrichment score (ES) reflects the degree of which each gene set is over-represented at the extremes (top or bottom) of genes expressed in phenotype 0 (EVT cells) versus phenotype 1 (stem state cells). Estimated significance of ES were accounted for multiple testing and corresponded to (**b**) Phenotype 0; FDR *q* value = NS; Phenotype 1; FDR *q* value = 0.00190 (**c**) Phenotype 0; FDR *q* value = 0.00196; Phenotype 1; FDR *q* value = NS and (**d**) Phenotype 0; FDR *q* value = NS; Phenotype 1; FDR *q* value = 0.0016. FDR = false discovery rate; NS = not significant.

as 97% of shared stem and EVT cell regions were previously annotated to regulatory DNA, including 11% that were specific to placental or embryonic/primitive origin. Interestingly, within accessible chromatin regions that were unique to EVT cells, 17% were placental-or embryonic/primitive specific (Supplementary Data 3a, b).

To characterize the regulatory regions identified by ATAC-Seq in further detail we integrated ChIP-Seq based histone modification data for stem state and EVT cells and noted enrichment of ATAC-Seq coverage near H3K27ac and H3K4me3 peaks for both stem state and EVT cells, respectively (Supplementary Fig. 5a)

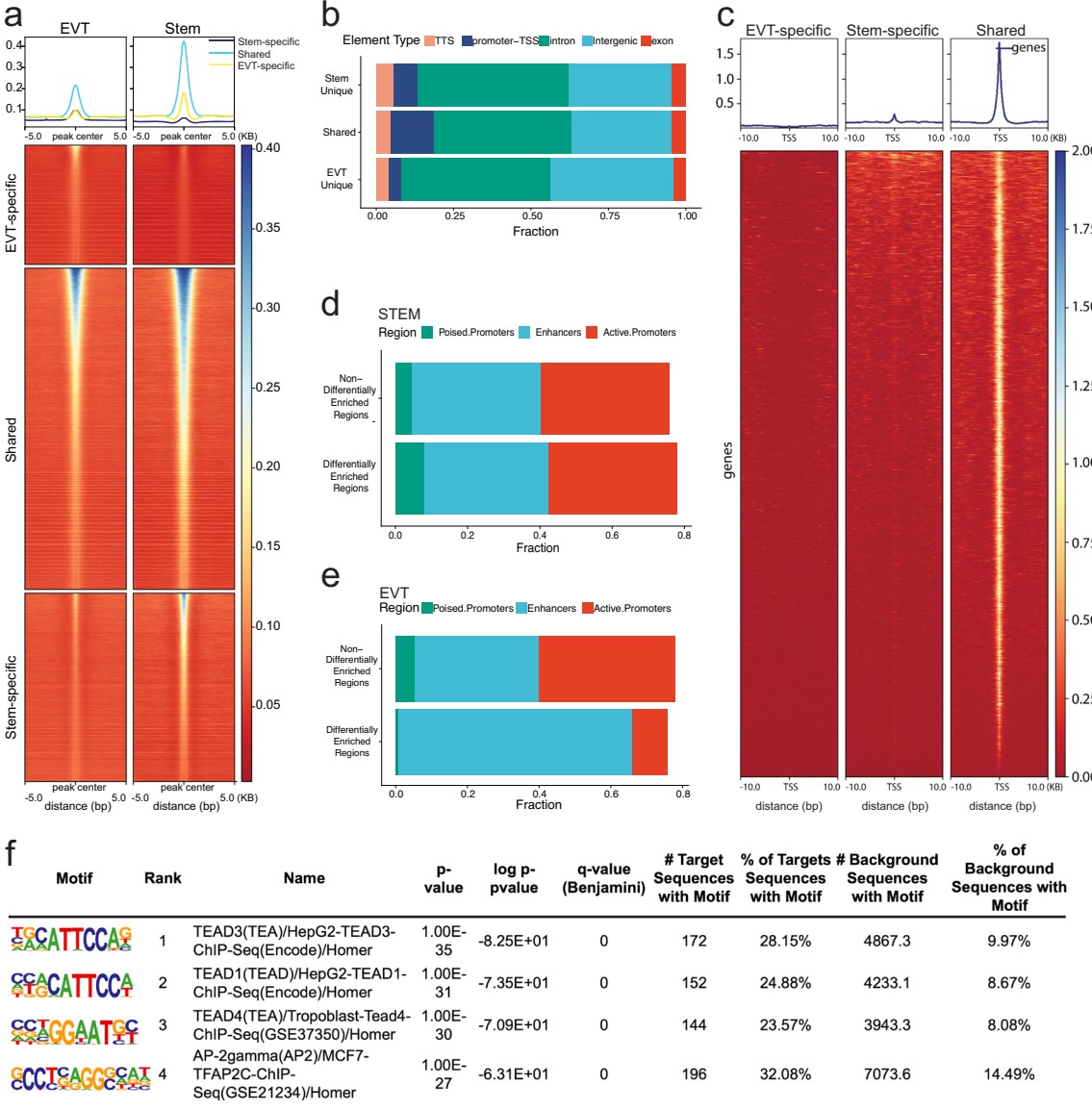

**Fig. 3 | EVT cell differentiation is associated with global changes in chromatin accessibility. a** Density plot and heatmap of coverages from extravillous tropho-blast (EVT) and stem state assay for transposase-accessible chromatin (ATAC)-seq assays showing stem-specific (dark blue), shared (light blue), and EVT-specific (yellow) regions centered around: EVT-specific peaks (top, ranked based on peak score from EVT ATAC-Seq in descending order), shared peaks across cell states (middle, ranked based on peak score from EVT ATAC-Seq in descending order) and stem state-specific peaks (bottom, ranked based on peak score from stem state ATAC-Seq in descending order), respectively. Each row represents one genomic region centered around (5kb) a peak and the depth of coverage is encoded in a colorimetric scale (red to blue corresponding low to high coverage). **b** Bar graphs depicting the fraction of chromatin accessible regions across five defined genomic regions (e.g., transcription termination site (TTS; pink), promoter-transcription start site (TSS; dark blue), intron (green), intergenic (light blue), and exon (red) regions). Chromatin accessible regions identified by ATAC-Seq are classified as

regions (1) unique to stem state cells, (2) shared between stem and EVT cells or (3) unique to EVT cells. **c** Density plot and heatmap of coverages from EVT and stem state ATAC-Seq assays limited to genomic regions in EVT-specific peaks (left, ranked based on coverage in EVT ATAC-Seq in descending order), stem state-specific peaks (middle, ranked based on coverage in stem state ATAC-Seq in des-cending order) and shared peaks across cell states (right, ranked based on coverage in EVT ATAC-Seq in descending order). Each row represents one genomic region centered around (10 kb) a TSS and the depth of coverage is encoded in a colori-metric scale (red to blue corresponding low to high coverage). **d, e** Bar graphs depicting the fraction of differentially enriched chromatin accessible regions identified by ATAC-Seq in stem state (**d**) or EVT (**e**) cells across three chromatin states defined by histone modifications: H3K4me3 (putative poised promoters; green), H3K27ac (active enhancers; light blue) and H3K4me3+H3K27ac (active promoters; red). **f** Schematic depicting transcription factor (TF) analysis results and motifs for top identified TFs.

To identify consensus regulatory regions that display different chromatin accessibility as an indication of different transcription fac-tor (TF) binding patterns in stem versus EVT cells, we next applied an alternative approach that utilized individual ATAC-Seq replicates in stem state cells ($n = 7$ biologically independent replicates) and EVT cells ($n = 7$ biologically independent replicates). The consensus regulatory regions we considered in this analysis were (1) covered by at least 10 counts per million reads, (2) fixed windows of 400 bp in size,

(3) mapped to an ATAC-Seq peak, and (4) appeared in more than two of the samples. For the 41,465 identified consensus regions, 40,508 and 38,785 mapped to ATAC-Seq peaks in stem state and EVT cells, respectively, and the individual replicates clustered together with the largest amount of variability attributed to differences between the two cell groups (Supplementary Fig. 5b, c). Based on estimated binding affinity measured by differences in ATAC-Seq read densities, we identified a total of 23,504 and 9334 regions that were differentially

enriched (5% FDR) and considered more accessible in the stem state and EVT cell state, respectively. We annotated these regulatory regions using the orthogonal ChIP-Seq based histone modification data for active enhancers (H3K27ac) and promoters (H3K4me3) (Supplementary Data 4a, b). In all, we found 78% versus 76% of the differentially enriched ATAC-Seq regions to map to an active enhancer (H3K27ac) or promoter (H3K4me3) in stem state versus EVT cells, respectively. However, when considering the epigenetic marks separately we noted a clear difference across the two cell states. Specifically, while 36% of differentially enriched ATAC-Seq regions in the stem state overlapped an active enhancer (Fig. 3d) as many as 65% of the differentially enriched ATAC-Seq regions in the EVT cell state overlapped an active enhancer marked by a H3K27ac peak (Fig. 3e). Interestingly, differentially enriched ATAC-Seq regions in the stem state were an order of magnitude more likely to map to a putative poised/bivalent promoter (marked by a H3K4me3 peak without an overlapping H3K27ac peak) than differentially enriched ATAC-Seq regions in the EVT cell state (Fig. 3d, e). Bivalent chromatin is characterized by having both repressing and activating roles in gene expression regulation with an important role in cell differentiation[31]. Although we note that the repressive chromatin mark H3K27me3 is required to fully distinguish a poised/bivalent promoter we speculate that these enriched promoter regions in the stem state may silence genes that will activate EVT cell differentiation as exemplified for *HLA-G* (Supplementary Fig. 6b and Supplementary Fig. 7a). On the other hand, the enrichment of H3K27ac peaks among the differentially enriched ATAC-Seq regions in the EVT cell state further point to significant increased enhancer-driven gene regulation in EVT versus stem state cells[31].

For the EVT cell differentially accessed regulatory regions, we integrated expression data measured by RNA-Seq for regions mapping within 10 kb of the TSS for a gene ($n = 1576$ genes) and also found a striking enrichment of EVT cell-upregulated genes mapping near these EVT cell differentially accessed regulatory regions (log2-fold $> 1$; adjusted $p < 0.05$), which corresponded to 2.6-fold enrichment (Chi$^2 = 615$; Fisher $p < 0.0001$) (Supplementary Data 5). We noted that many of these differentially bound active regulatory regions linked to differentially expressed genes were those known to be implicated in the regulation of intrauterine trophoblast cell invasion (e.g., *MMP2* and *ASCL2*) (Supplementary Fig. 6a–c and Supplementary Fig. 7a–c).

## Unique gene regulation machinery in EVT cells includes long-range chromatin interactions

Regulatory DNA that is specific to EVT cells can provide insight into potential transcriptional regulators of EVT cell differentiation and its associated gene expression profile. Due to the significant enrichment of EVT cell-specific regulatory regions near genes with EVT cell-specific expression patterns, we hypothesized that a subset of these regions would map to super-enhancers. A super-enhancer is operationally defined as a large cluster of transcriptional enhancers in close proximity that drive expression of a gene(s)[32]. Super-enhancers determine cell identity and can provide insight regarding key TFs underlying the differentiation process[12,33]. To test this, we used the ATAC-Seq data to identify super-enhancers based on established algorithms[34]. We identified a total of 1283 super-enhancers, of which 309 mapped to 611 differentially bound ATAC-Seq peaks (Supplementary Data 6).

To identify long-range chromatin interactions involving EVT cell super-enhancers and regulatory regions identified by ATAC-Seq we performed Hi-C chromatin capture. We obtained long-range intra-chromosomal contacts after aligning and filtering Hi-C read pairs across all Hi-C samples and called chromatin loops at 5 kb resolution. Using this threshold, we identified 11,999 stem state (Supplementary Data 7a) and 14,516 (Supplementary Data 7b) EVT cell chromatin loops with a minimum and median distance between anchors corresponding

to 30 kb and 260–275 kb, respectively. These observations are consistent with efforts using other cell models[35,36]. We found that CCCTC-binding factor (CTCF) binding motifs were the most enriched TF binding motifs within the loop anchors (covering 60% of all EVT loop anchors). This observation confirms previous efforts that indicate the important role of CTCF in shaping the three-dimensional structure of the genome[37,38], in addition to the function it serves as a transcriptional repressor and insulator[35]. The validity of the Hi-C chromatin capture was confirmed in the CT29 donor line showing high concordance for both stem state and EVT cell chromatin loops, respectively (Supplementary Fig. 8a).

To identify long-range chromatin interactions, or Hi-C loops, that are specific to either stem state or EVT cells (i.e., differential loops), Hi-C loops from the stem state were integrated using the same parameters described for EVT cells above (Fig. 4a, b). Using previously established algorithms[39], we identified 99 and 349 loops to be stem- and EVT-cell specific, respectively (Supplementary Data 7a, b). The latter set of differential loops was also shown to be more robust as shown in orthogonal validation analysis (Supplementary Fig. 8b). These results point to two major properties of cell-specific, Hi-C loops: (1) they are rare due to strongly conserved chromatin conformation across cell states as shown previously[39] (Fig. 4a, b) and (2) they are more common in EVT cells, which could be indicative of increased enhancer-driven gene regulation (Fig. 3c).

To examine whether the loop anchors identified across the cell states are enriched for regulatory regions and differentially expressed genes we integrated ATAC-Seq and RNA-Seq data from the respective cell types. Among all chromatin loop anchors, we found a clear enrichment of accessible chromatin near the center of the anchors (Fig. 4c, d) for each cell state. EVT cell-specific loop anchors displayed a similar pattern; however, there was no enrichment of accessible chromatin near the center of the anchors for stem state-specific loops likely due to the low number of loops that were identified (Supplementary Fig. 9a–d). We also were not able to detect an enrichment of cell type-specific chromatin accessibility data at cell-specific anchors (Supplementary Fig. 9a–d). We then narrowed in on the 9334 EVT cell differentially accessed regulatory regions (Supplementary Data 4b) and showed in aggregate analysis a similar enrichment of these near the center of EVT cell loop anchors (Supplementary Fig. 9e). Specifically, we found that 1239 EVT cell-specific regions defined by ATAC-Seq overlapped at least one Hi-C loop anchor across a total of 1518 EVT cell Hi-C loops. Interestingly, of the 1518 EVT cell chromatin loops identified, 119 contained differentially bound regulatory regions at both anchors and 1094 had at least one anchor mapped within 10 kb of a known gene (Supplementary Data 8a, b). Among the genes identified with long-range chromatin interactions of regulatory DNA, several have been implicated in EVT cell differentiation and function including *ASCL2*, *MMP2*, and *HLA-G*[20,40–42] (Supplementary Fig. 10a, b and Supplementary Fig. 11a, b). Specifically, we identified a super-enhancer 50 kb upstream of *ASCL2* interacting with its promoter region (Supplementary Fig. 12).

Next, we integrated gene expression data from stem state and EVT cells, respectively, mapping the four different sets of loop anchors. We found that loop anchors were enriched (Chi$^2$ test $p < 10^{-4}$) near differentially expressed genes (adjusted $p < 10^{-5}$) and the enrichment was marginally stronger for genes upregulated in the respective cell state (Supplementary Fig. 9e, f). Interestingly, this trend was more pronounced for cell-specific (differential) loop anchors especially those identified in EVT cells (Fisher's exact test $p < 0.05$) (Fig. 4e, f). Among these upregulated genes with long-range chromatin interactions identified only in EVT cells was *DLX6* (Fig. 4g)—a transcriptional regulator of cell differentiation during embryonic development[43]. Taken together, these results confirm previous findings[36] that cell state-specific (differential) loops associate with cell-type specific regulation and functions.

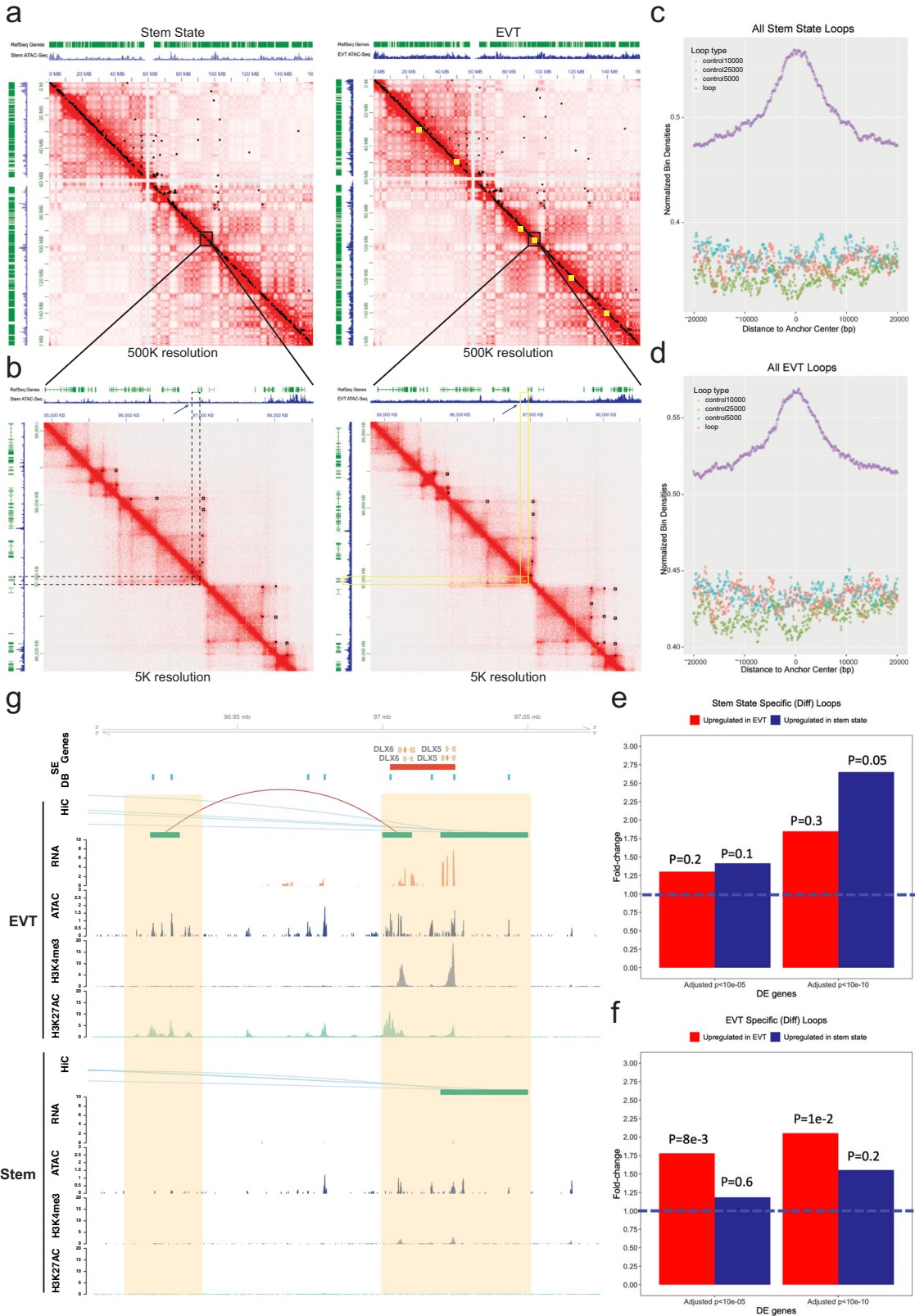

## Candidate transcriptional regulators of EVT cell differentiation

To identify transcriptional regulators of EVT cell differentiation we utilized the functional genomics resource in two ways. First, we used three sets of regulatory regions to perform motif enrichment of known TF families compared to the genome background. The regulatory regions included: (1) all differentially enriched EVT cell ATAC-Seq peaks based on estimated TF binding affinity, (2) differentially enriched EVT cell ATAC-Seq peaks based on estimated TF binding affinity mapping to an active enhancer, and (3) regions in (1) that overlap a super-enhancer (Fig. 3f). In all three analyses (Supplementary Data 9a–c), we found TEA Domain Transcription Factor 3 (TEAD3) and TEA Domain Transcription Factor 1 (TEAD1) as the top two ranked TF DNA binding motifs where both genes were also significantly upregulated in EVT cells (*TEAD3*: log2fold change = 1.6, adjusted *p* = 2E−10;

**Fig. 4 | Long-range chromatin interactions in stem state and EVT cells.** Contact matrices (chromosome 7) from stem (left) and extravillous trophoblast (EVT) cell (right) Hi-C contact maps. Pixel intensity represents a normalized (balanced) number of observed loci pair contacts. Contact maps are annotated with RefSeq gene positions (green tracks) and cell state-specific assay for transposase-accessible chromatin (ATAC)-Seq assay signals (blue tracks, counts per million (CPM) mapped reads, $y$-axis). Selected regions exemplify the quantity of all loop anchors (black dots) versus EVT cell state-specific loop anchors (yellow squares) for a region mapping near an EVT cell-state specific transcription factor. Contact maps include (**a**) 500 kb whole chromosome and (**b**) 5 kb resolution region subsets. EVT cell-specific (differential) loops (yellow) with one anchor overlapping an EVT-specific regulatory element indicated by ATAC-Seq data (blue) with corresponding stem state regions (dashed lines). Density plots show open chromatin distribution relative to loop anchor centers (bp, $x$-axis) for all (**c**) stem and (**d**) EVT cell loops (purple). Normalized bin densities shown for every 10 million total reads ($y$-axis) measured by ATAC-seq. Regions below peak signals represent 1000 randomly selected control sets that do not intersect loops with 5 kb (blue), 10 kb (red) and 25 kb (green) region sizes. Proportion of differential loop anchors in stem (**e**) and EVT (**f**) cells that mapped to a nearby (within 10 kb) gene. Enrichment of significantly upregulated genes (RNA-Seq) shown for stem state (blue) and EVT (red) cells presented as fold-change ($y$-axis). Loop anchors link to significant vs non-significant genes (dotted line). $P$-values obtained from two-sided Fisher's exact test. **g** Genomic sub-region highlighted in (**b**) (yellow box) annotated with Hi-C, RNA-Seq, ATAC-Seq, H3K4me3 and H3K27ac ChIP-Seq assessments in EVT (top) or stem state (bottom) CT27 cells. Regulatory elements near DLX5/DLX6 include Hi-C loops (red, both loop anchors in view; blue, loop anchors out of view), RNA-Seq (transcripts per million, $y$-axis), ATAC-Seq (CPM mapped reads, $y$-axis), H3K4me3, and H3K27ac ChIP-Seq (CPM mapped reads, $y$-axis) tracks. All datasets represented by individual tracks. Super-enhancers (SE, red) and differentially bound regions (DB, blue) are specific to EVT cells. Yellow highlights regulatory regions overlapping loop anchors.

*TEAD1*: log2 fold change = 1.3, adjusted $p$ = 1E−3; Supplementary Data 2). In addition to TEAD1 and TEAD3, another 18 TF DNA binding motifs were identified among the top-ranking TFs across all sets, including Activator Protein 1 (AP-1), Activating Protein 2 (AP-2), and GATA TF DNA binding motifs (Supplementary Data 10). TEAD1[44] and TEAD3 have been linked to trophoblast cell regulation, and a paralog, TEA Domain Transcription Factor 4 (TEAD4), is a known determinant of the stem state[24,45]. *TEAD1* and *TEAD3*, each displayed unique regulatory landscapes specific to the EVT cell state (Supplementary Fig. 13a, b and Supplementary Fig. 14a, b). *TEAD1 and TEAD3* are more highly expressed in EVT cells compared to stem state cells, while *TEAD4* is expressed more highly in the stem state (Supplementary Fig. 13c).

AP-2 and GATA TFs are known regulators of trophoblast cell development[46,47]. Indeed, the most significant motif enrichment among EVT-specific super-enhancers following those for the TEAD family members was TFAP2C (also referred to as AP-2 gamma), which has previously been implicated in the regulation of early decision making in trophoblast cell lineage development[47] (Supplementary Data 9c). However, the potential contributions of TFAP2C to human TS cell maintenance and terminal trophoblast differentiation remain elusive.

In our second attempt to identify transcriptional regulators of EVT cell differentiation, we followed-up the top 100 EVT cell-specific chromatin loop-gene associations (Supplementary Data 11) and used the single-cell gene expression data generated from normal first trimester placental samples for validation (Fig. 2a). Approximately 70% of these genes that were found to possess EVT-cell specific gene regulation and significantly upregulated in TS cell-derived EVT showed increased expression in the primary EVT cells from the single-cell dataset in comparison to SCT and VCT cells (Supplementary Data 11). We classified these EVT cell-specific genes based on TF status and revealed five TF genes (*ASCL2, DLX6, SNAI1, MYCN,* and *EPAS1*) (Fig. 5). The regions surrounding each of the five TFs show evidence of EVT cell-specific regulation (Figs. 4e, 8e, 9e, Supplementary Fig. 12, Supplementary Fig. 15a, Supplementary Fig. 16a–c, Supplementary Fig. 21a). While ASCL2 contributes to EVT cell differentiation[20], the contributions of the other identified TFs in trophoblast cell differentiation are less understood.

We investigated the role of TFAP2C, SNAI1, and EPAS1 in the regulation of trophoblast cell development and EVT differentiation.

## Functional investigation of transcriptional regulators of EVT cell lineage development

To evaluate the functions of TFAP2C in trophoblast cell development, we first localized *TFAP2C* in placental specimens and then performed knockdown experiments in the TS cell model. Using in situ hybridization, we found *TFAP2C* to be expressed within the EVT cell column of first trimester human placental specimens (Fig. 6a). *TFAP2C* co-localized primarily with *CDH1* (cytotrophoblast marker[48]) and less with *PLAC8* (EVT cell marker)[40,49] transcripts in EVT cell columns, consistent with its enrichment in the cytotrophoblast/trophoblast progenitor cell population at the base of the column (Fig. 6a). To determine expression profiles in the TS cell model, we applied RT-qPCR and western blotting to measure *TFAP2C* transcript and TFAP2C protein levels. *TFAP2C* transcript is expressed in stem cells and EVT cells (at 8 days) at comparable levels (Fig. 6b), with slightly higher expression of TFAP2C at the protein level in the stem state (Fig. 6c). However, results from the time-course analysis were more dynamic and showed a marked upregulation of *TFAP2C* at an early transitional stage in EVT cell differentiation (day 3) (Fig. 6d). These observations suggest that TFAP2C could be involved in both maintenance of the trophoblast cell stem state and the early stages of EVT cell differentiation.

The regulatory landscape surrounding the *TFAP2C* locus shows evidence of active chromatin near the transcription start site (TSS) of the gene in both stem state and EVT cells. (Supplementary Figs. 17 and 18). To elucidate the function of TFAP2C in TS cell differentiation we used a gene silencing approach. *TFAP2C*-specific short hairpin RNA (shRNA) stably delivered in the stem state interfered with *TFAP2C* transcript (Fig. 6e) and TFAP2C protein expression (Fig. 6f). TFAP2C disruption inhibited TS cell proliferation, altered TS cell morphology (Fig. 6g), and adversely affected cell survival in culture conditions promoting EVT cell differentiation. The effects of TFAP2C disruption were also observed in the CT29 line (Supplementary Fig. 19). Disruption of TFAP2C resulted in broad transcriptomic changes, as identified by RNA-Seq. Consistent with impaired cell proliferation, several known genes contributing to cell cycle progression were downregulated, including *AURKA, AURKB, PLK1, MYC, EGR1, CDK1, CCNA2, CCNB1,* and *MKI67* (Fig. 6h). Transcript signatures indicative of the stem state (e.g., *TEAD4, PEG3,* and *PEG10*) and EVT cell state (e.g., *IGF2, ISM2, CDKN1C, FOXO4, GCM1, ASCL2, SNAI1, FSTL3, PLAC8,* and *NOTUM*) were also inhibited (Fig. 6h). Notable upregulated genes following TFAP2C disruption included *CNN1* and *LAMB3*, which are associated with the stem state, as well as *MMP2, MCAM,* and *SPARC*, which are linked to the EVT cell state. Ingenuity pathway analysis of the top 500 differentially regulated genes following TFAP2C disruption highlighted kinetochore metaphase signaling, cell cycle control of chromosomal replication, as well as cyclins and cell cycle regulation as the top functional pathways (Supplementary Fig. 20a). To assess whether TFAP2C is a direct regulator of the differentially expressed genes identified upon silencing, we performed ChIP-Seq in the stem state. While we noted an overall overrepresentation (Chi[2] = 90; Fisher $p < 0.0001$) of TF peaks close to (within 2 kb) downregulated genes (log2fold <−1; adjusted $p < 0.05$) (Fig. 6h) the same pattern was not seen for genes upregulated genes (log2fold > 1; adjusted $p < 0.05$,

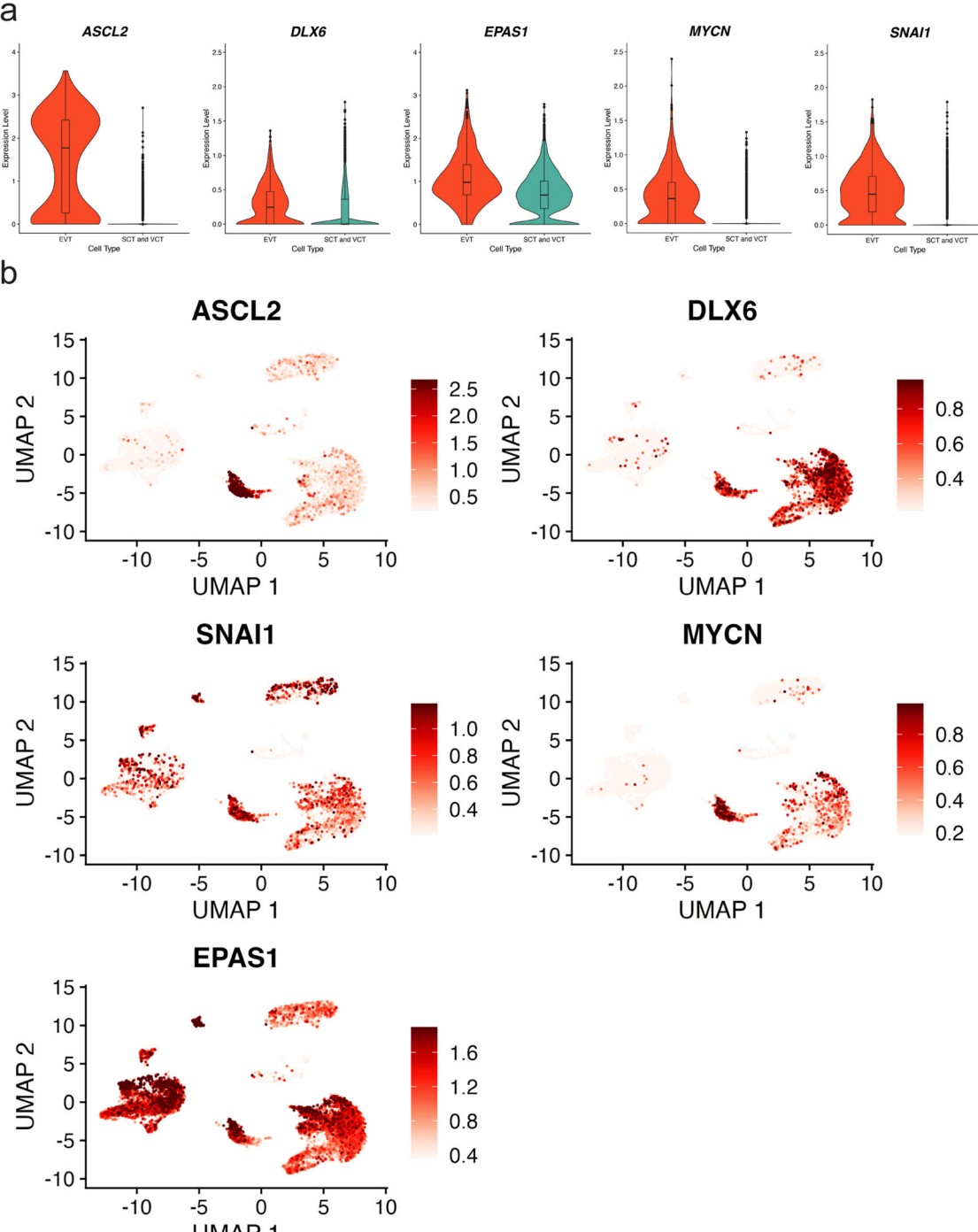

**Fig. 5 | Expression of transcription factors in single EVT cells derived from human placenta.** Data from two publicly available, independent, single-cell RNA sequencing (scRNA-Seq) datasets from first trimester placentas[27,28] are used and cell types are inferred using marker genes published previously[28] as decidual macrophages (dM), fetal (Endo f) and maternal endothelial cells (Endo m), epithelial glandular cells (Epi), extravillous trophoblast cells (EVT), fetal fibroblasts (FB), Hofbauer cells (HB), syncytiotrophoblast and villous cytotrophoblast (SCT/VCT) and cells not determined (ND). Single cell expression values (normalized, scaled and natural-log transformed) for five transcription factors (*ASCL2*; *DLX6*; *EPAS1*, *MYCN* and *SNAI1*) shown in (**a**) Violin plots (shown as median and 25th and 75th percentiles; points are displayed as outliers if they are above or below 1.5 times the interquartile range) comparing EVT cells (orange, $N = 693$ cells) with syncytiotrophoblast (SCT) and villous cytotrophoblast cells (VCT) combined (turquoise, $N = 7834$ cells) and (**b**) UMAP feature plots (red color scale of expression values thresholded at the 5th and 95th percentiles).

Supplementary Fig. 20b). Overall, we conclude that TFAP2C contributes to cell cycle regulation mainly as a transcriptional activator and is critical for maintenance of TS cells in a stem state and early EVT cell lineage development.

To evaluate the functions and potential contributions of SNAI1 and EPAS1 to EVT cell development, we again localized each target in placental specimens by in situ hybridization and performed loss-of-function experiments using the TS cell model in CT27 cells (Figs. 6–8) and CT29 cells (Supplementary Figs. 22–23). First, we examined the spatial distribution of each TF in first trimester human placental tissue and showed that both TFs were more abundant in distal regions of the EVT column and co-localized with *PLAC8*, a marker of differentiated

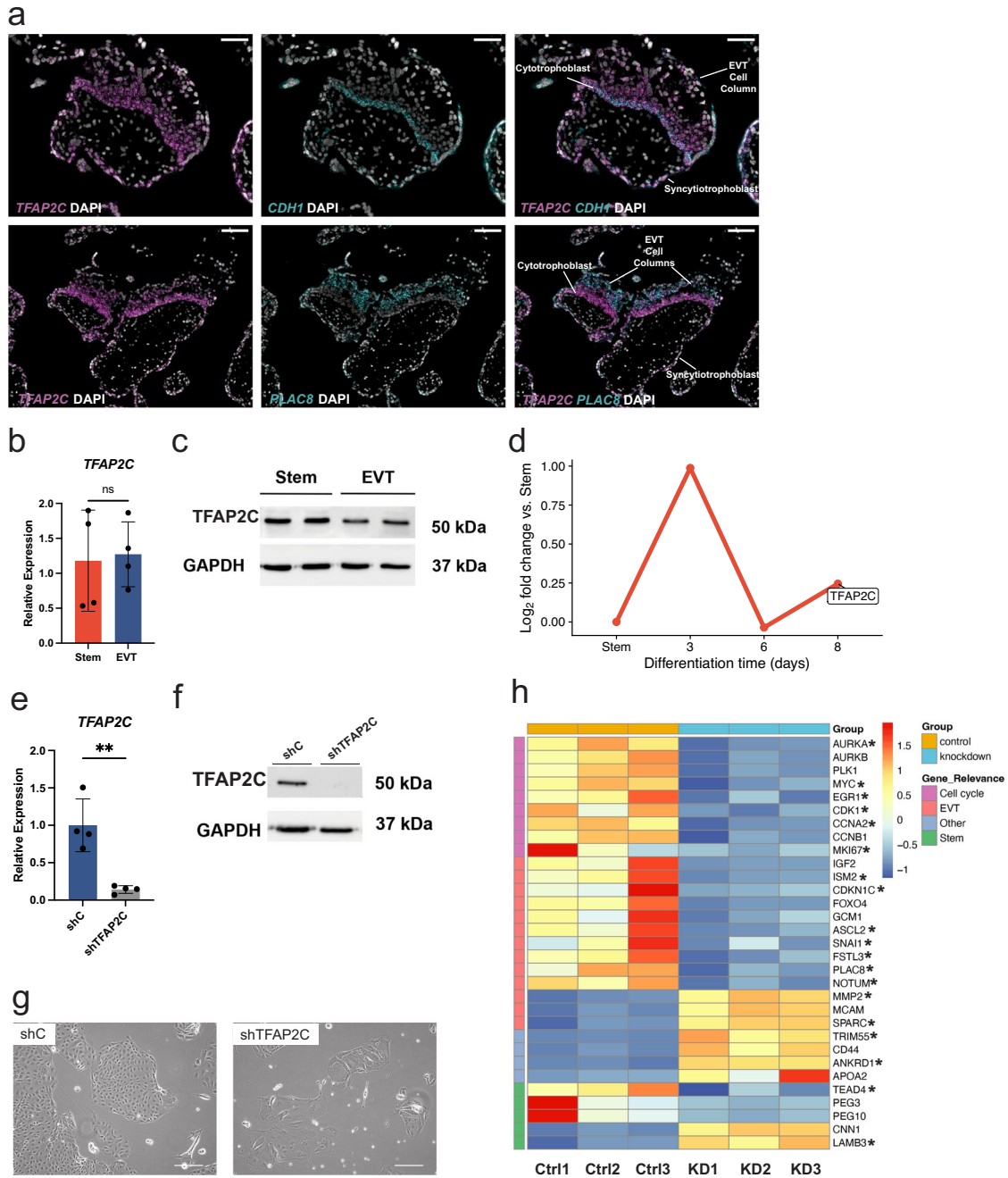

**Fig. 6 | Functional assessment of *TFAP2C* actions in trophoblast cell development. a** Representative human placental tissue specimen (12 weeks of gestation) probed for *TFAP2C*, *CDH1*, and *PLAC8* transcripts using in situ hybridization. 4′,6-diamidino-2-phenylindole (DAPI) labels cell nuclei. Overlay of merged immunofluorescence images: *TFAP2C* (magenta), DAPI (gray), and either *CDH1* (cyan) or *PLAC8* (cyan), respectively. Scale bars represent 50 μm (*CDH1* image panels) and 100 μm (*PLAC8* image panels). **b** RT-qPCR measurement of *TFAP2C* normalized to *POLR2A* in stem (red) and extravillous trophoblast (EVT) cells (blue). Data were analyzed by unpaired *t*-test and are presented as mean values ± standard deviation (SD; ns = not significant $p = 0.8374$; $n = 4$ biologically independent replicates per group). **c** TFAP2C (50 kilodaltons (kDa)) and GAPDH (37 kDa) proteins in stem and EVT cells assessed by western blot analysis. **d** $\text{Log}_2$ fold-change values of normalized read counts of *TFAP2C* from RNA-Seq in stem state and days 3, 6, and 8 of EVT cell differentiation compared to the stem state. **e** RT-qPCR measurement of *TFAP2C* normalized to *POLR2A* in stem state cells transduced with lentivirus containing a

control shRNA (shC; blue) or a *TFAP2C*-specific shRNA (gray). Data were analyzed by unpaired *t*-test and are presented as mean values ± standard deviation (SD; shTFAP2C; $n = 4$ biologically independent replicates per group; **$p = 0.0029$). **f** TFAP2C (50 kDa) and GAPDH (37 kDa) proteins in stem state cells transduced with shC or shTFAP2C and assessed by western blot. **g** Phase contrast images of stem state cells transduced with shC or shTFAP2C. Scale bar represents 250 μm. **h** Heat map based on scaled, normalized counts of selected differentially expressed transcripts generated from RNA-Seq in shC (control, Ctrl; orange) or shTFAP2C (knockdown, KD; blue) transduced cells using a red-blue diverging scale. Transcripts are clustered into four groups including cell cycle (purple), EVT-specific (pink), stem state-specific (green), or other (blue). *Depict genes identified as direct targets of TFAP2C based on ChIP-Seq analysis in stem state cells. Graphs in panels (**b**) and (**e**) depict mean ± standard deviation. Source data are provided as a Source Data file.

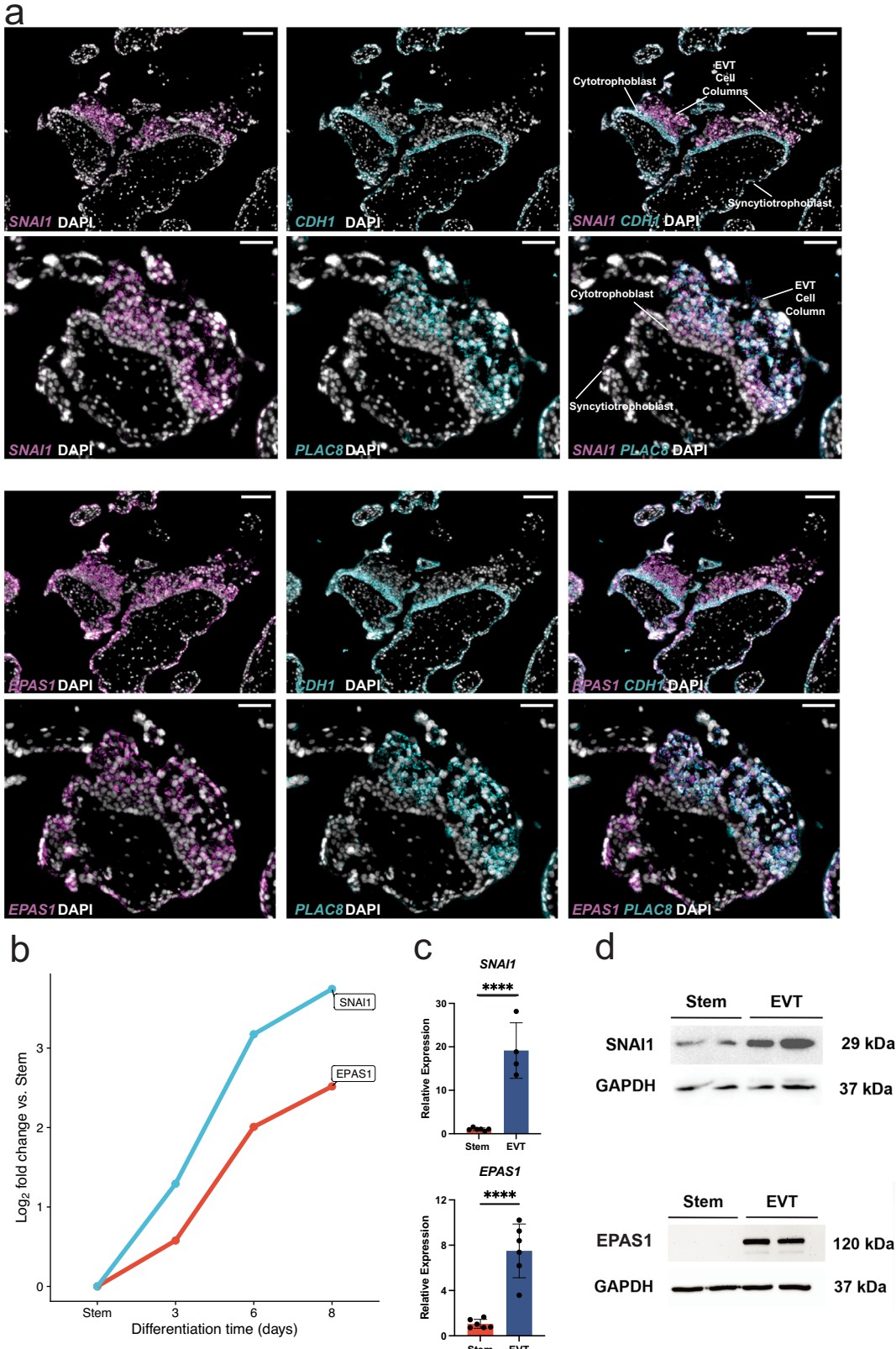

EVT cells[40,49] (Fig. 7a). *SNAI1* and *EPAS1* transcripts were significantly upregulated in EVT cells compared to stem state cells (Fig. 7b, c). This upregulation was also evident for SNAI1 and EPAS1 (also called HIF2A) at the protein level (Fig. 7d). Taken together, both TFs are expressed in EVT cells in situ and following their in vitro cell differentiation from TS cells.

To assess impacts on EVT cell development, we stably expressed shRNAs specific to each TF in the stem state and then attempted to differentiate the cells into EVT cells using CT27 (Figs. 8–9) and CT29 cell lines (Supplementary Figs. 22–23). shRNAs were identified that significantly inhibited *SNAI1* transcript (Fig. 8a, Supplementary Fig. 22A) and SNAI1 protein levels (Fig. 8b, Supplementary Fig. 22b).

**Fig. 7 | Expression patterns of *SNAI1*, and *EPAS1*. a** Human placental tissue specimen (12 weeks of gestation) probed for *SNAI1*, *EPAS1*, *CDH1* and *PLAC8* transcripts using in situ hybridization. 4′,6-diamidino-2-phenylindole (DAPI) labels cell nuclei. Overlay of merged immunofluorescence images: *SNAI1* (magenta), or *EPAS1* (magenta), with DAPI (gray), and *CDH1* (cyan) or *PLAC8* (cyan). Scale bars represent 100 μm in *CDH1* image panels and 50 μm in *PLAC8* image panels. **b** Log$_2$ fold-change values of normalized read counts of *SNAI1* (blue) and *EPAS1* (red) from RNA-Seq in stem state and days 3, 6, and 8 of extravillous trophoblast (EVT) cell differentiation compared to the stem state. **c** RT-qPCR measure of *SNAI1* (stem $n = 6$ (red) and EVT $n = 4$ (blue) biologically independent replicates per group), and *EPAS1* ($n = 6$ biologically independent replicates per group) normalized to *B2M* in stem and EVT cells (****$p < 0.0001$). Data were analyzed by unpaired *t*-test and are presented as mean values ± standard deviation (SD). **d** SNAI1 (29 kilodaltons (kDa)), EPAS1 (120 kDa), and GAPDH (37 kDa) protein in stem and EVT cells assessed by western blot analysis. Source data are provided as a Source Data file.

Similarly, *EPAS1*-specific shRNAs significantly inhibited expression at transcript (Fig. 9a, Supplementary Fig. 23a) and protein levels (Fig. 9b, Supplementary Fig. 23b) in CT27 and CT29 cell lines, respectively. Knockdown of either SNAI1, or EPAS1 were each compatible with maintenance of the TS cell stem state. However, deficits in each TF negatively affected aspects of EVT cell differentiation (Fig. 8c, Supplementary Fig. 22c), with EPAS1 disruption showing the most extensive morphologic impact (Fig. 9c, Supplementary Fig. 22c). EPAS1 knockdown cells consistently failed to exhibit the characteristic elongation normally associated with EVT cell differentiation.

Parameters of EVT cell differentiation in control and TF knockdown cells were further investigated using RNA-Seq which revealed that both TFs contributed to the regulation of several genes associated with EVT cell lineage development (Figs. 8–9). SNAI1 disruption led to downregulation of several genes associated with the EVT cell state (e.g., *FLT4*, *LAMA4*, *IGF2*, *ISM2*, *ASCL2*, *FSTL3*, *DLX6*, and *NOTUM*) and upregulation of EVT cell interferon-responsive transcripts (e.g., *IFIT1*, *IFIT2*, *IFIT3*, *IFITM1*, *IFI27*, *IFI35*, *IFIH1*, *OAS2*, *OAS3*, and *MX1*) as well as several stem state-associated transcripts (e.g., *ITGB6*, *GJA1*, *LAMB3*, *F3*, and *TAGLN*) (Fig. 8d). Similar transcriptomic changes were observed in CT29 cells following SNAI1 disruption (Supplementary Fig. 22d). IFN-responsive transcript expression appears to be a feature of human trophoblast cell differentiation[50] and may be associated with repetitive element expression and double-stranded RNA stress[51]. Consistent with morphological changes, disrupting EPAS1 had the most extensive impact on the EVT cell transcriptome. Signature EVT cell-specific transcripts were prominently downregulated (e.g., *NOTUM*, *DIO2*, *HTRA1*, *MMP2*, *PDE6H*, *ADAM12*, *MCAM*, *HLA-G*, *ASCL2*, *DLX6*, *DLX5*, *ISM2*, *CCR1*, *SNAI1*, *ITGA1*, and *GCM1*). GCM1 was recently reported to play an important role in EVT cell development[52]. In addition, several stem state markers (*ITGB6*, *LAMB3*, *F3*, *JUN*, *EGLN3*, *CYR61*, and *NPPB*) were upregulated in EPAS1 disrupted cells (Fig. 9d). Similar transcriptomic changes were observed in CT29 cells following EPAS1 disruption (Supplementary Fig. 23d). In summary, disruption of SNAI1 or EPAS1 each resulted in downregulation of EVT cell-specific transcripts.

Differences among the activities of each TF as a transcriptional repressor versus activator (calculated based on Log2FC > 3 or <−3) were evident. SNAI1 and EPAS1 had repression to activation ratios of 70:30 and 60:40, respectively. The known repressor actions of SNAI1[53] are consistent with the profile.

In addition to the common feature of the TFs impacting known EVT-cell markers, each TF also contributed to the regulation of unique transcript signatures. In fact, pathway analysis performed on the top 500 significantly altered (upregulated or downregulated) genes upon silencing of each TF revealed that the most significant biological function associated with the differentially regulated genes for both TFs was the function: Migration of Cells (SNAI1: $p = 1.5E−39$; EPAS1: $p = 8.3E−49$; Supplementary Data 12a, b, Supplementary Data 13). Additional pathway analysis performed on the subset of EPAS1 gene targets mapping to the Migration of Cell cluster ($n = 209$ genes) identified another top function: Invasion of Cells ($p = 5.3E−70$). Many genes mapping to the Invasion of Cells cluster are known contributors to the regulation of EVT cell invasion into the uterus (Supplementary Fig. 21b).

## EPAS1 is an upstream TF regulator and dysregulated in placenta failure

As noted in the functional assays, EPAS1 disruption had the greatest impact on EVT cell morphology and resulted in the downregulation of multiple key genes involved in EVT cell differentiation and invasion. To test the potential of EPAS1 to act as a direct transcriptional regulator, we used ATAC-Seq peaks containing EPAS1 (HIF2A) motifs in EVT cells and intersected with RNA-Seq data from control and EPAS1 knockdown cells (Supplementary Data 14). This analysis identified EPAS1-regulated genes, as detected by RNA-Seq analysis (log2 foldchange < −1; adjusted $p < 0.05$), being significantly correlated with chromatin accessible regions containing EPAS1 (HIF2A) binding motifs. This corresponded to a 1.5-fold enrichment (Chi$^2$ = 62; Fisher $p < 0.0001$) when considering all EVT cell peaks and a 2.5-fold enrichment when restricting to those EVT cell peaks with differential binding affinity (Chi$^2$ = 71; Fisher $p < 0.0001$). Among these putative direct targets of EPAS1 action were *PLAC8*, *DIO2*, *MMP2*, *MYCN*, *ADAM12*, *HLA-G*, and *DLX6*.

As failure of proper EVT cell invasion into the uterine compartment at the maternal fetal interface may lead to pregnancy complications, we next assessed whether there is evidence linking EPAS1 to adverse pregnancy outcomes such as pregnancy loss[54]. To this end, single-cell gene expression data were generated from first trimester placental samples (4787 cells) derived from individuals suffering from idiopathic RPL. Single EVT cells ($n = 194$ independent cells) were profiled, and expression profiles were compared to compatible single-cell expression profiles of EVT cells ($n = 693$ independent cells) from gestational age-matched normal control placental samples (Supplementary Fig. 24). Of the 2967 differentially expressed genes (absolute log2fold change >1, adjusted $p < 0.05$) identified in EVT cells isolated from RPL specimens (Supplementary Data 15), EPAS1 was ranked among the top 200. Further, *EPAS1* expression was significantly higher in placentas from RPL patients (log2fold change = 1.6, adjusted $p = 7.2E−9$, Fig. 9f).

To assess the potential role of EPAS1 in gene expression dysregulation in RPL, a comparison was performed between upregulated genes identified in EVT cells derived from RPL placentas (log2 fold change >1; adjusted $p < 0.05$) and genes downregulated by EPAS1 disruption in TS cell-derived EVT cells (log2 fold change <−1; adjusted $p < 0.05$). This comparison identified a 1.7-fold enrichment (Chi$^2$ = 53; Fisher $p < 0.0001$) indicative of common gene targets. Among these genes (i.e., upregulated in RPL and positively regulated by EPAS1 during EVT cell differentiation) were *NOTUM*, *DIO2*, *ADAM12*, *ITGA1*, *CSH1*, *PAPPA*, *PAPPA2*, *CDKN1C*, *LIFR*, *FN1*, and *FLT1*. Upregulation of *FLT1* has been shown to be linked to EPAS1 upregulation during hypoxia-induced placental failure[55] and upregulation of *CDKN1C* has been observed in placental samples derived from pregnancy loss cases[54]. In all, these RPL-associated changes in EPAS1-driven EVT cell gene expression may reflect an adaptive response of a failing pregnancy. A healthy placenta can be defined by its ability to effectively adapt to stressors within the uterine environment[56]. Pregnancy failure

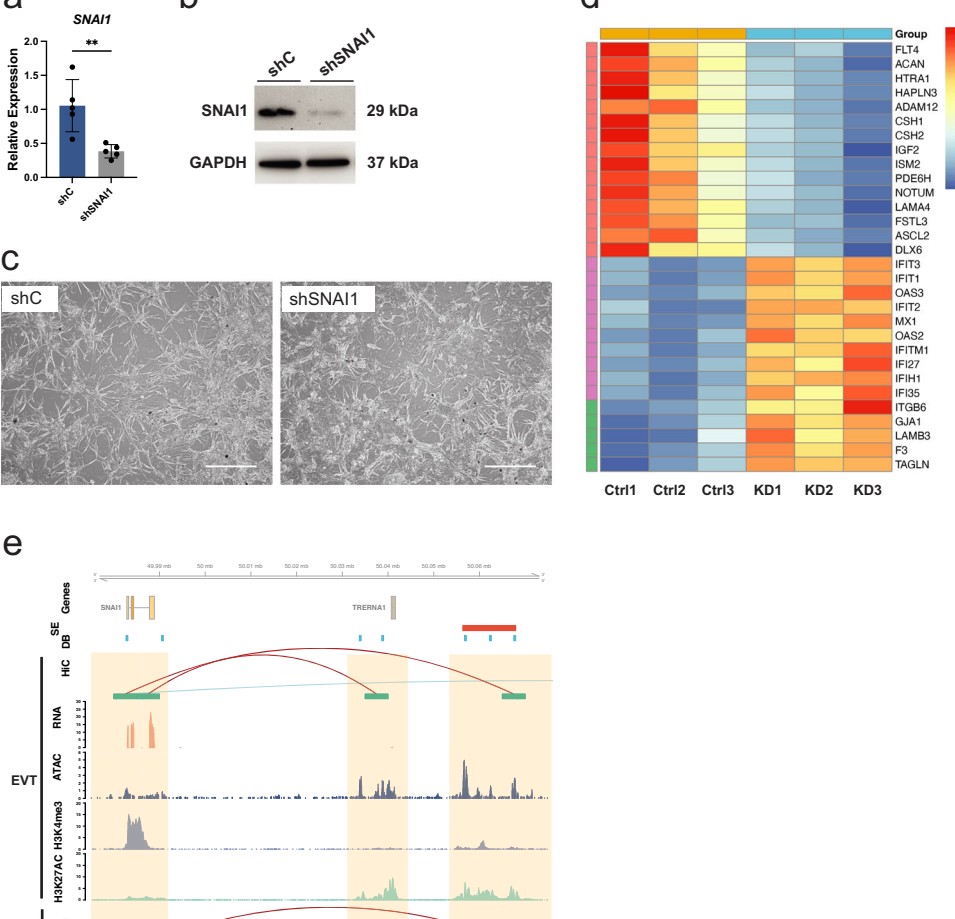

**Fig. 8 | Functional investigation of *SNAI1* on EVT cell differentiation in CT27 cells. a** RT-qPCR measurement of *SNAI1* (*n* = 5 biologically independent replicates per group) normalized to *B2M* in extravillous trophoblast (EVT) cells differentiated from stem state cells transduced with lentivirus containing a control shRNA (shC; blue) or SNAI1-specific shRNA (shSNAI1; gray). Data were analyzed by unpaired t-test and are presented as mean values ± standard deviation (SD) and **\*\*p* = 0.0054. **b** SNAI1 (29 kilodaltons (kDa)) and GAPDH (37 kDa) protein in EVT cells from shC control or shSNAI1 treated cells measured by western blot. **c** Phase contrast images of shC control or shSNAI1 EVT cells following eight days of differentiation. Scale bars represent 500 μm. **d** Heat map based on scaled, normalized counts of selected transcripts generated from RNA-Seq in shC (control, Ctrl; orange) or shSNAI1 (knockdown, KD; blue) transduced cells (*n* = 3 biologically independent replicates per group) using a red-blue diverging scale. Transcripts are clustered into three

groups including EVT-specific (pink), EVT interferon (IFN) responsive (purple), or stem state-specific (green). **e** Hi-C, RNA-Seq, ATAC-Seq, and H3K4me3 and H3K27ac ChIP-Seq assessments performed in CT27 cells differentiated into EVT cells (top panel) or maintained in the stem state (bottom panel). Identified regulatory elements near *SNAI1* are highlighted in orange and overlap Hi-C loops (red, both loop anchors in view; blue, loop anchors out of view), open chromatin by ATAC-Seq (counts per million mapped reads, *y*-axis), and promoter or enhancer mark by H3K4me3, or H3K27ac ChIP-Seq, respectively (counts per million mapped reads, *y*-axis). Expression level is shown by RNA-Seq (transcripts per million, *y*-axis). All datasets are shown in individual tracks. Super-enhancers (SE, red) and differentially bound regions (DB, blue) are specific to the EVT cell state. Source data are provided as a Source Data file.

in RPL may be exacerbated because EPAS1-driven adaptations are insufficient to sustain the pregnancy.

### Genetic variants at the EPAS1 locus are linked to pregnancy complications

To test the hypothesis of a potential genetic predisposition for pregnancy loss, we accessed summary statistics from genome-wide association studies (GWAS) of related phenotypes available in the UK

Biobank. In addition to pregnancy loss phenotypes, we also included analysis of maternal and fetal genetic effects on birth weight (Supplementary Table 1) as failure of the placenta can result in fetal growth restriction as well. Using these phenotypes, we computed the odds ratio of the proportion of nominally significant ($p < 0.05$) hits in the *EPAS1* gene region (chr2:46,150,000–46,450,000) encompassing our regulatory elements compared with variants in a randomly selected matched window. We found a striking enrichment of *EPAS1*

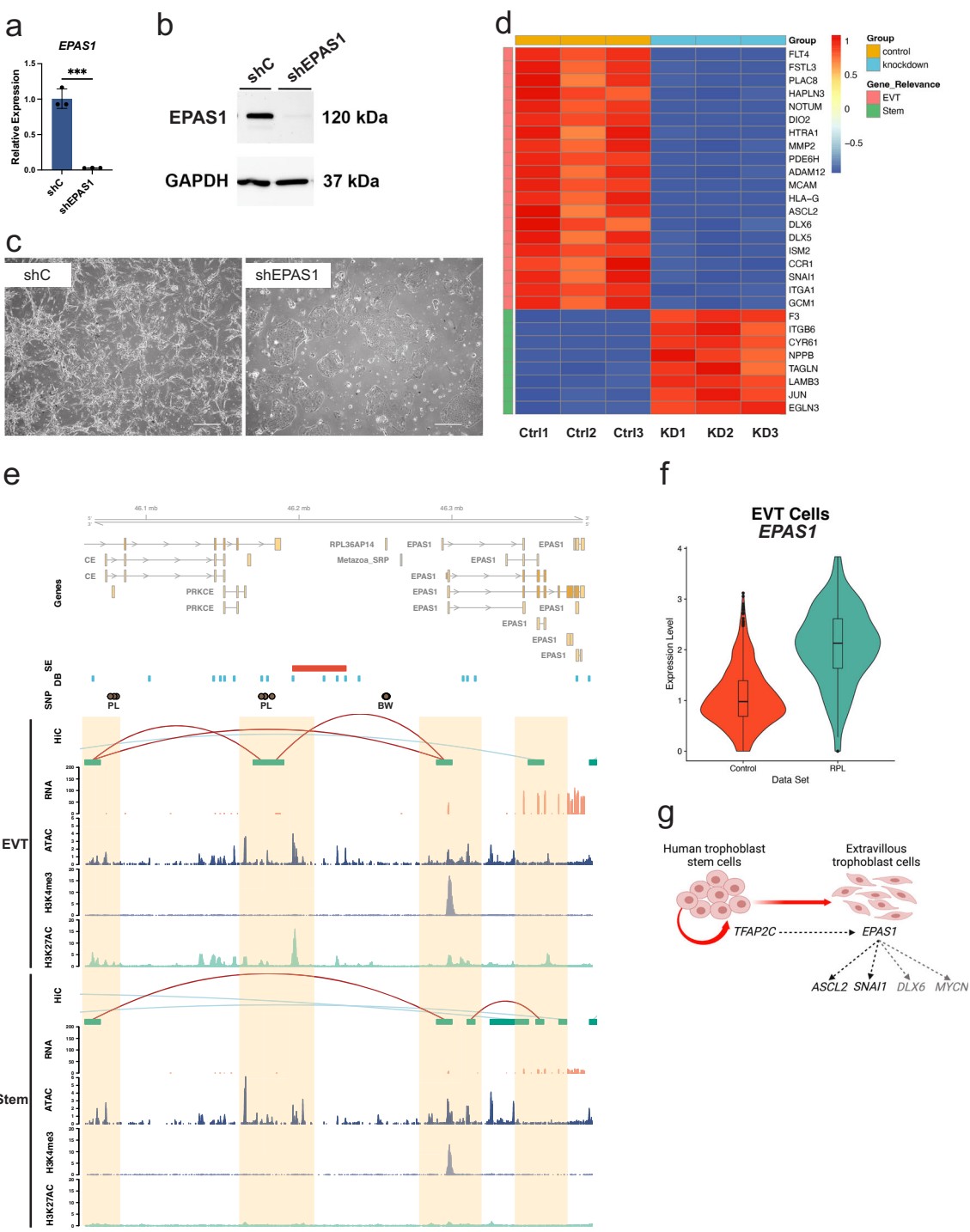

genetic variants for not only pregnancy loss but also for birth weight compared to background (Fig. 9e; Supplementary Fig. 21a; Supplementary Fig. 25; Supplementary Data 16). While there is to our knowledge no other large-scale GWAS done on pregnancy loss that can be used for additional replication, we accessed summary statistics from an independent, similar sized GWAS of birth weight[57] ($N = 321,223$) and were able to confirm the effect of the *EPAS1* locus (rs17034876; $p_{UKBB} = 7.1E{-}28$ and $p_{Warrington} = 3.1E{-}47$). Finally, we used all loci associated with birth weight ($N = 273$ genes, $p < 0.001$) and found a significant link (1.9-fold change, $Chi^2 = 32.7$; Fisher $p < 0.0001$) between fetal genetic loci for birth weight and genes dysregulated by EPAS1 disruption (absolute log2fold change >1, adjusted $p < 0.05$). Genes included not only *EPAS1* itself (Fig. 9e and

Supplementary Fig. 21a), but also *FLT1*, PAPPA, PAPPA2, and *CDKN1C* (Supplementary Data 17).

To conclude, these predisposing *EPAS1* genetic variants associated with pregnancy complications may lead to EPAS1 dysregulation, abnormalities in transcriptional control, and ineffective placentation leading to increased risk of placental disorders. However, further mechanistic, and population-based studies are needed to establish these relationships.

## Discussion

EVT cell invasion into the uterine compartment is a critical component of spiral artery remodeling and successful pregnancy[2,58]. Despite our understanding of the importance of EVT cell function, our knowledge

**Fig. 9 | Functional investigation of *EPAS1* on EVT cell differentiation in CT27 cells. a** *EPAS1* (*n* = 3 biologically independent replicates per group) normalized to *B2M* in extravillous trophoblast (EVT) cells differentiated from stem cells transduced with control shRNA (shC; blue) or EPAS1-specific shRNA (shEPAS1; gray) lentivirus. Data were analyzed by unpaired *t*-test and are presented as mean values ± standard deviation (SD) and ****p* = 0.0002. **b** EPAS1 (120 kilodaltons (kDa)) and GAPDH (37 kDa) in EVT cells from shC control or shEPAS1 transduced cells. **c** Phase contrast images of EVT cells (day 8). Scale bars represent 500 μm. **d** Heat map based on scaled, normalized counts of selected transcripts (RNA-Seq) in shC control (Ctrl, orange) or shEPAS1 (KD, blue) transduced cells (*n* = 3 biologically independent replicates per group) using a red-blue diverging scale. Transcripts cluster into two groups (i) EVT-specific (pink) or (ii) stem state-specific (green). **e** Hi-C, RNA-Seq, assay for transposase-accessible chromatin (ATAC)-Seq, and H3K4me3 and H3K27ac chromatin immunoprecipitation (ChIP)-Seq assessments performed in EVT differentiated (top) or stem state (bottom) CT27 cells. Regulatory elements near *EPAS1* (orange) overlap Hi-C loops (red, both loop anchors in view; blue, loop anchors out of view), open chromatin identified by ATAC-Seq (counts per million (CPM) mapped reads, *y*-axis), and promoter or enhancer marks identified by H3K4me3, or H3K27ac ChIP-Seq, respectively (CPM mapped reads, *y*-axis). Expression level is shown by RNA-Seq (transcripts per million, *y*-axis). All datasets shown as individual tracks. Super-enhancers (SE, red) and differentially bound regions (DB, blue) are EVT cell-specific. Brown circles denote SNPs associated with pregnancy loss (PL) or birth weight (BW) in large GWAS studies. **f** Violin plot (shown as median and 25th and 75th percentiles; points are displayed as outliers if they are above or below 1.5 times the interquartile range) of *EPAS1* in EVT cells from single-cell RNA-Seq analysis of human first trimester placental samples from control (Control, orange, *N* = 693 cells) or recurrent pregnancy loss (RPL, turquoise, *N* = 194 cells) patients. *Y*-axis represents normalized and natural log transformed *EPAS1* expression levels. **g** Schematic of transcriptional regulatory hierarchy controlling EVT cell differentiation (BioRender). Source data are provided as a Source Data file.

---

of the underlying molecular mechanisms driving EVT cell development is limited. Here, we utilized the human TS model[22] to track regulatory events controlling EVT cell differentiation and generated an NGS-based functional genomics resource comprising atlases for transcriptomes by RNA-Seq, open and active chromatin by ATAC-Seq and ChIP-Seq of histone modifications, and chromatin interactions by Hi-C. Although, all in vitro models have limitations, human TS cells represent a remarkable high fidelity stem cell system with the capacity to exhibit many traits intrinsic to EVT cells developing in situ. Later stages of differentiation associated with EVT cells penetrating the uterine parenchyma are unlikely to be captured as well in this cell culture model and would require additional refinements.

The NGS datasets from the TS model were used to identify higher-order regulatory events driving EVT cell lineage specification including characterization of signature epigenomic landmarks characteristic of EVT cell lineage development. Global epigenomic events were reinforced by inspection at the level of individual genes known to drive EVT cell differentiation (e.g., *ASCL2*)[20] and downstream genes responsible for the phenotypic features of an EVT cell (e.g., *MMP2*, and *HLA-G*).

The architecture of the epigenome undergoes profound changes as stem cells differentiate into specialized cells[59,60]. The transformation is apparent at multiple levels. Some chromatin regions become more accessible and other regions are closed to the requisite machinery controlling gene transcription. This regulatory process is facilitated by formation of chromatin loops allowing for long-range interactions of regulatory elements including enhancers and promoters. The net effect is distinct transcript profiles associated with stem versus differentiated cells, resulting in cells possessing different morphologies and functional attributes. These architectural and functional changes are exemplified in human TS cells differentiating into EVT cells. TS cells cultured in the stem state are tightly associated, and cellular efforts are directed to proliferation, whereas differentiation into EVT cells is characterized by cell colony dispersal, the manifestation of prominent cellular extensions, and cell migration. Trophoblast cell differentiation status is further characterized by cell-specific patterns of gene expression, chromatin accessibility and long-range chromatin interactions—all occurring in a coordinated manner. This map of trophoblast-lineage specific regulatory DNA generated here consists not only of distinct active chromatin regions specific to EVT cells but also extends current maps of open chromatin for human cell types and tissues[29]. Increased chromatin accessibility in EVT cells was enriched in intronic and intergenic regions and linked to increased RNA expression. These observations are in line with reference epigenome mapping efforts[29] and point towards pronounced activity of distal regulatory regions (i.e., enhancers) controlling cell lineage development through involvement of master TFs regulating gene expression in EVT cells. Indeed, the appearance of increased enhancer-driven gene regulation in EVT

versus stem state cells were confirmed by ChIP-Seq of histone modifications marking active enhancer regions (H3K27ac).

Motif enrichment within chromatin accessible regions revealed potential 'core' TF-DNA interactions associated with the EVT cell developmental state including TEAD3 and TFAP2C. TEAD3 was identified in a large integrative analysis of over 100 cell types to bind to enhancers that were specific to embryonic stem cell differentiated into trophoblast by treatment with bone morphogenetic protein 4[29]. TFAP2C is a known regulator of the mouse TS cell stem state[6,9]. Consistent with these observations we showed that TFAP2C is also critical for human TS cell expansion and the onset of EVT cell differentiation. In addition, TFAP2C represents a key factor in the derivation of TS cells from fibroblasts and pluripotent stem cells[15,16,47,61]. A recent CRISPR-Cas9 screen has identified other transcriptional regulators critical to maintaining the TS cell stem state (ARID3A, GATA2, TEAD1, and GCM1)[44].

Our genome-wide integrative analyses with multiple layers of epigenome information including active regulatory regions and their interactions led to the identification of an established TF regulator of EVT cell development, ASCL2[20]. The epigenomic landscape of *ASCL2* changed as trophoblast cells differentiated from the stem state to EVT cells. This transition was characterized by the appearance of a long-range interaction of a super-enhancer with the *ASCL2* promoter. We also identified compelling TF candidates controlling EVT cell lineage development including SNAI1, DLX6, MYCN, and EPAS1—each possessing epigenomic signatures indicative of gene activation in both cultured EVT cells and primary EVT cells. Silencing of SNAI1 resulted in disruptions in EVT cell development and affected the expression of a subset of transcripts characteristic of EVT cells. SNAI1 is an established transcriptional repressor with known roles in promoting epithelial to mesenchymal transition[53], including a potential involvement in EVT cell development[62]. *EPAS1* encodes hypoxia inducible factor 2 alpha (HIF2A), a basic helix-loop-helix domain TF, coordinating cellular responses to low oxygen and developmental processes with a connection to trophoblast cell biology[63]. The actions of EPAS1 on EVT cell development were much more extensive than observed for SNAI1 and included transcriptional regulation of virtually all transcripts defining the EVT cell phenotype. Interestingly, these transcriptional EPAS1 targets also included TFs contributing to the regulation of EVT cell differentiation (e.g., ASCL2 and SNAI1). Thus, EPAS1 can be placed upstream relative to other transcriptional regulators directing EVT cell differentiation (Fig. 9G). *EPAS1* mRNA and EPAS1 protein accumulate as EVT cells differentiate and do so independent of low oxygen tensions. Mechanisms controlling EVT cell dependent *EPAS1* transcriptional activation and EPAS1 protein stabilization are unknown. Elucidation of factors controlling EVT cell EPAS1 will add to understanding of the regulation of EVT cell differentiation.

The biology of EPAS1 in placentation is complicated. EPAS1 expression is not restricted to EVT cells. Syncytiotrophoblast also

express EPAS1[64,65]. Thus, additional layers of regulation impart specificity for EPAS1 actions in EVT cells versus syncytiotrophoblast. These disparate actions may be linked to contributions of cell specific co-regulators[66,67], canonical versus non-canonical EPAS1 signaling[68], or the involvement of other unappreciated modulator(s) of EPAS1 action. The relevance of EPAS1 to placentation is evident in its association with placental disease. RPL is characterized by elevated EVT cell *EPAS1* expression. EPAS1 is also dysregulated in preeclampsia[69]. Under such circumstances, EPAS1 may be responding to a failed placenta and the ensuing hypoxia. Context is important, and inappropriate EPAS1 may hinder rather than support placentation. Compromised placental function negatively impacts fetal growth, which is also connected to misexpression of EPAS1 gene targets and is demonstrated by genetic linkage to low birth weight[57].

In summary, we have provided insights into a trophoblast regulatory network controlling EVT cell lineage development. Datasets generated from this research represent a resource for developing new hypotheses that can be tested in vitro with human TS cells and in vivo using appropriate animal models. They will also provide a valuable reference dataset for newly established TS cell lines derived from pluripotent stem cells[47,70,71]. TFAP2C, SNAI1, and EPAS1 provide entry points and a framework for understanding EVT cell differentiation and human placentation. The challenge will be to harness this knowledge to develop effective strategies for diagnosing and treating diseases of placentation.

## Methods

### Ethical declarations
This study using deidentified TS cells complies with all relevant ethical regulations as approved by the Human Research Protection Program and the Human Stem Cell Research Oversite Committee at the University of Kansas Medical Center who determined the study as non-human subjects' research. Prior approval for use of human placental tissue specimens was granted by the respective local human research ethics review committees at the Mount Sinai Hospital and the University of Kansas Medical Center. The RPL study was approved by the Children's Mercy Institutional Review Board (Study No. 11120514). Informed written consent was obtained from the RPL participants before study inclusion. Participants were not compensated for study participation.

### Human TS cell culture
Human TS cells (CT27, 46, XX and CT29, 46, XY)[22] were cultured in 100 mm tissue culture dishes coated with 5 μg/mL collagen IV (CB40233, Thermo-Fisher, Waltham, MA). Human TS cells were maintained in Complete Human TS Cell Medium [DMEM/F12 (11320033, Thermo-Fisher), 100 μm 2-mercaptoethanol, 0.2% (vol/vol) fetal bovine serum (FBS), 50 U/mL penicillin, 50 μg/mL streptomycin, 0.3% bovine serum albumin (BSA, BP9704100, Thermo-Fisher), 1% Insulin-Transferrin-Selenium-Ethanolamine solution (ITS-X, vol/vol, Thermo-Fisher)], 1.5 μg/mL L-ascorbic acid (A8960, Sigma-Aldrich, St. Louis, MO), 50 ng/mL epidermal growth factor (EGF, E9644, Sigma-Aldrich), 2 μM CHIR99021 (04-0004, Reprocell, Beltsville, MD), 0.5 μM A83-01 (04-0014, Reprocell), 1 μM SB431542 (04-0010, Reprocell), 0.8 mM valproic acid (P4543, Sigma-Aldrich), and 5 μM Y27632 (04-0012-02, Reprocell). EVT cell differentiation was induced by plating human TS cells onto 6-well plates pre-coated with 1 μg/mL collagen IV at a density of 80,000 cells per well. Cells were cultured in EVT Differentiation Medium [DMEM/F12 (11320033, Thermo-Fisher), 100 μM 2-mercaptoethanol,, 50 U/mL penicillin, 50 μg/mL streptomycin, 0.3% BSA, 1% ITS-X solution (vol/vol)], 100 ng/mL of neuregulin 1 (NRG1, 5218SC, Cell Signaling, Danvers, MA), 7.5 μM A83-01 (04-0014, Reprocell, Beltsville, MD), 2.5 μM Y27632, 4% KnockOut Serum Replacement (KSR, 10828028, Thermo-Fisher), and 2% Matrigel® (CB-40234, Thermo-Fisher). On day 3 of EVT cell differentiation, the

medium was replaced with EVT Differentiation Medium excluding NRG1 and with a reduced Matrigel® concentration of 0.5%. On culture day 6 of EVT cell differentiation, the medium was replaced with EVT Differentiation Medium excluding NRG1 and KSR, and with a Matrigel concentration of 0.5%. Cells were analyzed in the stem state and at days 3, 6, and 8 of EVT cell differentiation.

Cells were harvested from plates with trypsin, counted, and viability determined with a Countess II automated cell counter (Thermo-Fisher). Sample replicates with the highest viability and an adequate cell count were selected to be used for ATAC-Seq. Cells used for RNA isolation were centrifuged at $300 \times g$ for 8 min, supernatants were removed, and cell pellets were resuspended in 1 mL of TRIzol (15596026, Thermo-Fisher).

### Flow cytometric analysis
Cells in culture were washed with phosphate-buffered saline (PBS), detached with TrypLE Express (12604021, Thermo-Fisher Scientific), and collected in basal culture medium. Cell suspensions were centrifuged, cell pellets were washed with PBS, and resuspended with 4% paraformaldehyde in PBS for 20 min at room temperature with gentle agitation. Fixed cell suspensions were centrifuged, and cell pellets were washed twice with PBS and stored at 4 °C. Fixed cells were permeabilized with PBS containing 3% BSA and 0.2% Triton X-100 for 30 min at room temperature with gentle agitation. Cells were washed with PBS and blocked with PBS containing 3% BSA for 15 min and then incubated overnight with anti-HLA-G-phycoerythin (1:1500, 1P-292-C100, Cedarlane Labs) prepared in PBS containing 3% BSA and 0.2% Triton X-100 in the dark at 4 °C with gentle agitation. Cells were washed twice with PBS, filtered, and analyzed using a BD LSR II flow cytometer (BD Biosciences) at the University of Kansas Medical Center Flow Cytometry Core Laboratory. Data was analyzed with FlowJo v10.9.

### Omni-ATAC library preparation, sequencing, and analysis
The Omni-ATAC library preparation was performed on freshly isolated cells according to an established protocol[72] until the post-transposition cleanup step. After cleanup of the transposed DNA was complete, samples were stored at −20 °C until library amplification. Samples were subsequently thawed at room temperature and library construction completed. The samples underwent 5 cycles of PCR before an aliquot was used in a quantitative PCR (qPCR) reaction to determine whether additional amplification was required. An Applied Biosciences Viia7 Real-Time PCR System (Applied Biosciences, Beverly Hills, CA) was used for qPCR and the number of additional PCR cycles needed was determined from the multicomponent plot, which plots fluorescence versus the number of PCR cycles. The maximum fluorescence for each sample was found from the plot, ¼ of the maximum fluorescence value was calculated, and the number of additional PCR cycles needed was calculated by determining the cycle number that reached ¼ maximum fluorescence on the graph. In cases where the ¼ maximum fluorescence value was halfway between two cycle numbers, the lower number of cycles was selected since slight under-amplification was more favorable than overamplification. The original samples were then PCR amplified with their determined number of additional cycles. Cleanup of the samples was performed after PCR amplification according to the published protocol[72]. Libraries were quantified using a Qubit dsDNA BR Assay Kit (Q32853, Thermo-Fisher) and the size was determined with a High Sensitivity DNA Bioanalyzer Kit (5067-4626, Agilent, Santa Clara, CA) and sequenced on a Nova-Seq 6000 (Illumina, San Diego, CA)using Nextera Sequencing primers.

Basic quality control of raw FASTQ files was performed using HTStream (Version 1.3.2; https://github.com/s4hts/HTStream). PhiX reads were removed using hts_SeqScreener with default parameters. Duplicates were removed using hts_SuperDeduper (-e 250000). Adapter sequences were trimmed with hts_AdapterTrimmer (-p 4). Unknown nucleotides (N) were removed using hts_Ntrimmer. Base

quality trimming was performed using a minimum average quality score of 20 (-q 20) in a 10-bp sliding window (-w10) with hts_QWindowTrim. Trimmed reads shorter than 50 bp as well as orphaned reads from a pair were removed using hts_LengthFilter (-n -m 50). Reads were aligned to the human genome (GRCh38.86) using BWA mem (Version 0.7.17-r1188). The alignments were shifted using alignmentSieve (deepTools; Version 3.5.1) using the --ATACshift option. Alignments in the ENCODE blacklisted regions (https://doi.org/10.5281/zenodo.1491733) were removed using Bedtools intersect (with option '-v'; Version 2.30.0). Peaks for each sample were called independently using the callpeak function in MACS3 (Version 3.0.0a6) with a minimum FDR of 0.01 (-q 0.01). Peak regions were annotated in regard to the genomic feature of their location using HOMER's annotatePeaks.pl script as well as overlap with DNAse I hypersensitive sites (min overlap of 1 bp)[29]. Unique peaks (e.g., EVT cell) were determined based on complete lack of overlap with a comparing set of peak regions (e.g., stem state).

The webtool GREAT was used to retrieve genes with an overlapping regulatory domain, where the regulatory domain of a gene is the region 5 kb upstream and 1 kb downstream of the transcription start site (TSS). In addition, distal gene elements were retrieved where the regulatory domain of the genes extends up to 1 Mb. Finally, each peak was annotated by mapping it to the most significant differentially expressed gene (EVT vs stem) with a TSS within 10 kb of the peak region.

ATAC-seq read counts per peak were generated using the Bioconductor package DiffBind (Version 3.2.7). Peaks with <10 counts per million (CPM) were excluded. The differential binding affinity of a region between EVT and stem cell samples was assessed in the Bioconductor package limma (Version 3.48.3) with an FDR of 0.05.

Super-enhancers were called using HOMER (version 4.7.2) findpeaks (http://homer.ucsd.edu/homer/ngs/peaks.html) using the -style super mode.

### RNA library preparation, sequencing, and analysis

RNA was isolated with a TRIzol/chloroform precipitation followed by a cleanup with a RNeasy Mini Kit (74104, Qiagen, Germantown, MD). Frozen cell isolates in TRIzol were thawed on ice prior to starting the isolation. Once thawed, the samples were placed at room temperature for 5 min to promote dissociation of nucleoprotein complexes. Next, 200 µL of chloroform was added to each sample with vigorous shaking for 15 s. Samples were placed at room temperature for 3 min before centrifugation at $12,000 \times g$ for 15 min at 4 °C. After centrifugation, the upper, colorless, aqueous phase was removed and transferred to a new 1.5 mL tube and 1.5 volumes of 100% ethanol added. Isolates containing ethanol were mixed thoroughly by pipetting and 650 µL transferred to a RNeasy spin column. The protocol for the RNeasy Mini Kit was then followed to clean the sample post-TRIzol/chloroform extraction. RNA was then quantified using a Qubit RNA BR Assay Kit and RNA integrity assessed using RNA ScreenTape (5067−5577 and 5067−5576, Agilent) on the Agilent TapeStation platform (Agilent) prior to library preparation.

RNA libraries were prepared using a TruSeq Stranded RNA HT Sample Prep Kit (RS-122-2303, Illumina on a Caliper Sciclone G3 platform (PerkinElmer, Waltham, MA). Manual bead cleanup was performed using AMPure XP Beads (A63881, Beckman Coulter, Brea, CA) after library preparation to remove primer-dimers. 1.0x volume of beads was added to the entire library volume. Samples were mixed by pipetting and incubated at room temperature for 5 min before being placed on a magnetic stand until the supernatant cleared. Once clear, the supernatant was removed and discarded, and the beads were washed twice with 80% ethanol while still on the magnet. Any residual ethanol was removed, and the beads were air-dried on the magnet for 5 min at room temperature. Once dry, 30 µL of Resuspension Buffer was added to each sample, the beads were resuspended by pipetting,

and the solution incubated for 2 min at room temperature. Samples were then placed on a magnet until the supernatant was clear. The supernatant was then transferred to a new plate. A Qubit dsDNA BR Assay kit was used to determine the concentration of the library and a Fragment Analyzer Standard Sensitivity NGS Fragment Kit (DNF-473-10000, Agilent) was used to detect the size of the library and to verify removal of excess primer-dimers. Standard Illumina Free-Adapter Blocking was performed on the libraries. Cleaned, adapter-blocked libraries were loaded on a NovaSeq 6000 with a run configuration of $151 \times 8 \times 8 \times 151$ and an average depth of 70 M paired-end reads per library.

Raw FASTQ files were trimmed using default parameters (-r 0.1 -d 0.03) in Skewer (Version 0.2.2) and reads shorter than 18 bp were discarded. Transcripts were quantified using Kallisto (Version 0.46.2). Differentially expressed genes at FDR of 0.05 were discovered using the Bioconductor package DESeq2 in R (Version 1.32.0). CT27 and CT29 RNA-sequencing data were processed individually following standard processing.

### Histone modification ChIP-Seq and analysis

ChIP was performed on CT27 human TS cells[22] using commercially available ChIP Reagents (Nippon Gene, Tokyo, Japan). The following antibodies were used: H3K4me3 (1:500, MABI0304, Clone No. CMA304, MBL International, Woburn, MA), and H3K27ac (1:150, MABI0309, Clone No. CMA309, MBL International). The ChIP-Seq library was constructed using the Ovation Ultralow System V2 (NuGEN Technologies, Redwood City, CA) and sequenced on the Illumina HiSeq 2500 platform (Illumina). Basic quality control of raw FASTQ files was performed using HTStream (Version 1.3.2; https://github.com/s4hts/HTStream). PhiX reads were removed using hts_SeqScreener with default parameters. Duplicates were removed using hts_SuperDeduper (-e 250000). Adapter sequences were trimmed with hts_AdapterTrimmer (-p 4). Unknown nucleotides (N) were removed using hts_Ntrimmer. Base quality trimming was performed using a minimum average quality score of 20 (-q 20) in a 10-bp sliding window (-w10) with hts_QWindowTrim. Trimmed reads shorter than 50 bp as well as orphaned reads from a pair were removed using hts_LengthFilter (-n -m 50). Reads were aligned to the human genome (GRCh38.86) using BWA mem (Version 0.7.17-r1188). Alignments in the ENCODE blacklisted regions (https://doi.org/10.5281/zenodo.1491733) were removed using Bedtools intersect (with option '-v'; Version 2.30.0). Peaks for each sample were called independently using the callpeak function in MACS3 (Version 3.0.0a6) with a minimum FDR of 0.01 (-q 0.01).

### Chromatin capture by Hi-C

Cells were rinsed with 5 mL of PBS and then 5 mL PBS containing 2% formaldehyde were added and incubated for 10 min. Then, 550 µL of glycine 1.25 M (final conc 0.125 M) were added and incubated for 5 min at room temperature before an additional 15 min incubation on ice. Liquid was removed and cells were rinsed two times with 5 mL ice-cold PBS. Cells were scraped and transferred into 2 mL low bind tubes to have the equivalent of $1 \times 10^6$ cells (1 Petri Dish). Nuclei were pelleted at 2500 rpm for 10 min at 4 °C (repeat if necessary) and all supernatant was removed, and flash frozen on dry ice and stored at −80 °C until ready for use. Crosslinked samples were thawed on ice and prepared for Hi-C using the Arima-HiC kit (Arima Genomics, San Diego, CA). Cross-linking was performed according to the Arima-HiC protocol with the following modifications. At step 5, the incubation at 37 °C was increased to a duration of 60 min and the samples were maintained overnight at 4 °C after step 11. Samples were then cleaned with AMPure XP Beads as described in the protocol and all quantification and quality control steps that required the use of a Qubit were performed using a Qubit dsDNA HS Assay Kit. Proximally ligated DNA (1000 ng/sample) was used as an input into the library preparation protocol. Samples

were sheared to 400 bp using a Covaris LE220-plus system using the manufacturer's recommended settings. Sizing was confirmed using a High Sensitivity DNA Bioanalyzer kit. Size-selection as described in the Arima-HiC protocol was followed and the entire size-selected sample was used as input into biotin enrichment and library preparation with a KAPA HyperPrep kit and Roche SeqCap Adapters (7141530001 or 7141548001, Roche). Instead of performing the Arima-HiC quality control step 2, the number of cycles needed for library amplification was determined as follows: perform 10 PCR cycles for 125–200 ng of input, perform 9 PCR cycles for 200–400 ng of input, perform 8 PCR cycles for 400–600 ng of input, and perform 7 cycles of PCR for inputs >600 ng. The completed libraries were then quantified using a Qubit dsDNA BR Assay Kit and the library size was determined with a High Sensitivity DNA Bioanalyzer Kit. Standard Illumina Free-Adapter Blocking was performed on the libraries. Cleaned, adapter-blocked libraries were loaded on a NovaSeq 6000 with a run configuration of $151 \times 8 \times 8 \times 151$ and an average depth of 900 M PE reads per library.

To discover chromatin loops the standard Juicer pipeline[39] was used where contact maps for each sample were generated using the Arima Genomics fragment map file specific to GRCh38 and default parameters. After merging contact maps for each cell state, stem and EVT, chromatin loops were discovered with HiCCUPS (Juicer Tools Version 1.22.01[39];) using a matrix size of 1000 (-m 1000) at 5, 10, and 25 kb resolution (-r 25000,10000,5000). Differential loops between EVT and stem cells were identified using HiCCUPSDiff (Juicer Tools Version 1.22.01[39];) with a matrix size of 1000 (-m 1000). The chromatin loops as well as the differential loops were filtered to exclude contacts in the ENCODE blacklisted regions ([73]; https://doi.org/10.5281/zenodo.1491733) using Bedtools intersect (with option '-v'; Version 2.30.0[74];). Furthermore, genes with a TSS within 10 kb of a loop region were linked to each loop.

The Bedtools toolkit was used to compute the depth of coverage of the ATAC read data genome-wide in 100 bp bins, normalizing number of ATAC reads in each bin for every 10 M total reads for a given sample. All bins within 20 kb of either end of each of the EVT and stem cell loops were then extracted. To generate control sets, 1000 randomly selected intervals (chose from chr1-22, X, Y) that do not intersect any of the loops and do not intersect our list of blocked regions were generated. Three different control sets with intervals of size 5 kb, 10 kb and 25 kb were generated. All ATAC bins within 20 kb of the intervals in each of these control sets were extracted.

Aggregate contact plots were generated with hicAggregateContacts (HiCExplorer Version 3.7.2)[75] using intra-chromosomal contacts at the 5-kb resolution. Number of bins was set to 20 and the obs/exp transformation was used for the contact matrix. Heatmaps were generated with Juicebox 1.11.08.

## Transcription factor motif analysis
Motif analysis for ATAC-Seq data was performed using HOMER's findMotifsGenome.pl script (option '-size-given200'; Version 4.11). Motif analysis for Hi-C data in the chromatin contact regions were also discovered using HOMER's findMotifsGenome.pl script (option '-size given'; Version 4.11[76];).

## TFAP2C ChIP-seq and analysis
ChIP was performed on CT27 human TS cells[22] in three replicates using the SimpleChiP Enzymatic Chromatin IP kit (Cell Signaling Technology. Anti-rabbit AP-2 gamma antibody (1:50, 2320, Cell Signaling) was used in the analysis. A total of 100 ng of fragmented DNA in 25 μL of water was used as input in library preparation. A KAPA HyperPrep Kit (07962347001, Roche) was used for library preparation, following the protocol associated with the kit with the following modifications. During end repair and A-tailing, 35 μL of water was added to the master mix containing 7 μL of end-repair and A-tailing buffer and 3 μL of end-repair and A-tailing enzyme. Forty-five μL of

the end-repair and A-tailing master mix was added to each sample and was mixed thoroughly by pipetting. During the adapter ligation step, Illumina TruSeq DNA LT adapters, supplied at 15 μM, were diluted 1:100 in water and 5 μL of diluted adapters were added to the sample. A master mix of 30 μL of ligation buffer and 10 μL of DNA ligase was made and 40 μL of ligation master mix was added to each sample. After adapter ligation, a 0.8x bead clean-up was performed using AMPure XP Beads (A63881, Beckman Coulter) with a 10 min incubation at room temperature to allow the DNA to bind to the beads and a 7 min incubation after two 80% ethanol washes to allow the beads to dry. Once the beads were dry, the beads were resuspended in 25 μL of water and incubated at room temperature for 2 min. Twenty μL of post-ligation sample was then transferred to a new tube to be used in library amplification. The PCR library amplification master mix consisted of 25 μL of 2x KAPA HiFi HotStart ReadyMix, 1.5 μL of 10x PCR primer cocktail, and 3.5 μL of water. Thirty μL of the PCR master mix was added to each post-ligation sample and 13 cycles of PCR were performed. After library amplification, the library underwent a 0.6/0.8x double-sized size selection with 10 min incubations to allow the library to bind to the beads. Two 80% ethanol washes were performed, and the beads were allowed to dry at room temperature for 2 min. Thirty-six μL of water was added to the dried beads and was incubated at room temperature for 2 min to elute the DNA. Thirty-four μL of the final, size-selected library was transferred to a new tube. The final library concentration was assessed with a Qubit dsDNA HS Kit (Q32854, Thermo-Fisher), and the final size was determined, centering at ~400 bp, with a D1000 ScreenTape and D1000 reagents (5067- 5582 and 5067- 5583, Agilent). Cleaned, adapter-blocked libraries were loaded on a NovaSeq 6000 with a run configuration of $151 \times 8 \times 8 \times 151$ and sequenced to an average depth of 20 M PE reads per library. Basic quality control of raw FASTQ files was performed using HTStream (Version 1.3.2; https://github.com/s4hts/HTStream). PhiX reads were removed using hts_SeqScreener with default parameters. Duplicates were removed using hts_SuperDeduper (-e 250000). Adapter sequences were trimmed with hts_AdapterTrimmer (-p 4). Unknown nucleotides (N) were removed using hts_Ntrimmer. Base quality trimming was performed using a minimum average quality score of 20 (-q 20) in a 10-bp sliding window (-w10) with hts_QWindowTrim. Trimmed reads shorter than 50 bp as well as orphaned reads from a pair were removed using hts_LengthFilter (-n -m 50). Reads were aligned to the human genome (GRCh38.86) using BWA mem (Version 0.7.17-r1188). Alignments in the ENCODE blacklisted regions (https://doi.org/10.5281/zenodo.1491733) were removed using Bedtools intersect (with option '-v'; Version 2.30.0). Peaks for each sample were called independently using the callpeak function in MACS3 (Version 3.0.0a6) with a minimum FDR of 0.01 (-q 0.01).

## Sample collection and processing for single cell RNA-sequencing
Study participants were screened at the University of Kansas Health System Advanced Reproductive Medicine Clinic.

Inclusion criteria were age >18 years and <42 years at the time of conception, BMI > 18 kg/m² and <30 kg/m², and a history of unexplained RPL with recent diagnosis of miscarriage. Diagnosis of RPL was made using recommendations by the American Society for Reproductive Medicine Practice Committee[77] with RPL defined as the spontaneous loss of two or more pregnancies. For all patients known etiologies for miscarriage were ruled out before allowing participation in study including: Mullerian anomaly, polycystic ovary disease, thyroid disease, diabetes, coagulopathy, balanced translocation or structural rearrangement from sperm or oocyte contributor, and obesity. Miscarriage was diagnosed using guidelines provided by the American College of Obstetrician and Gynecologists (ACOG) Committee on Practice[78]. The ACOG guidelines define miscarriage as a nonviable, intrauterine pregnancy with (i) an empty gestational sac or (ii) a

gestational sac containing an embryo or fetus without fetal cardiac activity within the first 12 6/7 weeks of gestation[78]. All study participants had transvaginal ultrasound to confirm diagnosis of miscarriage.

Placental and decidual tissue were collected from the study participants where all samples had normal karyotype based on clinical cytogenetics testing including chromosomal analysis (GTG banded chromosomes analyzed at the 450–550 band levels) and array comparative genomic hybridization performed in accordance with current International Standing Committee on Human Cytogenetic Nomenclature (ISCN 2009) (Supplementary Table 2). Fetal sex was determined from clinical cytogenetics testing but was not considered in any analysis as this was out of the scope of the study.

Each sample was rinsed in PBS before the addition of 5 mL of Digestion Medium [DMEM/F12 (11320033, Thermo-Fisher) with 10% FBS, and 100 units/mL of penicillin and 100 µg of streptomycin (15140122, Thermo-Fisher) with 2.0 mg/mL collagenase type 1A] and incubated for 50 min at 37 °C. The sample was strained through a prewetted 100 µm filter in a 50 mL conical tube and rinsed twice with 5 mL of DMEM/F12 with 10% FBS and 100 units/mL penicillin and 100 µg/mL streptomycin and centrifuged at $300 \times g$ for 8 min. Cell pellets were resuspended in 1 mL of cold Recovery Cell Culture Freezing Medium (12648010, Thermo-Fisher), and the cell suspension was transferred to a cryogenic storage vial. The cryogenic storage vial was placed in a Corning CoolCell FTS30, which was then placed in a −80 °C freezer overnight. Samples were stored at −80 °C before being thawed and processed for single-cell RNA-Seq.

## 10X Genomics single cell capture and sequencing
Prior to single-cell capture, samples were thawed in 10 mL of Thawing Medium consisting of DMEM/F-12 (11320033, Thermo-Fisher) supplemented with 10% FBS and 100 units/mL of penicillin and 100 µg/mL of streptomycin (15140122, Thermo-Fisher) that was prewarmed in a 37 °C bead bath and centrifuged at $300 \times g$ for 8 min. The supernatant was carefully removed without disturbing the cell pellets. The cell pellets were each resuspended in 0.5 mL of Thawing Medium, and the cell suspensions were placed on ice and passed through a pre-wetted 40 µm nylon mesh cell strainer. Cell suspensions were centrifuged at $300 \times g$ for 8 min at 4 °C, and the supernatant was carefully aspirated without disturbing the cell pellets. The cell pellets were resuspended in 100 µL of cold Thawing Medium, and cell count and viability were assessed using 0.4% Trypan Blue and a Countess II automated cell counter. For each sample, Chromium Chip B (100153, 10x Genomics) were loaded (Supplementary Table 2). Following cell loading, cDNA and library preparation was performed identically for all samples using the Chromium Single Cell 3′ Library & Gel Bead Kit v3 (1000075, 10x Genomics) according to the manufacturer's protocol. Libraries were sequenced with a NovaSeq 6000 using 2 × 94 cycle paired-end. Sequenced reads were initially processed by the CellRanger pipeline which includes FASTQ creation, read alignment, gene counting, and cell calling. The Cell Ranger GRCh38(v2020-A) genome was used as the reference for alignment.

## Post-sequencing analysis of single-cell RNA datasets
The Cell Ranger matrix file from the single-cell RNA sequencing (scRNA-Seq) of RPL samples was processed along with the Cell Ranger matrices of two publicly available datasets[27,28] of gestational age-matched normal control POC samples profiled using the same platform (10X Genomics, Supplementary Table 2) using standard workflow implemented in Seurat 4.1.1[79]. Each sample was individually processed with the SCTransform function[80] using the glmGamPoi package[81] in Bioconductor 1.30.18 using R 4.1 with capture as a batch variable when applicable. The three datasets were integrated using the standard integration workflow for data normalized with SCTransform[80] and 3000 integration features were selected for downstream analysis. After integration, linear dimensional reduction, nonlinear dimensional reduction, nearest neighbor finding, and unsupervised clustering were completed according to the standard workflow[79]. Cell type label transfer was performed by referencing the Vento-Tormo dataset[28]. The predicted cell identities (ID) were generated using the FindTransferAnchors and TransferData functions of Seurat[79]. If necessary, cell type labels were manually adjusted based on marker gene expression to exclude ambiguous cell clusters. Cells with >50% mitochondrial reads and <2000 molecules were removed from all three datasets. The Vento-Tormo dataset[28] had an additional filtering where cells with a mitochondrial ratio of >20% and <3500 Unique Molecular Identifiers (UMIs) and 1250 unique genes were removed.

## Differential expression analysis
For differential expression analysis, genes were filtered according to the parameters used for Vento-Tormo dataset[28]. The raw UMI counts of samples with multiple captures were summed prior to differential expression analysis.

Genes that were expressed in three or more cells per dataset were kept for pseudobulk differential expression analysis. A matrix was aggregated to find the UMI counts by gene per sample ID within a specific cell type. Using the raw counts and metadata of a specific cell type, a DESeq dataset[82] was created, and the counts were transformed by the vst function for principal component analysis. Differential expression testing was performed by the DESeq function with default parameters on the generated DESeq dataset[82]. Pairwise differential expression contrasts were performed according to the inferred cell type labels (EVT or SCT/VCT) where Vento-Tormo[28] and Suryawanshi[27] datasets were combined as the control group. The contrast was performed with the results function from DESEq2 using Bioconductor 1.30.18 with default parameters (Benjamini-Hochberg p-value adjustment) and alpha equal to 0.05[82]. Log2 fold changes were shrunk with the lfcShrink function with the apeglm package[82,83]. For a gene to be significantly differentially expressed, the Benjamini-Hochberg adjusted $p$-value was <0.05.

## Gene set enrichment analysis
The GSEA 4.3.2 software[84,85] was implemented using three gene sets ($N = 250$ genes/set representing upregulated genes from scRNA-Seq analysis in EVT and SCT/VCT, respectively, as well as a gene set representing genes not differentially expressed). Normalized gene expression count for each EVT ($n = 6$ biologically independent replicates) and stem ($n = 6$ biologically independent replicates) cell state from the TS cell in vitro model was used with EVT and stem labeled as phenotype 0 and 1, respectively.

## Data visualization
Data from the scRNA-Seq assays were normalized and scaled with the NormalizeData and ScaleData functions of Seurat[79]. Violin plots and feature plots were then generated using default parameter with the latter using expression values thresholded at the 5th and 95th percentiles using the FeaturePlot parameters "min.cutoff = q5" and "max.cutoff = q95".

## Human placental specimens
Sections of paraffin-embedded first trimester placenta tissue were obtained from the Lunenfeld-Tanenbaum Research Institute (Mount Sinai Hospital, Toronto, Canada) with written informed consent.

## In situ hybridization
Transcripts for *TFAP2C, EPAS1, SNAI1, PLAC8,* and *CDH1* were localized within human first trimester placental tissue sections. RNAscope Multiplex Fluorescent Reagent Kit version 2 (Advanced Cell Diagnostics) was used for in situ hybridization analysis. Probes were prepared to detect *TFAP2C* (NM_003222.3, 515921, target region: 664–1596), *EPAS1* (NM_001430.4, 410591, target region: 1332–2354),

*SNAI1* (NM_005985.3, 560421, target region: 17–1233), *CDH1* (NM_004360.3, 311091-C2, target region: 263–1255), and *PLAC8* (NM_016619.3, 858491-C2, target region: 5–1448). *CDH1* and *PLAC8* probes were used to identify cytotrophoblast/trophoblast progenitor cells and differentiated EVT cells of the EVT cell column, respectively[48,49]. Fluorescence images were captured on a Nikon 90i upright microscope (Nikon) with a Photometrics CoolSNAP-ES monochrome camera (Roper).

### shRNA-mediated gene silencing

Lentivirus-mediated shRNA delivery was used as a tool to test the biological roles for TFAP2C, EPAS1, and SNAI1 in human TS cells. Several shRNAs designed for each target were subcloned into pLKO.1 and their efficacy tested by RT-qPCR and western blotting. An shRNA which does not recognize any known mammalian gene, pLKO.1-shSCR (Plasmid 1864) was used as a control (Addgene, Cambridge, MA). Sequences for shRNAs used in the analysis are provided in Supplementary Table 3. shRNA sequences were subcloned into a pLKO.1 vector at restriction sites for *AgeI* and *EcoRI*. The three lentiviral packaging vectors included pMDLg/pRRE (plasmid 12251), pRSV-Rev (plasmid 12253), and pMD2.G (plasmid 12259; Addgene). Lentiviral particle production followed transient transfection of both the shRNA-pLKO.1 vector and packaging plasmids into Lenti-X cells (632180, Takara Bio USA, Inc., Mountain View, CA). Lenti-X cells were plated in 0.001% poly-L-lysine (P4707, Sigma-Aldrich) in PBS-coated 6-well tissue culture-treated plates. Transient transfection was performed with Attractene (301005, Qiagen, Redwood City, CA) in DMEM (11995-065, Thermo-Fisher) with 100 U/ml penicillin and 100 µg/ml streptomycin (15140122, Thermo-Fisher). Lenti-X cells were cultured in DMEM plus 10% FBS, 100 U/ml penicillin, and 100 µg/ml streptomycin until 24 h prior to supernatant collection, at which time the Lenti-X cells were cultured in Basal Human TS Cell Medium [DMEM/F12 (11320033, Thermo-Fisher), 100 µm 2-mercaptoethanol, 0.2% (vol/vol) FBS, 50 µM penicillin, 50 U/mL streptomycin, 0.3% bovine serum albumin (BP9704100, Thermo-Fisher), 1% Insulin-Transferrin-Selenium-Ethanolamine solution (vol/vol, Thermo-Fisher)], 1.5 µg/mL L-ascorbic acid (A8960, Sigma-Aldrich), and 50 ng/mL epidermal growth factor (E9644, Sigma-Aldrich)]. Viral supernatants were collected at 24 and 48 h post-basal human TS cell medium change and stored at −80 °C.

To perform lentiviral transductions, human TS cells were plated at 80,000 cells per well in six-well tissue culture treated plates coated with 5 µg/mL collagen IV (CB40233, Thermo-Fisher) and incubated for 24 h. Immediately before transduction, culture medium was changed, and human TS cells were incubated with 2.5 µg/mL polybrene for 30 min at 37 °C. Following polybrene treatment human TS cells were transduced with 500 µL of lentiviral supernatant and incubated for 24 h. Culture medium was changed 24 h post-transduction and transduced cells were selected with puromycin dihydrochloride (5 µg/mL, A11138-03, Thermo-Fisher) for two days. Recovered human TS cells were cultured for 1–3 days in Complete Human TS cell Medium prior to use in assays.

### Transcript quantification by RT-qPCR

RNA was isolated with TRIzol/chloroform precipitation (15596018, Thermo-Fisher) following the manufacturer's instructions. cDNA was synthesized from 1 µg of total RNA using the High-Capacity cDNA Reverse Transcription Kit (Thermo-Fisher, 4368813) and diluted 10 times with ultra-pure distilled water. qPCR was performed using PowerSYBR Green PCR Master Mix (4367659, Thermo-Fisher) and primers (250 nM each). PCR primer sequences are listed in Supplementary Table 4. Amplification and fluorescence detection were measured with a QuantStudio 5 Flex Real-Time PCR System (Thermo-Fisher). An initial step (95 °C, 10 min) preceded 40 cycles of a two-step PCR (92 °C, 15 s; 60 °C, 1 min) and was followed by a dissociation step

(95 °C, 15 s; 60 °C, 15 s; 95 °C 15 s). The comparative cycle threshold method was used for relative quantification of the amount of mRNA for each sample normalized to housekeeping genes *B2M* or *POLR2A*. Statistical analyses were performed using GraphPad Prism 8 software. Welch's *t* tests, Brown-Forsythe and Welch ANOVA tests, or Two-way ANOVA tests were applied when appropriate. Data is represented as mean ± standard deviation with the statistical significance level set at $p < 0.05$. Statistical test details and *p* values are presented in figure legends.

### Western blotting

Cell lysates were sonicated in radioimmunoprecipitation assay (RIPA) lysis buffer (sc-24948A, Santa Cruz Biotechnology, Dallas, TX). Protein concentrations were measured with the DC Protein Assay (5000112, Bio-Rad) and RIPA buffer was used as the blank standard. Proteins were separated by SDS-PAGE and transferred onto polyvinylidene difluoride (PVDF) membranes (10600023, GE Healthcare, Chicago, IL). Following the transfer, PVDF membranes were blocked with 5% non-fat milk in Tris-buffered saline with 0.1% Tween 20 (TBST) and probed with primary antibodies to SNAI1 (1:1000, 3879 S, Cell Signaling Technology), EPAS1 (1:1000, 66731-1-Ig, Proteintech), TFAP2C (1:750, 6E4/4, sc-12762, Santa Cruz Biotechnology), and glyeraldehyde-3-phosphate dehydrogenase (GAPDH, 1:5000, AM4300, Thermo-Fisher) overnight at 4 °C. PVDF membranes were washed three times for 5 min each with TBST and incubated with secondary antibodies (goat anti-rabbit IgG HRP, 1:5000, 7074S, Cell Signaling Technology and horse anti-mouse IgG HRP, 7076, Cell Signaling Technology) for 1 h at room temperature. Immunoreactive proteins were visualized using the Luminata™ Crescendo Western HRP Substrate (WBLUR0500, Millipore, Billerica, MA) according to the manufacturer's instructions. Full scan blots are presented in the Source Data file.

### RNA-seq analysis for silencing experiment

Stranded mRNA-Seq was performed using a NovaSeq 6000. Quality control was completed using the Agilent TapeStation 4200 with the RNA ScreenTape Assay kit (Agilent Technologies 5067–5576). Total RNA (1 µg) was used to initiate the library preparation protocol. The total RNA fraction was processed by oligo dT bead capture of mRNA, fragmentation, reverse transcription into cDNA, end repair of cDNA, ligation with the appropriate Unique Dual Index (UDI) adaptors, strand selection, and library amplification by PCR using the Universal Plus mRNA-Seq with NuQuant library preparation kit (0520-A01, Tecan Genomics). Library validation was performed using the D1000 ScreenTape Assay kit (5067–5582, Agilent) on the Agilent TapeStation 4200. The concentration of each library was determined with the NuQuant module of the library prep kit using a Qubit 4 Fluorometer (Thermo-Fisher/Invitrogen). Libraries were pooled based on equal molar amounts and the multiplexed pool was quantified, in triplicate, using the Roche Lightcycler96 with FastStart Essential DNA Green Master (06402712001, Roche 0) and KAPA Library Quant (Illumina) DNA Standards 1–6 (KK4903, KAPA Biosystems). Using the qPCR results, the RNA-Seq library pool was adjusted to 2.125 nM for multiplexed sequencing. Pooled libraries were denatured with 0.2 N NaOH (0.04 N final concentration) and neutralized with 400 mM Tris-HCl, pH 8.0. A dilution of the pooled libraries to 425 pM was performed in the sample tube on the instrument and followed by onboard clonal clustering of the patterned flow cell using the NovaSeq 6000 S1 Reagent Kit (200 cycle) (20012864, Illumina). A 2 × 101 cycle sequencing profile with dual index reads was completed using the following sequence profile: Read 1 – 101 cycles x Index Read 1 – 8 cycles x Index Read 2 – 8 cycles x Read 2 – 101 cycles. Following collection, sequence data were converted from.bcl file format to fastq file format using bcl2fastq software and de-multiplexed into individual sequences and downstream analysis performed as described above.

## GWAS data analysis and integration

Summary statistics from UK Biobank GWAS was accessed at http://www.nealelab.is/uk-biobank. Odds ratio of the proportion of nominally significant ($P < 0.05$) hits in the *EPAS1* gene region (GRCh38; chr2:46,150,000–46,450,000) was computed and compared with variants in a randomly selected matched window. This step was repeated 1000 times for each phenotype to generate a distribution of odds ratio. To compute significance using Wilcoxon test, a null distribution was generated to compare against. For each permutation, another randomly selected matched window size was selected and compared with the first randomly selected region chosen in step 1.

## Statistics and reproducibility

No statistical method was used to predetermine sample size. The experiments were not randomized. No data sets derived from the human TS cell model were excluded from any analyses. All genome-wide data sets were done on two independent TS cell donor lines. Experiments utilizing the TS cell model were independently repeated a minimum of three times for all results from experiments displayed as representative data (e.g., immunolocalization and immunocytochemistry images).

Prior to integration of publicly available single-cell RNA data sets from 1st trimester placental samples, three data sets were excluded from the Vento-Tormo study as outlined in Supplementary Table 2. Following data integration, one sample was excluded from the Suryawanshi study due to poor quality as outlined in Supplementary Table 2. Cause of pregnancy loss was blinded during data integration and initial analysis that included filtering and clustering of the data.

## Reporting summary

Further information on research design is available in the Nature Portfolio Reporting Summary linked to this article.

## Data availability

Source data are provided with this manuscript. Raw and processed sequencing data from ATAC-Seq, ChIP-Seq, RNA-Seq, Hi-C and scRNA-Seq have been submitted to the NCBI Gene Expression Omnibus (GEO) under the following accession numbers GSE204722 and GSE204723. Raw and processed data are available under unrestricted access. Processed data can be visualized in the UCSC Genome Browser using the following link: https://genome.ucsc.edu/s/cmri_gmc_bioinformatics/EVT_STEM_Epigenome_Review. Additional scRNA-Seq data sets used in the study are available in ArrayExpress under the accession number E-MTAB-6701 and in BioProject under the accession number PRJNA492324. Summary statistics from UK Biobank GWAS was accessed at http://www.nealelab.is/uk-biobank. The reference genome GRCh38 (GCA_000001405.22) used in this study is available at http://ftp.ensembl.org/pub/release-86/fasta/homo_sapiens/dna/Homo_sapiens.GRCh38.dna.primary_assembly.fa.gz.

## Code availability

Only publicly available tools were used in data analysis as described wherever relevant in the Methods.

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

## Acknowledgements

We thank Stacy Oxley, Brandi Miller, and Nhu Bui for their administrative assistance, Dan Louiselle, and Rebecca Biswell for their work in sample processing, Adam Walters and Margaret Gibson for their work in library preparation and NGS and Bradley Belden and Mackenzie Stevens for their work in clinical coordination. Figures 1a and 9g were created with Biorender.com and the figures were exported under a paid subscription. The research was supported by the National Institutes of Health (F32HD096809 (K.M.V.), HD020676 (M.J.S.), HD099638 (M.J.S) HD105734 (M.J.S.), GM146966 (C.S.), HG012422 (C.S.), the Sosland Foundation (E.G. and M.J.S.), KAKENHI Grant Number 19H05757 (T.A.), and AMED Grant Number JP22gm1310001 (T.A.). E.G holds the Roberta D. Harding & William F. Bradley, Jr. Endowed Chair in Genomic Research.

## Author contributions

K.M.V., H.O., T.A., M.J.S., and E.G. conceived and designed the research; K.M.V., E.M.D, A.M., R.P.M., and H.O. performed experiments; H.O., T.A., M.L., K.H., and C.M. provided reagents. K.M.V, E.M.D., B.K., J.M.V., A.M., E.W., R.P.M., K.I., W.C., C.S-S., C.S., M.J.S., and E.G. analyzed the data and interpreted results of experiments; K.M.V., B.K., C. S-S., M.J.S., and E.G. prepared figures and drafted manuscript; K.M.V., M.J.S., and E.G. edited and revised manuscript; All authors approved the final version of manuscript.

## Competing interests

The authors declare no competing interests.
