## [Peer Review File · Nature Communications]

REVIEWER COMMENTS

Reviewer #1 (Remarks to the Author):

This study by Varberg & Dominguez et al. provides, in principle, a most valuable study on the regulatory changes associated with differentiation of human trophoblast stem cells into the extravillous cytotrophoblast lineage. A better understanding of the shifts in transcriptional regulation will be most informative for the identification of transcription factors directing this pivotal differentiation pathway, and the genomic regulatory elements (e.g., enhancers) associated with it.

However, unfortunately, this study suffers from too many pitfalls. Thus, the flow of the text is a back and forth between genomic analyses and individual gene studies, with none of the avenues properly pursued and ultimately resulting in a picture that makes this manuscript appear very cobbled together.

Major issues revolve around the samples chosen for sequencing analysis. Albeit stating that RNA-Seq and ATAC-Seq profiles were generated for "multiple replicates and donor TS cell lines", this is truly only the case for the RNA-seq data for which 5 replicates each of two independent TS cell lines were chosen. All other analyses are only performed on a single TS cell line, CT27. This singularity constitutes a major limitation of the biological robustness of these data. Moreover, only a single end point was chosen for investigation. The study would have benefited tremendously from at least a limited time course set-up studying the transition from stem cells towards EVT.

In the struggle of this manuscript between a multi-omic and gene-specific focus, multiple aspects are left open-ended. For example, if focussing on ATAC-seq data, it would have been valuable to follow these up with H3K27ac- and H3K4me1-ChIP-seq experiments to validate and unambiguously identify stem and EVT enhancer elements. As it stands, the ATAC-seq data are valuable but not explored in sufficient detail. Lines 158/159 state that 73% of sites were shared and 27% unique to EVT. Surely there must be some TS-specific peaks (as displayed in the figures), the numbers do not add up here. Given that only half of the EVT-unique peaks fall outside a TSS region (line 165), why does Fig. 2B not indicate any accessible peaks in the heatmap?

Moreover, it does not seem adequate to analyze the genomic datasets with a singular focus on EVTs, the converse enrichment of stem cell-specific enhancers should also be included, as this is the essential starting point of these cells as they differentiate into EVTs.

When it comes to the integration of TF binding sites, some fundamental information must be missing from the text (lines 179ff). What was integrated here, and from which databases, as the current study does not include any TF CHIP-seq in itself.?

If focussing on individual genes that stand out from the analysis, it would have been desirable to follow up any one of them in more detail than solely by ISHs on first trimester placentas, knockdown and RNA-seq. If the point is made that these genes are direct regulators of specific pathways, select factors would have been needed to be CHIP'ed to identify direct binding sites (that would be expected to show some significant overlap with ATAC-seq and HiC sites etc).

It would appear that EPAS1 is the most promising factor from all those that have been touched on. Why was a single cell study of RPL patients needed to establish a role of this TF in this condition? Why was RPL chosen (that often has an endometrial etiology) and not a more closely EVT-linked pregnancy complication, notably preeclampsia? If EPAS1 positively regulates EVT differentiation, as implied from the various genomic data, why is it up-regulated in RPL patients? This is contrary to expectations.

TFAP2C is identified because of its motif enrichment in EVTs; however, the KD experiments would indicate a more prominent role in stem cell maintenance. How can this discrepancy be reconciled?

Why was the integration of a single cell analysis needed to identify 5 EVT-enriched genes that are the single display item of Fig. 5?

Overall, these points exemplify the multitude of aspects on which this manuscript combines too many separate story lines, leaving the analyses and insights of any one of them at a somewhat superficial level.

Reviewer #2 (Remarks to the Author):

This of Vanberg and colleagues provides interesting insights into the regulatory DNA landscape that controls the identity and function of human extravillous trophoblast. Although the authors start with an in vitro differentiation model, validation of key observations in vivo is detailed and compelling. The paper is well-written and the figures and supplementary information are appropriate and clear. As listed

below, some points require further clarification. Perhaps the most contentious issue relates to the extrapolation of the findings to the pathophysiology of recurrent pregnancy loss.

1. The authors use an in vitro system to model EVT differentiation from TS. Induced marker genes are indeed bona fide EVT genes when cross-referenced with the maternal-fetal cell atlas. However, the authors do not quantify how many differentially expressed genes in this model system are concordant/discordant in vitro versus in vivo. This is important as a significant discordance rate signals usage of differential regulatory DNA regions in culture. In view of the access to clinical samples, it seems puzzling why no attempt was made to subject isolated primary EVT to ATAC-seq.

2. At a functional level, human EVT in vivo exhibits genome-wide polyploidization, cell cycle exit, and senescence (<https://doi.org/10.1371/journal.pgen.1007698>). Do these defining characteristics apply to EVT derived from TS in vitro? If so, how do endoreplication and cellular senescence impact/drive chromatin changes?

3. To which extent are unique accessible loci upon EVT differentiation accounted for by co-option of (lineage-specific) transposons into the regulatory DNA network?

4. L153: The description of the ATAC-seq peak calling indicates that TS and EVT were analysed separately, instead of peak calling on the combined data set. While the same FDR threshold was used for TS and EVT, the characteristics (height, width, footprints etc.) of so-called overlapping peaks can potentially be very different, resulting in the loss of valuable information. Likewise, it is unclear if the unique EVT ATAC-seq peaks not previously identified in the placenta have distinct characteristics? Again, ATAC-seq of purified EVT would have been informative.

5. The authors attempt to demonstrate the relevance of their findings by focussing on recurrent pregnancy loss. Clinical definitions are based on an arbitrary number of pregnancy losses. They have no biological basis as the risk of pregnancy loss increases stepwise by 8-10% which each additional loss, independently of maternal age (see <https://pubmed.ncbi.nlm.nih.gov/33394013/>). Without suitable controls, for example, samples of aneuploid pregnancy losses or so-called 'explained' miscarriages, the authors cannot infer that induction of EPAS1 is specific to recurrent (euploid) pregnancy loss. Upregulation of EPAS1 and associated genes may merely reflect an adverse intrauterine environment imposing stress on EVT (induction of DIO2, CDKN1C etc..)

6. The authors seem to construct a link between RPL and fetal genetic loci for birth weight. As over 93% of miscarriages occur before maternal-placental vascular connections are established at the end of the 1st trimester, the underlying pathological pathways are likely very different. Not only are GWAS data on miscarriage/RPL available, but there is a strong epidemiological link between the number of pregnancy

losses and the risk of preterm birth. Are SNPs of EPAS1 or its target genes implicated in the genetics of miscarriage or preterm birth?

7. Perhaps I overlooked but the authors should present a demographic table that includes sufficient detail on the number of samples, type of miscarriage presentation, length of gestation, and previous pregnancy outcomes.

Reviewer #3 (Remarks to the Author):

In this manuscript, Varberg et al utilize a relatively recent human TSC-to-EVT (extravillous trophoblast) differentiation approach to (i) map epigenetic and transcriptional changes in the starting and end populations and (ii) to identify and validate candidate key TFs that drive this cell fate transition. Strengths of the work include the high relevance for various placenta-associated phenotypes and diseases (e.g recurrent pregnancy loss), the use of both in vitro and in vivo samples for some validations, and the discovery of new regulators of EVT state, such as EPAS1. However, the limitations of the study are quite extensive on multiple levels, including the quality of the genomics data presented, the depth of the bioinformatics analyses, and reliability of the functional experiments. Overall, the study in its current form appears preliminary and with questionable impact and novelty. Below, are some of my key concerns:

1. The system that the authors utilize for EVT differentiation requires better characterization for efficiency and homogeneity. The authors should provide a quantitation of the percentage of EVT cells (e.g based on HLA-G or other specific marker) at the day of collection to help appreciate the potential contamination of their bulk analysis by other lineages (e.g, cytotrophoblasts or undifferentiated TS). Is sorting needed to improve homogeneity? They should also utilize the available scRNA-seq data from human placentas (as shown in Figure S8) to project the EVT vs TS signatures as they identified them by bulk RNA-seq, and vice versa (perform GSEA analysis of the EVT signatures as found by the scRNA-seq data on their differential TS vs EVT analysis). This could be critical for properly interpreting their results and gain confidence on the EVT specificity that they claim throughout the text. This is also crucial to appreciate novelty of their findings, since many of these factors have been previously reported to play roles in TS differentiation but for other lineages.

2. There are several issues with ATAC-seq and Hi-C data that the authors generate and present for TS and TS-derived EVT cells. First of all, more information and some examples of the raw data (e.g more examples of ATAC-seq tracks for shared and unique peaks; tornado plots of all peaks in TS and EVT clustered by category; read numbers, QC and actual heatmaps of the Hi-C data) would be very informative and help the reader appreciate the quality of the results or suggest improvements. In

addition, there are so many well-established methods that allow a better integration of these datasets (e.g aggregate plots of HiC signal around shared or cell-type specific peaks, virtual 4C around genes of interest) that are underutilized here. Also, as the authors acknowledge based on their results, HiC is limited in detecting potential regulatory loops. Therefore, the authors should perform a different orthogonal analysis, at least 4C-seq around genes or SE of interest to validate loops or detect cell-type specific connections that are likely missed by HiC.

3. The authors move on with several functional validations of candidate TFs in the TS-to-EVT differentiation system. Although, many of phenotypes are strong and interesting, many of their conclusions are compromised by the constitutive expression of shRNAs and the limited quantitation of the phenotypic effects -other than bulk RNA-seq. For example, for TFAP2C -given that this factor is important both TS biology, an inducible deletion system will be important to uncouple effects on the starting cells versus the potential role in differentiation. Moreover, a detailed quantitation of all the possible lineages (TS, EVT, other potential byproducts) based on IF analysis (or intercellular FACs) of multiple independent replicates will be very important to assess reproducibility and understand the specific role of each factor.

4. In general, although the manuscript is easy to read and understand, it is hard to follow the logic on the different types of integrative analysis that they used to nominate the few candidates that the focus on. A better organization of the manuscript with the integrative analysis first, followed by all in situ validations and then functional interrogations will improve the flow and help the reader compare and contrast results.

5. The figures overall seem very preliminary with superficial genomics analysis and poor presentation of the results.

Overall, the study is interesting but preliminary.

We are pleased to see the mutual enthusiasm from all Reviewers about our study.

- Reviewer #1 states that we provide **“a most valuable study on the regulatory changes associated with differentiation of human trophoblast stem cells into the extravillous cytotrophoblast lineage”**.
- Reviewer #2 indicates we provide **“interesting insights into the regulatory DNA landscape...”** ... **“validation of key observations in vivo is detailed and compelling”** and **“The paper is well-written and the figures and supplementary information are appropriate and clear”**.
- Reviewer #3 notes that **“Strengths of the work include the high relevance for various placenta-associated phenotypes and diseases (e.g., recurrent pregnancy loss), the use of both in vitro and in vivo samples for some validations, and the discovery of new regulators of EVT state, such as EPAS1. “**

We also thank them for their careful review and constructive comments. We acknowledge their requests and comments and have thoroughly and extensively responded to all the overarching concerns as outlined by the editor. Additional specific comments from each Reviewer are similarly addressed and outlined further below.

Include additional replicates (Reviewer #1)

“Albeit stating that RNA-Seq and ATAC-Seq profiles were generated for “multiple replicates and donor TS cell lines”, this is truly only the case for the RNA-seq data for which 5 replicates each of two independent TS cell lines were chosen. All other analyses are only performed on a single TS cell line, CT27. This singularity constitutes a major limitation of the biological robustness of these data.”

- We acknowledge this limitation and have now included not only technical and biological replicates but also additional donor trophoblast stem (TS) cell lines for all major NGS-based assays (i.e., RNA-Seq, ATAC-Seq and HiC) on both stem state and differentiated EVT cells. Using these new data sets, we now provide strengthened evidence about the robustness and consistency of the TS cell model. The main results are described in detail in the Results section (Page 5-6) and presented in Supplementary Figures 3, 7, 11, 12, 14, 16, 18, and 21A with updated sequencing statistics outlined in Supplementary Table 1. In addition, we also now present confirmatory results from both donor TS cell lines in all specific gene-based analysis and follow-up gene silencing experimentations, respectively (Supplementary Figure 19; Supplementary Figure 22-23).

Include timepoints (Reviewer #1)

“Moreover, only a single end point was chosen for investigation. The study would have benefited tremendously from at least a limited time course set-up studying the transition from stem cells towards EVTs.”

- We agree with the reviewer that a timeseries would be informative for the study of the transition from the stem state towards EVT cells. As such, we have now performed a time course experiment studying the global transcriptomic and morphological transition from the stem state to EVT cells at four time points (day 0, day 3, day 6 and day 8). We show that the global transcriptomic patterns as well morphology for EVT cells at 6 and 8 days of differentiation are similar and clearly distinct from the stem state and cells at 3 days of EVT cell differentiation. In addition, we show that expression of commonly used EVT cell lineage markers is maximal at days 6 and 8 of differentiation. The main results are described in detail in the Results section (Page 5) as well as presented in Figure 1, Figure 6, and Figure 7.

Provide further detailed analysis of ATACseq (Reviewer #1)

“...it would have been valuable to follow these up with H3K27ac- and H3K4me1-ChIP-seq experiments to validate and unambiguously identify stem and EVT enhancer elements. As it stands, the ATAC-seq data are valuable but not explored in sufficient detail.”

- Although we argue that ATAC-Seq is an efficient approach to identify open chromatin regions and inform about active regulatory regions, we acknowledge that it will not fully distinguish enhancers from promoter elements. To address this limitation, we now follow-up our ATAC-Seq results by integrating ChIP-Seq based histone modification data for active enhancers (H3K27ac) and promoters (H3K4me3) mapped in the CT27 line for stem state and EVT cells, respectively. Using these data, we found that 78% versus 76% of the differentially enriched ATAC-Seq regions map to an active enhancer (H3K27ac+ and H3K4me3-) or promoter (H3K4me3) in stem state versus EVT cells, respectively. However, when considering the epigenetic marks separately we noted a clear difference across the two cell states. Specifically, while 36% of differentially enriched ATAC-Seq regions in the stem state overlapped an active enhancer as many as 65% of the differentially enriched ATAC-Seq regions in the EVT cell state overlapped an H3K27Ac peak. Interestingly, differentially enriched ATAC-Seq regions in the stem state were an order of magnitude more likely to map to a poised/bivalent promoter (marked by a H3K4me3 peak without an overlapping H3K27Ac peak) than differentially enriched ATAC-Seq regions in the EVT cell state. Bivalent chromatin is characterized by having both repressing and activating histone marks and are importantly linked to cell differentiation. Thus, these enriched poised/bivalent promoter regions in the stem state likely silence genes that will activate EVT differentiation as exemplified for *HLA-G*. On the other hand, the enrichment of H3K27Ac peak among the differentially enriched ATAC-Seq regions in the EVT cell state further point to significant increased enhancer-driven gene regulation in EVT versus stem state cells. These results are described in detail in the Results section (Page 8) as well as presented in Figure 3C-D; Supplementary Figure 6; Supplementary Table 4a-b

Broaden analysis to stem cells (Reviewer #1)

*“Moreover, it does not seem adequate to analyze the genomic datasets with **a singular focus on EVT**s, the converse enrichment of stem cell-specific enhancers should also be included, as this is the essential starting point of these cells as they differentiate into EVT*s.”

- Although we agree with the reviewer that the stem state is the starting point of the TS model as they differentiate into EVT cells, the regulatory landscape of the stem state cells has been recently mapped (*Dong et al., eLIFE 9:e52504, 2020; Dong et al., Nat Commun 13:2548, 2022*) and thus remains less novel. As such we focus our in-depth analysis on mapping regulators of EVT cell differentiation. Nevertheless, we have now broadened the main analysis of the genomic datasets to also highlight the regulatory landscape of the TS cells in the stem state as presented in Supplementary Table 3a and Supplementary Table 7a. We now also show that for the identified consensus open chromatin regions (n=41,465 in total), 40,508 and 38,785 mapped to ATAC-Seq peaks in stem state and EVT cells, respectively. Based on estimated binding affinity measured by differences in ATAC-Seq read densities, we identified a total of 23,504 versus 9,334 regions that were differentially enriched (5% FDR) and considered more accessible in stem state versus EVT cells, respectively. As described above, by integrating ChIP-Seq based histone modification data for active enhancers (H3K27ac) and promoters (H3K4me3) for both the stem state and EVT cells we show clear differences in regulatory element activity across the cell states. These results are described in detail in the Results section (Page 6-7) as well as presented in Figure 3C-D, and Supplementary Table 4a and Supplementary Table 7a.

Include additional detailed follow up on regulators (Reviewers #1, #3)

“If focusing on individual genes that stand out from the analysis, it would have been desirable to follow up any one of them in more detail than solely by ISHs on first trimester placentas, knockdown and RNA-seq. If the point is made that these genes are direct regulators of specific pathways, select factors would have been needed to be ChIP’ed to identify direct binding sites (that would be expected to show some significant overlap with ATAC-seq and HiC sites etc.). “

- In the manuscript, we are careful not make inferences regarding direct targets for specific transcription factors. Antibodies to TFAP2C are available and well characterized for ChIP-Seq analysis, which is unfortunately not the case for SNAI1 or EPAS1. Consequently, in the revision we have included ChIP-Seq experimentation and analysis for TFAP2C and made some inferences about direct targets for TFAP2C. These results are described in detail in the Results section (Page 13) as well as presented in Figure 6H, and Supplementary Figure 20B.

“The authors move on with several functional validations of candidate TFs in the TS-to-EVT differentiation system. Although, many of phenotypes are strong and interesting, many of their conclusions are compromised by the constitutive expression of shRNAs and the limited

quantitation of the phenotypic effects -other than bulk RNA-seq. For example, for TFAP2C -given that this factor is important both TS biology, an inducible deletion system will be important to uncouple effects on the starting cells versus the potential role in differentiation. Moreover, a detailed quantitation of all the possible lineages (TS, EVT, other potential byproducts) based on IF analysis (or intercellular FACs) of multiple independent replicates will be very important to assess reproducibility and understand the specific role of each factor.”

- The functional validation performed for the candidate TFs provide support for the potential contributions of each TF to TS cell biology within the *in vitro* model system. An shRNA-based approach was chosen for the stability of target knockdown to enable phenotypic assessment in the stem state and following up to eight days of EVT cell differentiation. We are utilizing a well- established and validated EVT cell differentiation model (Okae et al. Cell Stem Cell 22:50, 2018). Additionally, a control shRNA was utilized for all experimental analyses to assess protocol-associated non-specific effects. Most importantly, we have now replicated **all functional analyses** on a second TS cell line (CT29). These new results are described in detail in the Results section (Page 13-16) as well as presented in Figure 5H and Supplementary Figure 19 Supplementary Figure 22-23.
- An inducible system could be informative for additional follow up investigations. Specifically, as indicated by the reviewer, an inducible system could provide insights into the action of a transcription factor (TF) such as *TFAP2C*, that contributes to the stem state and possibly to EVT cell differentiation. Inducible systems are not without concerns. Frequently used inducers such as doxycycline and tamoxifen are not inert, especially regarding the biology of trophoblast cells. Finally, it is important to recognize that *TFAP2C* is not the only TFAP2 family member expressed in EVT cells. *TFAP2A* is more abundantly expressed in distal regions of the EVT cell column and may have more relevance to differentiated trophoblast cell states.

Quantify DEGs between in vitro/vivo (Reviewer #2)

“The authors use an in vitro system to model EVT differentiation from TS. Induced marker genes are indeed bona fide EVT genes when cross-referenced with the maternal-fetal cell atlas. However, the authors do not quantify how many differentially expressed genes in this model system are concordant/discordant in vitro versus in vivo. This is important as a significant discordance rate signals usage of differential regulatory DNA regions in culture.”

- We thank the reviewer for this excellent suggestion. We have now included results from quantification of differentially expressed genes in the TS cell model (*in vitro*) that are concordant/discordant with the results from single-cell expression data (*in vivo*). Using highly upregulated genes (\log_2 fold change >5) in EVT cells from the *in vitro* TS cell model, we show that 70% are also upregulated in *in vivo* (\log_2 fold change >0.5) with similar positive trend for genes upregulated at different thresholds. These results are

described in detail in the Results section (Page 5-6) as well as presented in Supplementary Figure 4.

- In addition, in response to a request by Reviewer #3 (see detail below), we have now also performed gene set enrichment analysis (GSEA) of the EVT and SCT/VCT signatures as found by the single-cell expression data (*in vivo*) on the differential stem state vs EVT analysis from the *in vitro* TS cell model. Specifically, we generated three gene sets each consisting of 250 markers from the *in vivo* single-cell expression data representing genes upregulated in 1) EVT, 2) SCT/VCT as well as 3) genes not differentially expressed between EVT and SCT/VCT. Using the GSEA computational method (<https://www.gsea-msigdb.org/gsea/index.jsp>), we now present that the two a priori defined sets of genes from the *in vivo* data set (representing EVT and SCT/VCT markers) show statistically significant, concordant differences between the two *in vitro* biological states (stem state and EVT) in the expected direction. Importantly, the a priori defined set of genes from the *in vivo* data set of non-differentially expressed genes between EVT and SCT/VCT did not show statistically significant, concordant differences between the two *in vitro* biological states (stem state and EVT). These new results are described in detail in the Results section (Page 5-6), as well as in Figure 2B-D.

Further characterize *in vitro* EVT as well as differentiation system (polyploidization, cell cycle exit, heterogeneity, differentiation efficiency etc.) (Reviewers #2, #3)

*“At a functional level, human EVT *in vivo* exhibits genome-wide polyploidization, cell cycle exit, and senescence (<https://doi.org/10.1371/journal.pgen.1007698>). Do these defining characteristics apply to EVT derived from TS *in vitro*? If so, how do endoreduplication and cellular senescence impact/drive chromatin changes?”*

- Findings generated from our group and others with human TS cells provide strong evidence that the *in vitro* model system used in this experimentation most closely recapitulates trophoblast differentiation events occurring within the EVT cell column (Perez-Garcia et al., *eLife* 10:e635254, 2021; Varberg et al., *PNAS* 118:e2016517118, 2021; Muto et al., *PNAS* 118:e2111267118, 2021; Sheridan et al., *Development* 148:dev199749, 2021; Dong et al., *Nat Commun* 13:2548, 2022; Mizutani et al., *Mol Hum Reprod* 28:gaac032, 2022; Ray et al., *PNAS* 119:e2204069119, 2022; Kuna et al., *bioRxiv*, 202206.15.496287, 2022). The *in vitro* transition from stem state to EVT cell state best represents the transition of cells from the proximal to distal end of the cell column. Endoreduplication is initiated *in vitro* during TS cell differentiation to EVT cells (R.L Scott and M.J. Soares, unpublished findings) and is widespread in EVT cells isolated from the uterine decidua (Velicky et al. *Plos Genet* 14:e1007698, 2018; Morey et al., *Front Cell Dev Biol* 9:702046, 2021). Examining the effects of endoreduplication and cellular senescence on the epigenome are worthwhile efforts but are beyond the scope of this manuscript. In summary, the human TS cell model, as investigated in this report, best reflects events transpiring within the EVT cell column and not fully mature EVT cells

situated within the uterine parenchyma. This important issue has been addressed in the revised Discussion (Page 19).

*“The system that the authors utilize for EVT differentiation **requires better characterization** for efficiency and homogeneity. The authors should provide a quantitation of the percentage of EVT cells (e.g., based on HLA-G or other specific marker) at the day of collection to help appreciate the potential contamination of their bulk analysis by other lineages (e.g., cytotrophoblasts or undifferentiated TS). Is sorting needed to improve homogeneity?”*

- Culture and characterization of the cytotrophoblast-derived TS cells and the protocol for EVT cell differentiation were originally described by Okae and colleagues (*Okae et al. Cell Stem Cell 22:50, 2018*). EVT cells generated using these culture conditions abundantly and specifically express *HLA-G*, a non-classical MHC-I molecule, but a specific marker of *in vivo* EVT cells. Nevertheless, to further characterize the EVT cells used in our experimentation we have now assessed the percentage of HLA-G+ cells by flow cytometry in CT27 and CT29 cell lines at days 0, 3, 6, and 8 of EVT cell differentiation. HLA-G is significantly and consistently induced in both CT27 and CT29 cells starting on day 3 of EVT cell differentiation and reaches maximal expression by days 6 and 8. With 80-90% of cells staining positive for HLA-G by day 8 of EVT cell differentiation, reasonable homogeneity is achieved without the need for sorting in this system. These new results are described in the Results section (Page 5) as well as presented in Figure 1D; Supplementary Figure 1B; Supplementary Figure 2.

Assess relationship between EPAS1 SNPs and genetics of miscarriage or preterm birth (Reviewer #2)

“The authors seem to construct a link between RPL and fetal genetic loci for birth weight. As over 93% of miscarriages occur before maternal-placental vascular connections are established at the end of the 1st trimester, the underlying pathological pathways are likely very different. Not only are GWAS data on miscarriage/RPL available, but there is a strong epidemiological link between the number of pregnancy losses and the risk of preterm birth. Are SNPs of EPAS1 or its target genes implicated in the genetics of miscarriage or preterm birth?”

- First, we would like to clarify that we did not intend to construct a link between RPL and birth weight but rather highlight that failure of proper EVT cell invasion into the uterine compartment may lead to various pregnancy complications, such as pregnancy loss or fetal growth restriction leading to low birth weight and that *EPAS1* may play a role in these complications. While we are not aware of any publicly available, robust GWAS data linking fetal genetics to risk of miscarriage, we attempted to access summary statistics from GWAS results from large efforts in fetal genetics of preterm birth (*Liu et al, Nat Commun 10:3927, 2019*) and birth weight (*Warrington et al, Nat Genet 51:804-814, 2019*), respectively. Unfortunately, Liu et al are not presenting full summary statistics in their *Nature Communication* publication of preterm birth (but only reporting top two candidate genes) nor are they responsive to email request by us to access the

summary data and as such we have not been able to interrogate whether fetal genetic variation in *EPAS1* shows links to preterm birth. Instead, by accessing data from Warrington et al. we show links between fetal genetic variation and birth weight. As presented in our manuscript, the integrational analysis on all identified birth weight loci linked to 273 genes ($p < 0.001$). Interestingly, we found a significant link (1.9-fold change, $\chi^2 = 32.7$; Fisher $p < 0.0001$) between fetal genetic loci for birth weight and genes dysregulated by *EPAS1* disruption (absolute log2fold change > 1 , adjusted $p < 0.05$). Genes included not only *EPAS1* itself, but also *FLT1*, *PAPPA*, *PAPPA2*, and *CDKN1C*.

phenotype	description	variable_type	n	n_controls	n_cases
2744	Birth weight of first child	ordinal	155202		
3829	Number of stillbirths	ordinal	60453		
3839	Number of spontaneous miscarriages	ordinal	60300		
2774	Ever had stillbirth, spontaneous miscarriage or termination	binary	191252	130687	60565
20022_irnt	Birth weight	continuous_irnt	205475		

- However, to further address the comment by the reviewer, we have now expanded these integrational analyses and accessed summary statistics from GWAS of pregnancy related phenotypes available in the UK Biobank that are made available at <http://www.nealelab.is/uk-biobank> and outlined in the table below. Using GWAS data from these traits, we computed the odds ratio of the proportion of nominally significant ($P < 0.05$) hits in the *EPAS1* gene region compared with variants in a randomly selected matched window. We repeated this 1000 times for each GWAS phenotype, which yielded a distribution of odds ratios as shown in blue in the added plots. The dashed line indicated median odds ratio across permutations. To compute significance, we generated a null distribution to compare against. For each permutation, we randomly selected another region and compared this with the first randomly selected region chosen in step 1. These odds ratios are shown in gray in the added plot. We found a striking enrichment of *EPAS1* genetic variants for pregnancy loss (stillbirth and spontaneous miscarriages) compared to background as well as confirmed the link between fetal and maternal genetic variants for birthweight as shown by Warrington et al.

These new results are described in detail in a new paragraph on Page 17-18 entitled "*Genetic variants at the EPAS1 locus are linked to pregnancy complications*" as well as presented in Supplementary Figure 26; Figure 9; Supplementary Table 16-18.

REVIEWER COMMENTS

Reviewer #1 (Remarks to the Author):

This study by Varberg & Dominguez et al. provides, in principle, a most valuable study on the regulatory changes associated with differentiation of human trophoblast stem cells into the extravillous cytotrophoblast lineage. A better understanding of the shifts in transcriptional regulation will be most informative for the identification of transcription factors directing this pivotal differentiation pathway, and the genomic regulatory elements (e.g., enhancers) associated with it.

However, unfortunately, this study suffers from too many pitfalls. Thus, the flow of the text is a back and forth between genomic analyses and individual gene studies, with none of the avenues properly pursued and ultimately resulting in a picture that makes this manuscript appear very cobbled together.

Major issues revolve around the samples chosen for sequencing analysis. Albeit stating that RNA-Seq and ATAC-Seq profiles were generated for “multiple replicates and donor TS cell lines”, this is truly only the case for the RNA-seq data for which 5 replicates each of two independent TS cell lines were chosen. All other analyses are only performed on a single TS cell line, CT27. This singularity constitutes a major limitation of the biological robustness of these data. Moreover, only a single end point was chosen for investigation. The study would have benefited tremendously from at least a limited time course set-up studying the transition from stem cells towards EVT.

- Please see detailed response above.

In the struggle of this manuscript between a multi-omic and gene-specific focus, multiple aspects are left open-ended. For example, if focusing on ATAC-seq data, it would have been valuable to follow these up with H3K27ac- and H3K4me1-ChIP-seq experiments to validate and unambiguously identify stem and EVT enhancer elements. As it stands, the ATAC-seq data are valuable but not explored in sufficient detail.

- Please see detailed response above.

Lines 158/159 state that 73% of sites were shared and 27% unique to EVT. Surely there must be some TS-specific peaks (as displayed in the figures), the numbers do not add up here. Given that only half of the EVT-unique peaks fall outside a TSS region (line 165), why does Fig. 2B not indicate any accessible peaks in the heatmap?

- We apologize for the confusion in this statement and have now clarified that there are indeed TS-specific peaks as presented in Supplementary Table 3a and Figure 3A. We have also clarified this in the main text (Page 6-7).

- As shown in Figure 3A and outlined in the text, 14 % of the accessible regions that were shared between stem and EVT cells map to promoter regions. When contrasting stem versus EVT cell-specific regions, we noted that the genomic distribution of EVT cell-specific regions was shifted, with a smaller proportion mapping to promoter regions (4% versus 8%). This is reflected by the difference in transcription state site enrichment in Figure 3B.

Moreover, it does not seem adequate to analyze the genomic datasets with a singular focus on EVTs, the converse enrichment of stem cell-specific enhancers should also be included, as this is the essential starting point of these cells as they differentiate into EVTs.

- Please see detailed response above.

When it comes to the integration of TF binding sites, some fundamental information must be missing from the text (lines 179ff). What was integrated here, and from which databases, as the current study does not include any TF CHIP-seq in itself.?

- We apologize for the confusion with regards to the wording of the differential binding of accessible regions which we now have clarified. We identify statistically significant differences in peak regions as measured by read densities, which provides an indication of differentially bound sites. These results are described on Page 7.

If focusing on individual genes that stand out from the analysis, it would have been desirable to follow up any one of them in more detail than solely by ISHs on first trimester placentas, knockdown and RNA-seq. If the point is made that these genes are direct regulators of specific pathways, select factors would have been needed to be CHIP'ed to identify direct binding sites (that would be expected to show some significant overlap with ATAC-seq and HiC sites etc.).

- Please see detailed response above.

It would appear that EPAS1 is the most promising factor from all those that have been touched on. Why was a single cell study of RPL patients needed to establish a role of this TF in this condition? Why was RPL chosen (that often has an endometrial etiology) and not a more closely EVT-linked pregnancy complication, notably preeclampsia? If EPAS1 positively regulates EVT differentiation, as implied from the various genomic data, why is it up-regulated in RPL patients? This is contrary to expectations.

- The critical events underlying failed placentation which is characterized by placental hypoxia leading to preeclampsia transpire early during pregnancy (first trimester) prior to the diagnosis of preeclampsia. Obtaining first trimester placental tissues from pregnancies destined to become preeclamptic is thus difficult. However, we note that a common feature of preeclampsia is elevated levels of sFlt-1 in maternal circulation. In the present report, we identify *EPAS1* as an upstream regulator of *FLT1* and others have

shown that that hypoxia-induced activation of *EPAS1* is essential for the increased production of sFlt-1 proteins in trophoblasts (*Sasagawa et al; Sci Rep 2018 Nov 26;8(1):17375.*).

- Previous reports have utilized targeted approaches and identified links between expression changes of genes in first and second trimester placental samples and spontaneous and recurrent pregnancy losses (RPL) as well as intrauterine growth restrictions. For instance, Vasconcelos et al (*Epigenetics 14, 1234–1244, 2019*) showed that *TET3*, *IGF2* and *CDKN1C* were all upregulated in early trimester spontaneous pregnancy losses. In our report we show that these genes are not only highly EVT specific (as presented in Supplementary Table 2) but also upregulated specifically in EVT cells in RPL cases (as presented in Supplementary Table 15) corroborating previous results as well as indeed indicate a role of the studied cells in these conditions. *CDKN1C* is an important negative regulator of cell proliferation and gain of function variants in *CDKN1C* have been shown to cause growth restriction. We also show in this report that, like *EPAS1*, common genetic variants in *CDKN1C* are linked to birth weight in large GWAS (as presented in Supplementary Table 18). More importantly, *EPAS1* appears in our analysis to be an upstream regulator of *CDKN1C* as shown in our knockdown experiments (as presented in Supplementary Table 14), supporting the findings that both genes are upregulated in RPL cases. Whether the upregulation of *EPAS1* in spontaneous pregnancy loss is the cause or consequence of the condition remains to be fully elucidated. We speculate that these RPL associated changes in *EPAS1* driven EVT cell gene expression may reflect an adaptive response of a failing pregnancy. A healthy placenta can be defined by its ability to effectively adapt to stressors within the uterine environment (*Soares et al. Int J Dev Biol 58:247-259, 2014; Soares et al. Biol Reprod 99:196-211, 2018*). Pregnancy failure in RPL may be exacerbated because *EPAS1* driven adaptations are insufficient to sustain the pregnancy. Another potential mechanism could be that predisposing genetic variants cause dysregulation of *EPAS1* which leads to placenta failure and increased risk of pregnancy complications. We have revised the Results section (Page 18-19) to more clearly highlight the potential link between *EPAS1* expression and RPL.

TFAP2C is identified because of its motif enrichment in EVTs; however, the KD experiments would indicate a more prominent role in stem cell maintenance. How can this discrepancy be reconciled?

- Although motif enrichment provides valuable information for potential TF binding sites within chromatin accessible regions in each cell state, this information must also be coupled with expression of a given TF for that cell state. In the case of *TFAP2C* we observe motif enrichment in the EVT cell state, but similar transcript and protein levels in stem state and EVT differentiated cells *in vitro* at day 8. It is apparent that *TFAP2C* transcript expression is dynamic throughout the *in vitro* EVT cell differentiation process as shown in our time course analysis. *TFAP2C* co-localization in first trimester cell

columns indicates *TFAP2C* is expressed in a gradient, with higher expression in proximal column cells (less differentiated) compared to distal column cells (more differentiated). This is consistent with the observation *in vitro* that *TFAP2C* is critical to the TS cell stem state and early stages of EVT cell differentiation. As stated above, future efforts to establish an inducible system for gene silencing will enable assessment of factors, like *TFAP2C*, that may contribute to both stem state as well as differentiated cell state biology. Additionally, *TFAP2C* is part of a family of *TFAP2* transcription factors. *TFAP2A* is also expressed in the EVT cell column. *TFAP2A* expression is more prominent than *TFAP2C* in distal regions of the EVT cell column. *TFAP2C* and *TFAP2A* share DNA binding motifs. Thus, the enrichment of *TFAP2* motifs in the EVT state may be engaged by *TFAP2A* rather than *TFAP2C*.

Why was the integration of a single cell analysis needed to identify 5 EVT-enriched genes that are the single display item of Fig. 5?

- Using integrational analysis of the presented genomics data sets we used chromatin interactions measured by Hi-C to identify EVT-specific regulatory elements associated with activity of EVT-specific genes. As described on page 1 in the revised manuscript, we followed-up the top EVT cell-specific chromatin loop-gene associations and used single-cell expression data of markers of EVT cells (*in vivo* model) for orthogonal validation. Approximately 70% of these genes that were found to possess EVT-cell specific gene regulation and significantly upregulated in TS cell-derived EVT showed increased expression in the primary EVT cells from the single-cell dataset in comparison to SCT and VCT cells. Next, to identify candidate transcriptional regulators of EVT cell differentiation we classified this set of validated EVT cell-specific genes based on TF status and revealed five TF genes (*ASCL2*, *DLX6*, *SNAI1*, *MYCN*, and *EPAS1*).

Reviewer #2 (Remarks to the Author):

This of Vanberg and colleagues provides interesting insights into the regulatory DNA landscape that controls the identity and function of human extravillous trophoblast. Although the authors start with an in vitro differentiation model, validation of key observations in vivo is detailed and compelling. The paper is well-written and the figures and supplementary information are appropriate and clear. As listed below, some points require further clarification. Perhaps the most contentious issue relates to the extrapolation of the findings to the pathophysiology of recurrent pregnancy loss.

1. The authors use an in vitro system to model EVT differentiation from TS. Induced marker genes are indeed bona fide EVT genes when cross-referenced with the maternal-fetal cell atlas. However, the authors do not quantify how many differentially expressed genes in this model system are concordant/discordant in vitro versus in vivo. This is important as a significant discordance rate signals usage of differential regulatory DNA regions in culture.

- Please see detailed response above.

In view of the access to clinical samples, it seems puzzling why no attempt was made to subject isolated primary EVT cells to ATAC-seq.

- To isolate primary EVT cells from first or second trimester placental samples, we would need access to elective pregnancy terminations which is banned in our states. We do have access to placental tissues from term pregnancies and miscarriages, but they would not be satisfactory for gaining insights into EVT cells using presented approaches. EVT cells isolated from diseased placental tissue and placental specimens recovered after 9 months of pregnancy will not be effective model systems for the biological events that are associated with EVT cell differentiation in the developing placenta.

2. At a functional level, human EVT in vivo exhibits genome-wide polyploidization, cell cycle exit, and senescence (<https://doi.org/10.1371/journal.pgen.1007698>). Do these defining characteristics apply to EVT derived from TS in vitro? If so, how do endoreplication and cellular senescence impact/drive chromatin changes?

- Please see detailed response above.

3. To which extent are unique accessible loci upon EVT differentiation accounted for by co-option of (lineage-specific) transposons into the regulatory DNA network?

- This is an interesting point by the reviewer and, indeed, a recent preprint by Frost et al (preprint doi: <https://doi.org/10.1101/2022.04.26.489485>) identified multiple endogenous retrovirus (ERV) families with regulatory potential that lie close to genes with preferential expression in trophoblast. They also showed that some ERVs (such as MER11D and LTR3A) become active upon EVT cell differentiation. In an exploratory analysis, we overlapped the sequences of these two ERVs with our differentially bound EVT regulatory regions identifying a 0.2% overlap. Of note was active regulatory regions (as mapped by H3K27Ac) of the PSG (pregnancy-specific glycoproteins) locus validating results presented by Frost et al., although they link the activity of these elements to syncytiotrophoblast-specific expression. However, we show that in addition to EVT-specific chromatin activity these PSGs also show EVT-specific expression (as compared to the stem state), likely indicating these ERVs regulating PSG expression in EVT cells too. Given the relatively low overlap of these lineage-specific ERVs with our EVT-specific regulatory regions together with the lack of novelty given the results presented in the preprint, we opt to leave these results out of the current report.

4. L153: The description of the ATAC-seq peak calling indicates that TS and EVT were analysed separately, instead of peak calling on the combined data set. While the same FDR threshold was used for TS and EVT, the characteristics (height, width, footprints etc.) of so-called overlapping peaks can potentially be very different, resulting in the loss of valuable information. Likewise, it

is unclear if the unique EVT ATAC-seq peaks not previously identified in the placenta have distinct characteristics? Again, ATAC-seq of purified EVT would have been informative.

- We agree with the reviewer that combining the ATAC-Seq reads across cell state adds valuable information with regards to for instance height, width, and footprint. That is the reason we (as described on page 7) applied an alternate approach that utilized individual ATAC-Seq replicates in stem state cells (n=7) and EVT cells (n=7) taking these characteristics into account on the combined set. As described above (in response to Reviewer 1) we have now mapped these regulatory regions identified in the stem state and EVT by histone modification for the assessment of active enhancers and promoters, respectively. These new results are described in the Results section (Page 7-8) as well as presented in Figure 3.
- As explained above, while we agree with the reviewer that purified EVT cells could be informative for regulatory element mapping, it remains a significant challenge to obtain these cell types technically and because of regulatory issues, since first trimester placental tissues need to be obtained from elective terminations.

5. The authors attempt to demonstrate the relevance of their findings by focusing on recurrent pregnancy loss. Clinical definitions are based on an arbitrary number of pregnancy losses. They have no biological basis as the risk of pregnancy loss increases stepwise by 8-10% which each additional loss, independently of maternal age (see <https://pubmed.ncbi.nlm.nih.gov/33394013/>). Without suitable controls, for example, samples of aneuploid pregnancy losses or so-called 'explained' miscarriages, the authors cannot infer that induction of EPAS1 is specific to recurrent (euploid) pregnancy loss. Upregulation of EPAS1 and associated genes may merely reflect an adverse intrauterine environment imposing stress on EVT (induction of DIO2, CDKN1C etc...)

- Please see detailed response above (Reviewer #1)

6. The authors seem to construct a link between RPL and fetal genetic loci for birth weight. As over 93% of miscarriages occur before maternal-placental vascular connections are established at the end of the 1st trimester, the underlying pathological pathways are likely very different. Not only are GWAS data on miscarriage/RPL available, but there is a strong epidemiological link between the number of pregnancy losses and the risk of preterm birth. Are SNPs of EPAS1 or its target genes implicated in the genetics of miscarriage or preterm birth?

- Please see detailed response above

7. Perhaps I overlooked but the authors should present a demographic table that includes sufficient detail on the number of samples, type of miscarriage presentation, length of gestation, and previous pregnancy outcomes.

- Cohort demographic and clinical presentation is presented on Page 31 and Supplementary Table 19.

Reviewer #3 (Remarks to the Author):

In this manuscript, Varberg et al utilize a relatively recent human TSC-to-EVT (extravillous trophoblast) differentiation approach to (i) map epigenetic and transcriptional changes in the starting and end populations and (ii) to identify and validate candidate key TFs that drive this cell fate transition. Strengths of the work include the high relevance for various placenta-associated phenotypes and diseases (e.g., recurrent pregnancy loss), the use of both in vitro and in vivo samples for some validations, and the discovery of new regulators of EVT state, such as EPAS1. However, the limitations of the study are quite extensive on multiple levels, including the quality of the genomics data presented, the depth of the bioinformatics analyses, and reliability of the functional experiments. Overall, the study in its current form appears preliminary and with questionable impact and novelty. Below, are some of my key concerns:

1. *The system that the authors utilize for EVT differentiation **requires better characterization** for efficiency and homogeneity. The authors should provide a quantitation of the percentage of EVT cells (e.g., based on HLA-G or other specific marker) at the day of collection to help appreciate the potential contamination of their bulk analysis by other lineages (e.g., cytotrophoblasts or undifferentiated TS). Is sorting needed to improve homogeneity?*

- Please see detailed response above

*They should also **utilize the available scRNA-seq data from human placentas** (as shown in Figure S8) to project the EVT vs TS signatures as they identified them by bulk RNA-seq, and vice versa (perform GSEA analysis of the EVT signatures as found by the scRNA-seq data on their differential TS vs EVT analysis). This could be critical for properly interpreting their results and gain confidence on the EVT specificity that they claim throughout the text. This is also crucial to appreciate novelty of their findings, since many of these factors have been previously reported to play roles in TS differentiation but for other lineages.*

- Please see detailed response above

2. *There are several issues with ATAC-seq and Hi-C data that the authors generate and present for TS and TS-derived EVT cells. First of all, **more information and some examples of the raw data** (e.g., more examples of ATAC-seq tracks for shared and unique peaks; tornado plots of all peaks in TS and EVT clustered by category; read numbers, QC and actual heatmaps of the Hi-C data) would be very informative and help the reader appreciate the quality of the results or suggest improvements.*

- We have added information about the raw data in Supplementary Table 1 and now present numerous examples of the data (ATAC-Seq, RNA-Seq, ChIP-Seq and Hi-C) using updated figure formats either as main figures or supplementary figure. We have also

updated our UCSC session to now also visualize the Hi-C data.

*In addition, there are so many well-established methods that allow a better integration of these datasets (e.g., **aggregate plots of HiC signal around shared or cell-type specific peaks, virtual 4C around genes of interest**) that are underutilized here. Also, as the authors acknowledge based on their results, HiC is limited in detecting potential regulatory loops. Therefore, the authors should perform a different orthogonal analysis, at least 4C-seq around genes or SE of interest to validate loops or detect cell-type specific connections that are likely missed by HiC.*

- We have now performed orthogonal analysis of our Hi-C loops identified in the CT27 line by including similar data from the CT29 line. These new results are presented on Page 9 and in Supplementary Figure 7; Supplementary Figure 11; Supplementary Figure 12; Supplementary Figure 14; Supplementary Figure 16; Supplementary Figure 18; and Supplementary Figure 21A.
- We now also present aggregate plot of Hi-C signals around specific regions in Supplementary Figure 9E.

3. The authors move on with several functional validations of candidate TFs in the TS-to-EVT differentiation system. Although, many of phenotypes are strong and interesting, many of their conclusions are compromised by the constitutive expression of shRNAs and the limited quantitation of the phenotypic effects -other than bulk RNA-seq. For example, for TFAP2C -given that this factor is important both TS biology, an inducible deletion system will be important to uncouple effects on the starting cells versus the potential role in differentiation. Moreover, a detailed quantitation of all the possible lineages (TS, EVT, other potential byproducts) based on IF analysis (or intercellular FACs) of multiple independent replicates will be very important to assess reproducibility and understand the specific role of each factor.

- Please see detailed response above

4. In general, although the manuscript is easy to read and understand, it is hard to follow the logic on the different types of integrative analysis that they used to nominate the few candidates that the focus on. A better organization of the manuscript with the integrative analysis first, followed by all in situ validations and then functional interrogations will improve the flow and help the reader compare and contrast results.

- We have re-organized the manuscript and now present the integrative analysis first followed by in situ validations and functional interrogations. We agree with the reviewer that this new presentation improved the flow of the paper aiding the reader to contrast the results better.

5. The figures overall seem very preliminary with superficial genomics analysis and poor presentation of the results.

- We have improved the presentations of the results throughout the paper and added significant amount of new data and analyses as described above.

REVIEWER COMMENTS

Reviewer #1 (Remarks to the Author):

The authors have performed extensive revisions to their manuscript which has added substantial amounts of high-throughput data to their previous submission, in particular ChIP-seq data. It remains the case that the most significant value of this study is as a resource, while the concrete novelty of insights remains relatively sparse. Nevertheless, this manuscript is now significantly improved.

The authors state that in response to the points raised, they now added the analysis of a second cell line, CT29, to their datasets. However, the number of differentially expressed genes remains identical to their previous submission, page 5: "...2267 transcripts that were significantly upregulated (log2 fold change >1, adjusted $p < 0.05$) in EVT cells compared to trophoblast cells in the stem state and 2284 transcripts that were significantly downregulated". This is virtually impossible, and I would doubt that these numbers truly reflect the required re-analysis of data after integration of a whole new series of data. This point needs to be revised.

Secondly, the description around the bivalent domains is confusing and as stated incorrectly described: "Interestingly, differentially enriched ATAC-Seq regions in the stem state were an order of magnitude more likely to map to a poised/bivalent promoter (marked by a H3K4me3 peak without an overlapping H3K27ac peak) than differentially enriched ATAC-Seq regions in the EVT cell state (Fig. 3c-d). Bivalent chromatin are characterized by

having both repressing and activating role of gene expression regulation..."

H3K4me3 peaks demarcate active promoters, not poised/bivalent promoters. The authors did not include a repressive chromatin mark, in this case the relevant mark would have been H3K27me3. So how were poised promoters called? It cannot/should not have been done based on H3K4me3 and H3K27ac peaks.

Reviewer #2 (Remarks to the Author):

The authors have adequately addressed/rebutted my concerns.

Reviewer #3 (Remarks to the Author):

The authors have performed an extensive revision including new experiments, timepoints, replicates and bioinformatics analysis to address the multiple concerns raised by all reviewers. The manuscript is certainly in much better shape than the original one. Their efforts addressed in a relatively satisfactory manner some of my main points. The characterization of the differentiation systems is now much more complete and convincing. The new presentation of multiple tracks and examples has been certainly useful to appreciate the quality of their ATAC-seq data. I was also happy to see that in response to other reviewer's comments, the authors have now added additional valuable genomics data (H3K27ac and H3K4me3). However, the entire integrative analysis remains very minimalistic and confusing. The comparison between in vitro and in vivo states should also be strengthened. Here are some specific points that I hope will help them improve this part of the manuscript:

-Generate one dedicated figure on the integration of ATAC-seq and ChIP-seq data:

(i) Generate and present an atlas ATAC-seq peaks for all samples and cluster them into stem-specific, EVT-specific and shared (currently the tornado in figure 3 shows NO signal in any of the stem or SVT specific peaks....).

(ii) Do something similar for the H3K27ac peaks to nominate cell-type specific H3K27ac peaks as candidate enhancers (or super enhancers).

(iii) do and present in the main figure (in addition to the suppl table) the key results from the motif enrichment analysis both for cell-type specific ATAC-seq and H3K27ac peaks to nominate candidate TFs.

-Generate a dedicated figure for HiC analysis:

(i) present some type of heatmap presenting all loops showing the relative strength between common, unique or enriched loops in each cell type.

(ii) show examples of actual normalized contact heatmaps (using Juicebox, higlass or any other software they prefer) along with the arcs of their called loops

(iii) generate APA plots around common or cell-type specific ATAC-seq peaks and H3K27ac Peaks using both the EVT and stem cell HiC datasets (total or 6 APA) followed by some locus-specific examples. The importance of this point is to support their title "Long-range chromatin interactions define unique gene regulation machinery in EVT cells". Currently, based on the very low numbers of EVT-specific contacts and the lack of convincing examples, this conclusion is not well-supported. 4C-seq around key example genes will be the ideal approach to prove rewiring.

-Although in Fig.2 they tried to integrate and compare the DEGs with in vivo EVT signatures, this analysis needs to be strengthened and presented in a more clear manner. How does the reader know that the signatures they called are indeed what they are supposed to? A simple projection of each gene signature on the tSNE plot to show the specific enrichment of the respective population, would be very useful.

More importantly, the authors should project the top DEGs from their in vitro comparison on the in vivo clusters. Do we see a clear enrichment on the EVT (or additional) clusters?

In general, although the manuscript has been drastically improved, and the message remains important and relevant, the study requires more rigorous analysis and better figure presentation for publication to this journal.

We are pleased to see the mutual enthusiasm from all Reviewers about our extensive revision.

- Reviewer #1 states that **“The authors have performed extensive revisions to their manuscript which has added substantial amounts of high-throughput data to their previous submission, in particular ChIP-seq data”**.
- Reviewer #2 has no further request as indicated by the statement: **“The authors have adequately addressed/rebutted my concerns.”**
- Reviewer #3 notes that **“The authors have performed an extensive revision including new experiments, timepoints, replicates and bioinformatics analysis to address the multiple concerns raised by all reviewers. “**

We also thank them again for their careful second review and address the additional comments raised by Reviewer#1 and #3, respectively, as outlined further below.

REVIEWER COMMENTS

Reviewer #1 (Remarks to the Author):

“The authors have performed extensive revisions to their manuscript which has added substantial amounts of high-throughput data to their previous submission, in particular ChIP-seq data. It remains the case that the most significant value of this study is as a resource, while the concrete novelty of insights remains relatively sparse. Nevertheless, this manuscript is now significantly improved.”

- We thank the reviewer again for their careful review and are pleased to see that our extensive revision was to large extent satisfactory.

“The authors state that in response to the points raised, they now added the analysis of a second cell line, CT29, to their datasets. However, the number of differentially expressed genes remains identical to their previous submission, page 5: “...2267 transcripts that were significantly upregulated (log2 fold change >1, adjusted p<0.05) in EVT cells compared to trophoblast cells in the stem state and 2284 transcripts that were significantly downregulated”. This is virtually impossible, and I would doubt that these numbers truly reflect the required re-analysis of data after integration of a whole new series of data. This point needs to be revised.”

- We apologize for this omission, which we now have corrected. These changes can be found on Page 5, line 16-19.

“Secondly, the description around the bivalent domains is confusing and as stated incorrectly described: “Interestingly, differentially enriched ATAC-Seq regions in the stem state were an order of magnitude more likely to map to a poised/bivalent promoter (marked by a H3K4me3

peak without an overlapping H3K27ac peak) than differentially enriched ATAC-Seq regions in the EVT cell state (Fig. 3c-d). Bivalent chromatin are characterized by having both repressing and activating role of gene expression regulation..." H3K4me3 peaks demarcate active promoters, not poised/bivalent promoters. The authors did not include a repressive chromatin mark, in this case the relevant mark would have been H3K27me3. So how were poised promoters called? It cannot/should not have been done based on H3K4me3 and H3K27ac peaks."

- Although we agree with the reviewer that poised/bivalent promoters are typically called by co-occurrence of H3K4me3 and H3K27me3 we respectfully disagree that H3K4me3 alone demarcate an active promoter. As shown by the NIH Roadmap Epigenomics Consortium in their flagship paper "Integrative analysis of 111 reference human epigenomes" (Nature 2015), active promoters are characterized by co-occurrence of H3K4me3 and H3K27ac. On the other hand, and as shown by the Roadmap Consortium, poised/bivalent promoters are not marked by H3K27ac. As our ChIP-Seq data was restricted to the most common and robust histone marks (H3K4me3 and H3K27ac) we used these to infer promoter states accordingly. While we maintain our classifications of active promoters, we have now revised the text describing the poised/bivalent state to clarify this. These changes can be found on Page 8 (line 23-25) in the revised version of the paper.

Reviewer #2 (Remarks to the Author):

"The authors have adequately addressed/rebutted my concerns."

- We thank the reviewer again for their careful review and are pleased to see that our extensive revision was satisfactory.

Reviewer #3 (Remarks to the Author):

"The authors have performed an extensive revision including new experiments, timepoints, replicates and bioinformatics analysis to address the multiple concerns raised by all reviewers. The manuscript is certainly in much better shape than the original one. Their efforts addressed in a relatively satisfactory manner some of my main points. The characterization of the differentiation systems is now much more complete and convincing. The new presentation of multiple tracks and examples has been certainly useful to appreciate the quality of their ATAC-seq data. I was also happy to see that in response to other reviewer's comments, the authors have now added additional valuable genomics data (H3K27ac and H3K4me3)."

- We thank the reviewer again for their careful review and are pleased to see that our extensive revision was to large extent satisfactory.

However, the entire integrative analysis remains very minimalistic and confusing. The comparison between in vitro and in vivo states should also be strengthened.

- We respectively disagree with the reviewer that our integrative analyses presented are ‘very minimalistic’. We present detailed annotation of ATAC-Seq peaks in EVT and stem states (**Figure 3a**) not only using conventional genomic element types (e.g., exon, intron, TSS etc.) but also by incorporating the largest to date atlas of regulatory DNA created from 16 different human cell types and states (*Meuleman et al, Nature 2020*). We show that shared regulatory regions between EVT and stem states are mapping more frequently to promoter regions which are accurately visualized and presented in tornado plots where coverage data is centered around TSS (**Figure 3b**). In addition, in response to Reviewer#1, we now also annotate our identified regions by orthogonal assessment of common histone modifications marking regulatory element activity (H3K4me3 and H3K27ac). These annotations reveal a more enhancer-driven regulation in EVT cells whereas stem state cells appear to have primarily regulatory elements mapping to promoters (**Figure 3c-d**). We provide numerous examples of key genes where multi-layered data sets are presented as track-based figures (**Figure 4; Figure 8; Figure 9; Supplementary Figures 5-6; Supplementary Figures 8-16; Supplementary Figure 19**). Nevertheless, we have now added additional figures in response to the reviewer’s request as outlined below.

Here are some specific points that I hope will help them improve this part of the manuscript:

-Generate one dedicated figure on the integration of ATAC-seq and ChIP-seq data:

- (i) Generate and present an atlas ATAC-seq peaks for all samples and cluster them into stem-specific, EVT-specific and shared (currently the tornado in figure 3 shows NO signal in any of the stem or SVT specific peaks....).*
- (ii) Do something similar for the H3K27ac peaks to nominate cell-type specific H3K27ac peaks as candidate enhancers (or super enhancers).*
- (iii) do and present in the main figure (in addition to the suppl table) the key results from the motif enrichment analysis both for cell-type specific ATAC-seq and H3K27ac peaks to nominate candidate TFs.*

- As indicated above, the tornado plot (**Figure 3b**) shows an expected pattern of cell-specific versus shared ATAC-Seq coverage when centered around TSS. Specifically, as outlined in the figure legend, heat maps show the density of chromatin accessible regions within 10 kb of TSS that are unique to EVT cells, unique to stem state cells, or shared between the two cell states. Thus, these tornado plots are supporting our

findings that cell-specific ATAC-Seq regions are mapping less frequently to promoter regions compared to regions shared across cell states. We discuss these findings in detail on page 7 (line 1-12) in addition to **Figure 3**.

- We have now generated additional tornado plots integrating ATAC-seq and ChIP-seq data as presented in **Figure 3c** and discussed on Page 7 (line 20-23)
- We have now generated a schematic depicting how candidate regulatory regions were identified (based on the integration of super enhancers, ATAC-Seq and H3K27ac peaks, respectively) which is presented in **Figure 3f** and discussed on Page 11 (line 18-24).

-Generate a dedicated figure for HiC analysis:

- (i) present some type of heatmap presenting all loops showing the relative strength between common, unique or enriched loops in each cell type.
 - (ii) show examples of actual normalized contact heatmaps (using Juicebox, hiclass or any other software they prefer) along with the arcs of their called loops
 - (iii) generate APA plots around common or cell-type specific ATAC-seq peaks and H3K27ac Peaks using both the EVT and stem cell HiC datasets (total or 6 APA) followed by some locus-specific examples. The importance of this point is to support their title "Long-range chromatin interactions define unique gene regulation machinery in EVT cells". Currently, based on the very low numbers of EVT-specific contacts and the lack of convincing examples, this conclusion is not well-supported. 4C-seq around key example genes will be the ideal approach to prove rewiring.
- We have updated the presentation of the HiC results with a dedicated figure for HiC analysis (**Figure 4**) including stem and EVT heatmaps of a) all contacts across all chromosomes, b) contacts and loops (shared and cell-specific) exemplified for one chromosome, c) further zoomed in contacts for candidate regions showing differential looping, d) an aggregate plot showing the integration of EVT cell differentially accessed regulatory regions and EVT cell loop anchors and e) further visualization of the locus-specific example in c) showing unique gene regulation machinery in EVT cells that we validate across Hi-C datasets. The latter is presented in **Supplementary Figure 15**.

“Although in Fig.2 they tried to integrate and compare the DEGs with in vivo EVT signatures, this analysis needs to be strengthened and presented in a more clear manner. How does the reader know that the signatures they called are indeed what they are supposed to? A simple projection of each gene signature on the tSNE plot to show the specific enrichment of the respective population, would be very useful. More importantly, the authors should project the top DEGs from their in vitro comparison on the in vivo clusters. Do we see a clear enrichment on the EVT (or additional) clusters? “

- We have now updated **Figure 2** to further highlight similarities between the *in vitro* and *in vivo* data used. Specifically, we now project the top differently expressed genes from the TS model in *the vivo* clusters showing clear enrichment in EVT single cells. We further note that the paper is already showing similar projections for novel genes that we followed-up in more detail as presented in **Figure 5**.

REVIEWER COMMENTS

Reviewer #1 (Remarks to the Author):

The authors have addressed the final revisions in a satisfactory manner, although some of the graphs that Reviewer 3 suggested would have been useful and nice to see. Nevertheless, this is a comprehensive analysis that will be very valuable for the field to better understand the genomic regulatory processes during trophoblast differentiation.

Reviewer #3 (Remarks to the Author):

Unfortunately, most of my suggestions remain poorly addressed and I am not sure if it is due to simple miscommunication or lack of relevant bioinformatics expertise in the team. I tried to be as specific as possible to help with the representation of their data, but none of the revised graphs addressed my points. For example, when I am talking about the atlas of ATAC-seq peaks, I mean the unity of all called ATAC-seq peaks either in ETV or stem cells. People usually rank all peaks in their preferred manner (eg. strongest to weakest in the stem cells) and then they present the respective ATAC-seq signal centered on the peaks in both cell lines side-by-side, so readers can directly see and compare the relative signal in the two (or more lines) and be convinced that the ETV-specific peaks are more accessible in ETV cells, and vice versa. The representation in Figure 2b is still very confusing. The 3 tornado plots presented side by side, have -likely- nothing to do with each other. The confusing representation, the poor labeling (very small font or description missing) and the unclear message is a consistent problem in many figure panels. For example:

Figure 3c: why do they represent the ATAC-seq signal around these regions? What does this analysis offer? My suggestion was to perform some kind of clustering or differential analysis of all ChIP-seq peaks?

Figure 3f: Some quantitation/stats of the most enriched motifs either in ETV- or stem-specific peaks would have been much more meaningful (p-value, enrichment score ...)

Fig.4a: is completely uninformative.

Figure 4b: what do they want to show there? Reorganization on the compartment level? Again quantitation and labeling is missing...

Figure 4c: That could be promising with a better representation and quantitation (e.g plotting one sample on the top of the diagonal and the other at the bottom or better by plotting directly the differential HiC signal)

Figure 4d: Again the same problem. Why do they only show the aggregate HiC signal only on ETV cells and only centered around ETV-specific peaks. What is completely missing in Figure 4 is some robust statistical analysis and representation of the conserved vs differential loops in each cell type.

Figure 2b: Representation of individual genes is nice, but projection of the entire signature would be much more informative, as I suggested before.

AUTHOR COMMENTS

We are pleased to see that both Reviewer#1 and Reviewer#2 have no further requests and are supportive of our extensive revisions. We thank them again for their careful review and for their suggested edits, which we believe significantly improved the manuscript.

In this third revision, we address the remaining concerns presented by Reviewer#3 in point-by-point responses below. We very much appreciate Reviewer#3's sincere help in trying to improve our manuscript. We have provided additional figures, new analysis and clarifications including increasing font sizes and labeling as requested.

REVIEWER COMMENTS

Reviewer #3 (Remarks to the Author):

“Unfortunately, most of my suggestions remain poorly addressed and I am not sure if it is due to simple miscommunication or lack of relevant bioinformatics expertise in the team. I tried to be as specific as possible to help with the representation of their data, but none of the revised graphs addressed my points. For example, when I am talking about the atlas of ATAC-seq peaks, I mean the unity of all called ATAC-seq peaks either in ETV or stem cells. People usually rank all peaks in their preferred manner (eg. strongest to weakest in the stem cells) and then they present the respective ATAC-seq signal centered on the peaks in both cell lines side-by-side, so readers can directly see and compare the relative signal in the two (or more lines) and be convinced that the ETV-specific peaks are more accessible in ETV cells, and vice versa.”

- While these suggestions are valid, the logic for how the figures were organized and the way the data has been represented was driven by the key findings described in the Results. Our perspective is that these visual graphics are the best and most effective ways to convey the key points. Nevertheless, we have modified this section further to highlight ATAC-Seq coverage across cell state and peaks. Please see new **Figure 3a** to the right and in the main documents.

“The representation in Figure 2b is still very confusing. The 3 tornado plots presented side by side, have -likely- nothing to do with each other. The confusing representation, the poor labeling (very small font or description missing) and the unclear message is a consistent problem in many figure panels. “

- We assume the Reviewer#3 is referring to the previously labeled Figure 3b. We strongly disagree with the statement that the plots “have -likely- nothing to do with each other”. As clearly stated in the previous revision, this tornado plot (including density plot and heatmap) shows an expected pattern of cell state-specific versus shared ATAC-Seq coverage when centered around transcription start sites (TSS). Specifically, as outlined in the figure legend, heat maps show the density of chromatin accessible regions within 10 kb of TSS that are 1) unique to EVT cells, 2) unique to stem state cells, or 3) shared between the two cell states. Thus, these tornado plots are supporting our findings that cell-specific ATAC-Seq regions are mapping less frequently to promoter regions compared to regions shared across cell states. We further show that this shift supports the finding of increased enhancer-driven gene regulation in EVT versus stem state cells. We discuss these findings in detail on **page 7 (line 1-12)** in addition to **Figure 3**. We kindly direct the Reviewer to this part of the Results section, as it serves an important aspect of our findings.
- We have increased the fonts for improved clarity as shown to the right and in the main documents.

“Figure 3c: why do they represent the ATAC-seq signal around these regions? What does this analysis offer? My suggestion was to perform some kind of clustering or differential analysis of all ChIP-seq peaks?”

- We state both in the Results section and in the figure legend that we use H3K4me3 and H3K27ac marks by ChIP-Seq for orthogonal validation and to further characterize the regulatory regions we identified by ATAC-Seq. We visually represent coverage around ChIP-Seq peaks that show enrichment of ATAC-Seq reads using tornado plots, thus, validating our discovery genomic data sets. As indicated above, we have added a new **Figure 3c** per the reviewer’s request and have shifted the presentation of the relevant data to **Supplementary Figure 5a** shown below on the next page.

“Figure 3f: Some quantitation/stats of the most enriched motifs either in ETV- or stem-specific peaks would have been much more meaningful (p-value, enrichment score …)”

- All quantitation/statistics are available across the four sets tested in **Supplementary Table 9**. These issues have now been further addressed on the behest of the reviewer. As such, **Figure 3f** has been updated accordingly with the panel shown below:

f

Motif	Rank	Name	p-value	log p-value	q-value (Benjamini)	# Target Sequences with Motif	% of Targets Sequences with Motif	# Background Sequences with Motif	% of Background Sequences with Motif
	1	TEAD3(TEA)/HepG2-TEAD3-ChIP-Seq(Encode)/Homer	1.00E-35	-8.25E+01	0	172	28.15%	4867.3	9.97%
	2	TEAD1(TEAD)/HepG2-TEAD1-ChIP-Seq(Encode)/Homer	1.00E-31	-7.35E+01	0	152	24.88%	4233.1	8.67%
	3	TEAD4(TEA)/Tropoblast-Tead4-ChIP-Seq(GSE37350)/Homer	1.00E-30	-7.09E+01	0	144	23.57%	3943.3	8.08%
	4	AP-2gamma(AP2)/MCF7-TFAP2C-ChIP-Seq(GSE21234)/Homer	1.00E-27	-6.31E+01	0	196	32.08%	7073.6	14.49%

“Fig.4a: is completely uninformative.”

- This figure demonstrates the high conservation of genome-wide chromatin contacts across cell types (**originally identified by Rao et al Cell 2014 who use similar visualization scheme as presented here**). However, to improve visualization and increase the resolution further we have decided to show contact maps across cell states for a representative chromosome (**new Figure 4a, see below and in main documents**) which we further zoom in at higher resolution (**new Figure 4b, see below and in main documents**). These visualization schemes are commonly used to exemplify chromatin contacts and its conservation (see **Rao et al, Cell 2014** or more recently **Greenwald et al, Nature Communications 2019**).

“Figure 4b: what do they want to show there? Reorganization on the compartment level? Again quantitation and labeling is missing...”

- As outlined in the associated figure legend, **Figure 4b** shows contact maps for a representative chromosome in stem state and EVT cells. We contrast the maps to highlight high conservation of chromatin loops (shown by black squares) and annotate the map with corresponding ATAC-Seq coverage in respective cell state in 2D. In this resubmission, we have further improved the figure by highlighting a conserved regulatory element overlapping chromatin anchors across both cell states as well as added labels and further clarified text in the figure legend. In addition, we now visualize differential loops in yellow instead of the darker blue color.

“Figure 4c: That could be promising with a better representation and quantitation (e.g plotting one sample on the top of the diagonal and the other at the bottom or better by plotting directly the differential HiC signal)”

- Although plotting the differential loops identified in stem state vs EVT cells in the same map is plausible, it becomes challenging to associate respective loop anchors to regulatory elements in 2D as these regulatory regions differ across cell state. Importantly, the purpose of **Figure 4b** is to exemplify an EVT-specific chromatin loop mapping to EVT-specific regulatory elements at both anchors which the reviewer requested in the previous revision. This locus corresponds to *DLX6* – a novel transcriptional regulator of EVT cell differentiation that we discuss in further detail in the Results section (**please see Page 12, Figure 4g, Supplementary Figure 16**). Instead, we believe plotting the two maps side by side with loops highlighted accordingly is preferable for the reader and for proper interpretations. We again refer to **Greenwald et al, Nature Communications 2019 (Figure 2a)** who present similar visualization across

cell states to contrast loops. We have further improved the figures as requested by the reviewer and we refer to the legend for details about quantitation.

“Figure 4d: Again the same problem. Why do they only show the aggregate HiC signal only on EVT cells and only centered around EVT-specific peaks. What is completely missing in Figure 4 is some robust statistical analysis and representation of the conserved vs differential loops in each cell type. “

- As indicated above, the logic for how these figures were organized and the way the data has been represented was driven by the key findings described in the Results. Again, our perspective is that these visual graphics are the best and most effective ways to convey the key points. As presented on **Page 11** we referred to density plots of the distribution of epigenetic marks relative to the center of loop anchors for all loops analyzed (4 sets) and not only EVT-specific loops. We note that density plots are not only commonly used for this purpose (e.g., see Greenwald et al, Nature Communications 2019) but also more informative than aggregate plots due to the small number of differential loops identified. Thus, we opt to visualize aggregation of epigenetic data around loop anchors across all sets of contact maps in density plots rather than using aggregate plots. These are presented in Supplementary Figure 9 which is included below and in supportive documents.
- However, in our modified **Figure 4** we now show density plots for the Stem state and EVT loop anchors in relation to coverage of respective ATAC-Seq data and in the extended **Supplementary Figure 9**.
- In addition, we have also performed new analysis as requested to contrast the function of all versus cell state-specific loop anchors and their relationship to gene regulation. Specifically, we integrated gene expression data from stem state and EVT cells, respectively, mapping close to the four different sets of loop anchors. We found that loop anchors were enriched (Chi2 test $p < 10^{-4}$) near differentially expressed genes (adjusted $p < 10^{-5}$) and the enrichment was marginally stronger for genes upregulated in the respective cell state. Interestingly, this trend was more pronounced for cell-specific (differential) loop anchors especially those identified in EVT cells (Fisher’s exact test $p < 0.05$). Among these upregulated genes with long-range chromatin interactions identified only in EVT cells was *DLX6* that we show in Figure 4 representing potentially a novel regulator of EVT differentiation and function and is a candidate for further experimentation. We discuss that these results confirm previous findings that cell state-specific (differential) loops associate with cell-type specific regulation and functions. These new figures are presented in **Figure 4** and **Supplementary Figure 9** which are included below and in submitted documents.

Figure 4:

Supplementary Figure 9

“Figure 2b: Representation of individual genes is nice, but projection of the entire signature would be much more informative, as I suggested before.”

- We appreciate the positive feedback for our improved single-cell plots showing validation of our *in vitro* data in primary placental tissue. We defer to **Figure 2c-e** for projection of *in vivo* data.

AUTHOR COMMENTS

We are pleased to learn that Reviewer #4 is of the opinion that the dataset will be of substantial interest to the scientific community. We thank them for their review and suggested edits, which we believe improved the manuscript.

In this fourth revision, we address the remaining concerns presented by Reviewer#4 in point-by-point responses below. We very much appreciate Reviewer#4's help in trying to improve our manuscript. We have provided clarifications and modified manuscript text as requested.

REVIEWER COMMENTS

Reviewer #4 (Remarks to the Author):

1. ***“Regarding figure 3 and respective results section. Please indicate in the manuscript exactly how the peaks in the ATAC-seq heatmaps were ranked in Figures 3a and 3c and Supplementary Figure 5a.”***

We have now clarified in each figure legend how the peaks in the heatmaps were ranked.

2. ***“We appreciate that the data shows that chromatin accessibility differs in differentiated EVT cells compared to stem cells. However, in our view, we think it would be somewhat premature to claim directly that “EVT cell differentiation DRIVES global changes in chromatin accessibility” or that “Active remodeling of the chromatin landscape DIRECTS extravillous trophoblast cell lineage development” as this would appear to imply causation between chromatin changes and differentiation. We would instead suggest that you state that changes in chromatin accessibility correlate or co-occur with EVT differentiation. Please edit the manuscript accordingly, including the title, abstract and subsection headings.”***

The manuscript text (including title, abstract, subsection headings, and figure titles) have been edited to remove words such as ‘drives’ and ‘directs’ to language such as ‘co-occurs’ or ‘is associated with’.

3. ***“Regarding figure 4 and respective results section. Please describe in the manuscript why these particular regions were selected for the figure as representative in Figure 4a and 4b.”***

We have now clarified in the figure legend how the particular regions were selected.

4. ***“Please modify the statement “Long-range chromatin interactions define unique gene regulation machinery in EVT cells”. Similarly to the comment above, in our view, the data supports correlation rather than causation between differential chromatin interactions and gene regulation.”***

The statement has been modified accordingly.